# AN INFORMATION-THEORETIC FRAMEWORK FOR LEARNING MODELS OF INSTANCE-INDEPENDENT LABEL NOISE

## ABSTRACT

Given a dataset $\mathcal{D}$ with label noise, how do we learn its underlying noise model? If we assume that the label noise is instance-independent, then the noise model can be represented by a noise transition matrix $Q_{\mathcal{D}}$. Recent work has shown that even without further information about any instances with correct labels, or further assumptions on the distribution of the label noise, it is still possible to estimate $Q_{\mathcal{D}}$ while simultaneously learning a classifier from $\mathcal{D}$. However, this presupposes that a good estimate of $Q_{\mathcal{D}}$ requires an accurate classifier. In this paper, we show that high classification accuracy is actually not required for estimating $Q_{\mathcal{D}}$ well. We shall introduce an information-theoretic-based framework for estimating $Q_{\mathcal{D}}$ solely from $\mathcal{D}$ (without additional information or assumptions). At the heart of our framework is a discriminator that predicts whether an input dataset has maximum Shannon entropy, which shall be used on multiple new datasets $\hat{\mathcal{D}}$ synthesized from $\mathcal{D}$ via the insertion of additional label noise. We prove that our estimator for $Q_{\mathcal{D}}$ is statistically consistent, in terms of dataset size, and the number of intermediate datasets $\hat{\mathcal{D}}$ synthesized from $\mathcal{D}$. As a concrete realization of our framework, we shall incorporate local intrinsic dimensionality (LID) into the discriminator, and we show experimentally that with our LID-based discriminator, the estimation error for $Q_{\mathcal{D}}$ can be significantly reduced. We achieved average Kullback–Leibler (KL) loss reduction from $0.27$ to $0.17$ for $40\%$ anchor-like samples removal when evaluated on CIFAR10 with symmetric noise. Although no clean subset of $\mathcal{D}$ is required for our framework to work, we show that our framework can also take advantage of clean data to improve upon existing estimation methods.

## 1 INTRODUCTION

Real-world datasets are inherently noisy. Although there are numerous existing methods for learning classifiers in the presence of label noise (e.g. Han et al. (2018); Hendrycks et al. (2018); Natarajan et al. (2013); Tanaka et al. (2018)), there is still a gap between empirical success and theoretical understanding of conditions required for these methods to work. For instance-independent label noise, all methods with theoretical performance guarantees require a good estimation of the noise transition matrix as a key indispensable step (Cheng et al., 2017; Jindal et al., 2016; Patrini et al., 2017; Thekumparampil et al., 2018; Xia et al., 2019). Recall that for any dataset $\mathcal{D}$ with label noise, we can associate to it a *noise transition matrix* $Q_{\mathcal{D}}$, whose entries are conditional probabilities $p(y|z)$ that a randomly selected instance of $\mathcal{D}$ has the given label $y$, under the condition that its correct label is $z$. Many algorithms for estimating $Q_{\mathcal{D}}$ either require that a small clean subset $\mathcal{D}_{\text{clean}}$ of $\mathcal{D}$ is provided (Liu & Tao, 2015; Scott, 2015), or assume that the noise model is a mixture model (Ramaswamy et al., 2016; Yu et al., 2018), where at least some anchor points are known for every component. Here, "anchor points" refer to datapoints belonging to exactly one component of the mixture model almost surely (cf. Vandermeulen et al. (2019)), while "clean" refers to instances with correct labels.

Recently, it was shown that the knowledge of anchor points or $\mathcal{D}_{\text{clean}}$ is not required for estimating $Q_{\mathcal{D}}$. The proposed approach, known as T-Revision (Xia et al., 2019), learns a classifier from $\mathcal{D}$ and simultaneously identifies anchor-like instances in $\mathcal{D}$, which are used iteratively to estimate $Q_{\mathcal{D}}$, which in turn is used to improve the classifier. Hence for T-Revision, a good estimation for $Q_{\mathcal{D}}$ is inextricably tied to learning a classifier with high classification accuracy. In this paper, we propose a

framework for estimating $Q_{\mathcal{D}}$ solely from $\mathcal{D}$, without requiring anchor points, a clean subset, or even anchor-like instances. In particular, we show that high classification accuracy is not required for a good estimation of $Q_{\mathcal{D}}$. Our framework is able to robustly estimate $Q_{\mathcal{D}}$ at all noise levels, even in extreme scenarios where anchor points are removed from $\mathcal{D}$, or where $\mathcal{D}$ is imbalanced.

Our key starting point is that Shannon entropy and other related information-theoretic concepts can be defined analogously for datasets with label noise. Suppose we have a discriminator $\Phi$ that takes any dataset $\mathcal{D}'$ as its input, and gives a binary output that predicts whether $\mathcal{D}'$ has maximum entropy. Given $\mathcal{D}$, a dataset with label noise, we shall synthesize multiple new datasets $\hat{\mathcal{D}}$ by inserting additional label noise into $\mathcal{D}$, using different noise levels for different label classes. Intuitively, the more label noise that $\mathcal{D}$ initially has, the lower the *minimum* amount of additional label noise we need to insert into $\mathcal{D}$ to reach near-maximum entropy. We show that among those datasets $\hat{\mathcal{D}}$ that are predicted by $\Phi$ to have maximum entropy, their associated levels of additional label noise can be used to compute a single estimate for $Q_{\mathcal{D}}$. Our estimator is statistically consistent: We prove that by repeating this method, the average of the estimates would converge to the true $Q_{\mathcal{D}}$.

As a concrete realization of this idea, we shall construct $\Phi$ using the notion of Local Intrinsic Dimensionality (LID) (Houle, 2013; 2017a;b). Intuitively, the LID computed at a feature vector $\mathbf{v}$ is an approximation of the dimension of a smooth manifold containing $\mathbf{v}$ that would "best" fit the distribution $\mathcal{D}$ in the vicinity of $\mathbf{v}$. LID plays a fundamental role in an important 2018 breakthrough in noise detection (Ma et al., 2018c), wherein it was empirically shown that sequences of LID scores could be used to distinguish clean datasets from datasets with label noise. Roughly speaking, the training data for $\Phi$ consists of LID sequences that correspond to multiple datasets synthesized from $\mathcal{D}$. In particular, we show that $\Phi$ can be trained *without* needing any clean data. Since we are optimizing the predictive accuracy of $\Phi$, rather than optimizing the classification accuracy for $\mathcal{D}$, we also do not require state-of-the-art architectures. For example, in our experiments on the CIFAR-10 dataset (Krizhevsky et al., 2009), we found that LID sequences generated by training on shallow "vanilla" convolutional neural networks (CNNs), were sufficient for training $\Phi$.

Our contributions are summarized as follows:

- We introduce an information-theoretic-based framework for estimating the noise transition matrix of any dataset $\mathcal{D}$ with instance-independent label noise. We do not make any assumptions on the structure of the noise transition matrix.

- We prove that our noise transition matrix estimator is consistent. This is the first-ever estimator that is proven to be consistent without needing to optimize classification accuracy. Notably, our consistency proof does not require anchor points, a clean subset, or any anchor-like instances.

- We construct an LID-based discriminator $\Phi$ and show experimentally that training a shallow CNN to generate LID sequences is sufficient for obtaining high predictive accuracy for $\Phi$. Using our LID-based discriminator $\Phi$, our proposed estimator outperforms the state-of-the-art methods, especially in the case when anchor-like instances are removed from $\mathcal{D}$.

- Given access to a clean subset $\mathcal{D}_{\mathrm{clean}}$, we show that our method can be used to further improve existing competitive estimation methods.

## 2 PROPOSED INFORMATION-THEORETIC FRAMEWORK

Our framework hinges on a simple yet crucial observation: Datasets with different label noise levels have different entropies. Although the entropy of any given dataset $\mathcal{D}$ is (initially) unknown to us, we do know, crucially, that a *complete* uniformly random relabeling of $\mathcal{D}$ would yield a new dataset with *maximum entropy* (which we call "baseline datasets"), and we can easily generate multiple such datasets. We could also use *partial* relabelings to generate a spectrum of new datasets whose entropies range from the entropy of $\mathcal{D}$, to the maximum possible entropy. We call them "$\alpha$-increment datasets", where $\alpha$ is a parameter that we control. The minimum value $\alpha_{\mathrm{min}}$ for $\alpha$, such that an $\alpha$-increment dataset reaches maximum entropy, would depend on the original entropy of $\mathcal{D}$. See Fig. 1 for a visualization of the spectrum of entropies for $\alpha$-increment datasets and baseline datasets.

Our main idea is to train a discriminator $\Phi$ that recognizes datasets with maximum entropy, and then use $\Phi$ to determine this minimum value $\alpha_{\mathrm{min}}$. Once this value is estimated, we are then able to estimate $Q_{\mathcal{D}}$. Specific realizations of our framework correspond to specific designs for $\Phi$. An

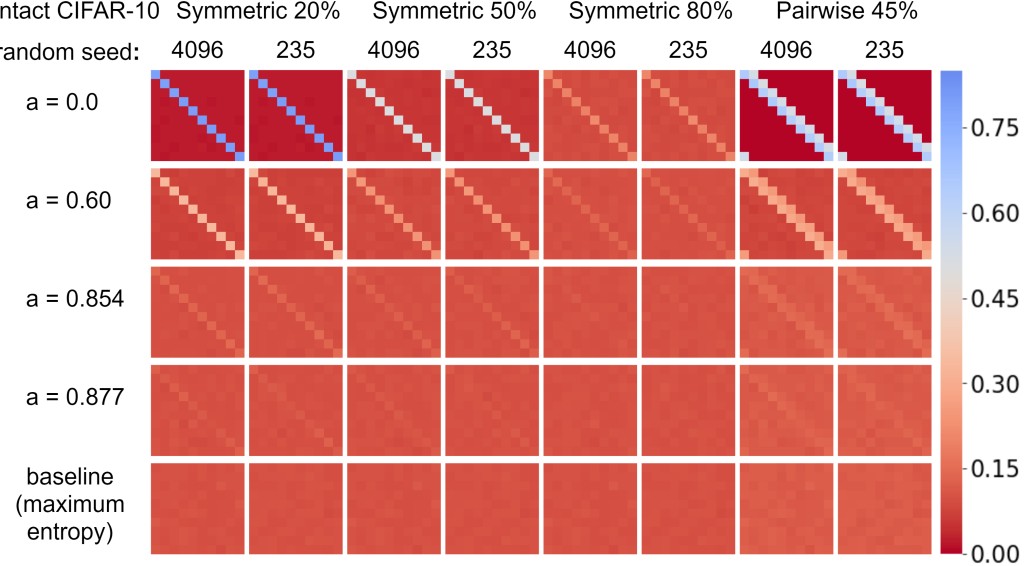

Figure 1: A visualization of the entropy maximization process. We illustrate the noise transition matrices of DILNs as heat maps. The intact CIFAR-10 dataset is used as the underlying clean dataset, and four noise models are considered (symmetric 20%/50%/80% noise and pairwise 45% noise), which are shown here as four pairs of columns. The first four rows depict heat maps for $\alpha$-increment DILNs, where $\alpha = (a, \dots, a)$ for four values $a = 0.0, 0.6, 0.854, 0.877$. The last row depicts heat maps for baseline DILNs, which have expected maximum entropy. Note that the minimum value $a_{\min}$ for $a$, such that a discriminator would find $\alpha$-increment DILNs "indistinguishable" from baseline DILNs, would depend on the base noise model that the $\alpha$-increment DILN is derived from. From this figure, we are able to infer, for example, that for symmetric noise models, $a_{\min} \approx 0.877$ for base noise level $50\%$, while in contrast, $a_{\min} \approx 0.854$ for base noise level $80\%$. In general, different noise levels correspond to different minimum values for $\alpha$.

illustration of our framework using LID-based discriminators is given in Fig. 2; details on LID-based discriminators can be found in Section 3, and will be further elaborated in the appendix.

Throughout this paper, given any discrete random variables $X, Y$, we shall write $p_X(x)$ and $p_{X|Y}(x|y)$ to mean $\Pr(X = x)$ and $\Pr(X = x|Y = y)$ respectively. We assume that the reader is familiar with the basics of information theory; see Cover & Thomas (2012) for an excellent introduction.

## 2.1 ENTROPY OF DATASETS WITH LABEL NOISE

Given $\mathcal{D}$ a dataset with instance-independent label noise (DILN), let $\mathcal{A}$ be its set of all label classes, and let $Y$ (resp. $Z$) be the given (resp. correct) label of a randomly selected instance $X$ of $\mathcal{D}$.[1] For convenience, we say that $\mathcal{D}$ is a DILN with *noise model* $(Y|Z; \mathcal{A})$. The *noise transition matrix* of $\mathcal{D}$ is a matrix $Q_{\mathcal{D}}$ whose $(i, j)$-th entry is $q_{i,j}^{\mathcal{D}} := p_{Y|Z}(j, i)$. We shall define the *entropy* of $\mathcal{D}$ by

$$H(\mathcal{D}) := -\sum_{i \in \mathcal{A}} p_Z(i) \sum_{j \in \mathcal{A}} q_{i,j}^{\mathcal{D}} \log q_{i,j}^{\mathcal{D}}.$$

Notice that $H(\mathcal{D})$ is precisely the conditional entropy of $Y$ given $Z$. (We use the convention that $0 \log 0 = 0$.) Hence, it is easy to prove that $0 \leq H(\mathcal{D}) \leq \log |\mathcal{A}|$. In particular, $H(\mathcal{D}) = 0$ if and only if every pair of instances of $\mathcal{D}$ in the same class have the same given labels. Note also that $\mathcal{D}$ has maximum entropy $\log |\mathcal{A}|$ if and only if every entry of $Q_{\mathcal{D}}$ equals $\frac{1}{|\mathcal{A}|}$ (i.e. the given labels of $\mathcal{D}$ are completely noisy). Thus, $H(\mathcal{D})$ could be interpreted as a measure of the label noise level of $\mathcal{D}$.

---

[1]Every datapoint of $\mathcal{D}$ is a pair $(\mathbf{x}, y)$, where $\mathbf{x}$ is an instance, and $y$ is its given label, which may differ from the correct label $z$ associated to $\mathbf{x}$. Note that $Z$ is a function of $X$, and $Y$ is a random function of $Z$. By *instance-independent* label noise, we mean that $\Pr(Y = y|Z = z, X = \mathbf{x}) = \Pr(Y = y|Z = z)$. A more detailed treatment of DILNs can be found in Appendix A. In particular, a DILN includes its noise model.

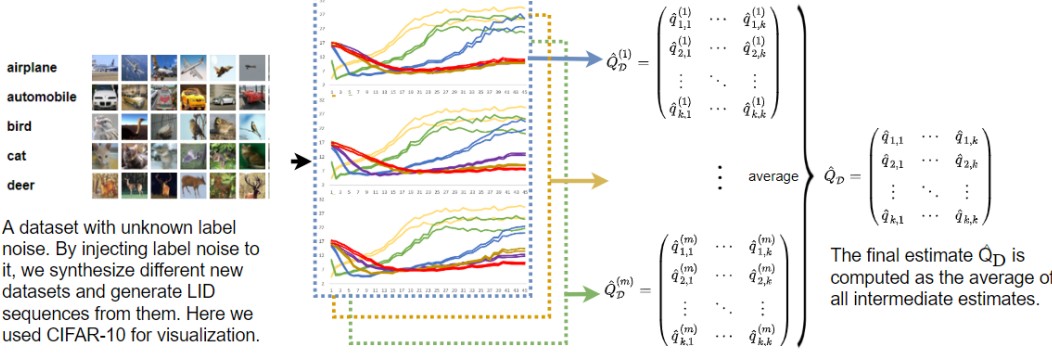

Each dotted box represents a discriminator trained on a collection of LID sequences, obtained via 3 random seeds. LID sequences generated from each random seed are depicted as a plot. The three plots shown represent collectively the training data for a single discriminator. Each well-trained discriminator produces a single intermediate estimate.

Figure 2: An overview of the proposed information-theoretic framework, using LID-based discriminators as a concrete example. Given an input dataset $\mathcal{D}$ with instance-independent label noise, several collections of new datasets are synthesized from $\mathcal{D}$. These synthesized datasets are "derived" via different partial relabelings (and hence have different entropies). The synthesized datasets in each collection are further processed to create training data for a single discriminator. In this illustration, LID sequences from the synthesized datasets become the training data for each discriminator. Every discriminator generates a single intermediate estimate for the noise transition matrix $Q_\mathcal{D}$. The final estimate is computed as the mean of all intermediate estimates.

A *derived* DILN of $\mathcal{D}$ shall mean a DILN $\mathcal{D}'$ with noise model $(Y'|Z; \mathcal{A})$ for some $Y'$ independent of $Z$, such that both $\mathcal{D}, \mathcal{D}'$ have the same underlying set of instances, given in the same sequential order. For example, $\mathcal{D}'$ could be "derived" from $\mathcal{D}$ by inserting additional instance-independent label noise, in which case $\mathcal{D}'$ can be interpreted as a partial relabeling of $\mathcal{D}$. For convenience, we say that $\mathcal{D}'$ is a $Y'$-*derived* DILN of $\mathcal{D}$.

## 2.2 Synthesis of new datasets from $\mathcal{D}$

Let $\mathcal{D}$ be a DILN with noise model $(Y|Z; \mathcal{A})$. Without loss of generality, assume $\mathcal{A} = \{1, \ldots, k\}$, assume that $\mathcal{D}$ has $N$ instances, and write $Q_\mathcal{D} = [q_{i,j}]_{1 \leq i,j \leq k}$. The correct labels for the instances are fixed and unknown to us, hence all entries of $Q_\mathcal{D}$ are fixed constants with unknown values. Our goal is to estimate $Q_\mathcal{D}$. As alluded to earlier, we shall be synthesizing two types of datasets from $\mathcal{D}$.

The first type is what we call a *baseline* dataset, described as follows: Let $\overline{\mathcal{D}}$ be a random dataset obtained from $\mathcal{D}$ by replacing the given label of each instance of $\mathcal{D}$ by a label chosen uniformly at random from $\mathcal{A}$. Hence $\overline{\mathcal{D}}$ is a random DILN with expected entropy $\log k$ (i.e. maximum entropy), which we shall denote by $\mathcal{D}_{\max} := \mathbb{E}[\overline{\mathcal{D}}]$. The noise transition matrix $Q_{\overline{\mathcal{D}}} = [Q'_{i,j}]_{1 \leq i,j \leq k}$ of $\overline{\mathcal{D}}$ is a random matrix whose entries $Q'_{i,j} = \frac{1}{k}(1 + E'_{i,j})$ are random variables, where each "error" $E'_{i,j}$ is a random variable with mean 0. Any observed $\overline{\mathcal{D}}$ is called a *baseline* DILN of $\mathcal{D}$.

The second type is what we call an $\alpha$-*increment* dataset, where $\alpha = (\alpha_1, \ldots, \alpha_k)$ is a vector whose entries satisfy $0 \leq \alpha_i \leq 1$ for all $i$. Let $\mathcal{D}_\alpha$ be obtained from $\mathcal{D}$ as follows: For each $1 \leq i \leq k$, select uniformly at random $\alpha_i \times 100\%$ of the instances with given label $i$, and reassign each selected given label to one of the remaining $k - 1$ classes, chosen uniformly at random. Hence $\mathcal{D}_\alpha$ is a random DILN, and its noise transition matrix $Q_{\mathcal{D}_\alpha} = [Q''_{i,j}]_{1 \leq i,j \leq k}$ is a random matrix whose entries

$$Q''_{i,j} = q_{i,j}(1 - \alpha_j) + \sum_{\substack{1 \leq t \leq k \\ t \neq j}} q_{i,t} \alpha_t \frac{1}{k-1}(1 + E''_{i,j}) \quad \text{(for all } 1 \leq i, j \leq k) \tag{1}$$

are random variables, where each "error" $E''_{i,j}$ is a random variable with mean 0. Any observed $\mathcal{D}_\alpha$ is called an $\alpha$-*increment* DILN of $\mathcal{D}$.

### 2.3 Underlying Intuition for distinguishing baseline DILNs

Suppose we have a discriminator $\Phi$ that is able to predict whether an input DILN is a baseline DILN of $\mathcal{D}$. We could try, as input to $\Phi$, an $\alpha$-increment DILN $\mathcal{D}'$ of $\mathcal{D}$ for different values of $\alpha$. For any given $\alpha$, we could try, as input to $\Phi$, multiple observed values of the random DILN $\mathcal{D}_\alpha$. Intuitively, the values of $\alpha$ for which "most" of the observed values for $\mathcal{D}_\alpha$ are predicted by $\Phi$ to be baseline DILNs, give non-trivial information about $Q_\mathcal{D}$. In this subsection, we explain the underlying intuition for how $\Phi$ can be trained, without requiring any knowledge of the correct labels $Z$.

Let $\mathcal{D}_0$ be a baseline DILN, and let $\mathcal{D}_1, \ldots, \mathcal{D}_\ell$ be $\alpha$-increment DILNs for $\alpha = \alpha^{(1)}, \ldots, \alpha^{(\ell)}$, respectively. Intuitively, if $\alpha^{(1)}, \ldots, \alpha^{(\ell)}$ cover a sufficiently wide range of vectors, then most of the DILNs among $\mathcal{D}_1, \ldots, \mathcal{D}_\ell$, would have entropies that are not near maximum entropy and hence would (in principle) be distinguishable from baseline DILNs. Ideally, we would want to train a discriminator $\Phi$ using baseline DILNs as "positive" data, and $\alpha$-increment DILNs as "unlabeled" data, via some positive-unlabeled learning algorithm. However, having entire datasets as training data (i.e. where each DILN is a single datapoint for training $\Phi$) may not necessarily be a good representation for the training data. Hence, we introduce the notion of a *separable* random function $g$, where effectively, we shall use $g$ to generate the training data for $\Phi$ (by applying $g$ on the DILN).

Our definition for the "separability" of $g$ relies on (a suitable analog of) the asymptotic equipartition property (AEP) from information theory, which is a key ingredient in proving Shannon's channel coding theorem. This AEP can be interpreted as a rigorous formulation of the idea that a "typical" sequence of observed values for $r$ i.i.d. random variables would belong to a tiny fraction of all possible sequences of observed values, when $r$ is sufficiently large. This notion of "typicality" can be explained with the example of flipping biased coins. Suppose we have two coins, with probabilities $0.1$ and $0.6$ respectively, for getting heads. Toss each coin a total of $r$ times, and record the corresponding sequence of outcomes. Now repeat the process multiple times to get multiple sequences, each of length $r$. If $r$ is sufficiently large, then with high probability, a randomly selected sequence for the first (resp. second) coin would have heads for $\approx 0.1$ (resp. $\approx 0.6$) of the outcomes; such sequences form only a vanishingly small fraction of all possible sequences. Hence, if we repeatedly generate these sequences, then with high probability, we would be able to distinguish our two coins.

We define $g$ to be an $\mathbb{R}^d$-valued random function. If $g(\mathcal{D}_0)$ is invoked $r$ times, then we get a randomly generated sequence of length $r$, where each entry is a vector in $\mathbb{R}^d$; this resulting output sequence shall be treated as a *single* datapoint in the "positive" class, for training $\Phi$. Analogously, for each $1 \le i \le \ell$, we shall invoke $g(\mathcal{D}_i)$ a total of $r$ times, and treat the resulting output sequence (of length $r$) to be a *single* "unlabeled" datapoint for training $\Phi$. We repeat this process to generate our training data for $\Phi$. If $r$ is sufficiently large, then with high probability, each output sequence (datapoint for $\Phi$) would be a typical sequence, whose statistics is based on the input DILN. Informally, we define $g$ to be "separable" if distinct DILNs have distinguishable typical sequences for sufficiently large $r$.

### 2.4 Discriminators for $\mathcal{D}$ and estimators for $Q_\mathcal{D}$

We now formalize our intuition presented in Section 2.3. Let $\mathfrak{D}[\mathcal{D}]$ be the set of all derived DILNs of $\mathcal{D}$, and let $\mathfrak{U}[\mathcal{D}]$ be the set of all possible underlying datasets for $\mathfrak{D}[\mathcal{D}]$ (i.e. we throw away information about the associated noise models). For notational ease, a DILN $\mathcal{D}'$ could be an element of either $\mathfrak{D}[\mathcal{D}]$ or $\mathfrak{U}[\mathcal{D}]$. Every $\mathcal{D}'$ in $\mathfrak{U}[\mathcal{D}]$ is uniquely determined by its sequence of given labels $(y_1, \ldots, y_N)$, which we call the *labeling* of $\mathcal{D}'$. We think of $(y_1, \ldots, y_N)$ as a *relabeling* of $\mathcal{D}$, generated from some (possibly unknown) noise model $(Y'|Z; \mathcal{A})$. Hence, $\mathfrak{U}[\mathcal{D}]$ is a finite set of size $k^N$. Formally, a *random derived DILN* of $\mathcal{D}$ is a discrete random variable $V : \mathfrak{D}[\mathcal{D}] \to \mathfrak{U}[\mathcal{D}]$, defined on some distribution on $\mathfrak{D}[\mathcal{D}]$. We shall also define a map $f_{\text{matrix}}$ with domain $\mathfrak{D}[\mathcal{D}]$, given by $\mathcal{D}' \mapsto Q_{\mathcal{D}'}$.

A *discriminator* for $\mathcal{D}$ is a prediction model $\Phi$ that takes any $\mathcal{D}'$ in $\mathfrak{U}[\mathcal{D}]$ as its input and gives a score $\Phi(\mathcal{D}')$ in $[0, 1]$, which could be interpreted as the likelihood that $\mathcal{D}'$ is a baseline DILN of $\mathcal{D}$. We say that $\mathcal{D}'$ is predicted "positive" if $\Phi(\mathcal{D}') \ge 0.5$, and predicted "negative" otherwise. Let $\Phi^+$ denote the subset of $\mathfrak{U}[\mathcal{D}]$ on which $\Phi$ predicts positive.

What is a "typical" value of $f_{\text{matrix}}(\overline{\mathcal{D}})$? Notice that all $k^N$ possible relabelings of $\mathcal{D}$ could occur as the labeling of a randomly generated baseline DILN of $\mathcal{D}$, so the set of possible outcomes for $f_{\text{matrix}}(\overline{\mathcal{D}})$ is the entire set of all possible noise transition matrices. Intuitively, we know for example that the

identity matrix is a "non-typical" value for $f_{\text{matrix}}(\overline{\mathcal{D}})$, even though its occurrence is possible. We shall adapt the notion of typical sets from information theory to capture this intuition of "typicality".

**Definition 2.1.** Given $V$ a random derived DILN of $\mathcal{D}$, let $V_1, V_2, \ldots$ be an infinite sequence of i.i.d. random derived DILNs of $\mathcal{D}$, with the same distribution as $V$.

(i) For any $\epsilon > 0$ and integer $n \geq 1$, the *n-fold $\varepsilon$-typical set of $V$* is defined to be the set $\Lambda_\varepsilon^{(n)}(V)$ consisting of all sequences $(\mathcal{D}_1, \ldots, \mathcal{D}_n) \in \mathfrak{U}[\mathcal{D}]^n$ of observed values of $V_1, \ldots, V_n$, with the property

$$\mathbb{E}[H(V)] - \varepsilon \leq \tfrac{1}{n} \sum_{1 \leq t \leq n} H(\mathcal{D}_t) \leq \mathbb{E}[H(V)] + \varepsilon. \tag{2}$$

(ii) Consider an arbitrary function $g : \mathfrak{U}[\mathcal{D}] \to \mathbb{R}^d$. Note that each $g(V_i)$ is an $\mathbb{R}^d$-valued random variable. For any $\epsilon > 0$ and integer $n \geq 1$, the *n-fold $\varepsilon$-typical set of $g(V)$* is defined to be the set $\Lambda_\varepsilon^{(n)}(g(V))$ consisting of all sequences $(u_1, \ldots, u_n) \in \mathbb{R}^d \times \cdots \times \mathbb{R}^d \cong \mathbb{R}^{dn}$ of observed vectors of $g(V_1), \ldots, g(V_n)$, with the property

$$\mathbb{E}[g(V)] - \varepsilon \mathbf{1}_d \leq \tfrac{1}{n} \sum_{1 \leq t \leq n} u_t \leq \mathbb{E}[g(V)] + \varepsilon \mathbf{1}_d.$$

**Remark 2.2.** Given a (non-random) derived DILN $\mathcal{D}'$ of $\mathcal{D}$, and an $\mathbb{R}^d$-valued *random* function $g$, we could treat $g(\mathcal{D}')$ equivalently as a composition of a random derived DILN of $\mathcal{D}$ with an $\mathbb{R}^d$-valued (non-random) function. Hence for any $\epsilon > 0$ and integer $n \geq 1$, in view of Definition 2.1(ii), the notion of an $n$-fold $\varepsilon$-typical set of $g(\mathcal{D}')$ is well-defined.

**Definition 2.3.** Let $\mathcal{D}_0 \in \mathfrak{U}[\mathcal{D}]$, and let $g$ be an $\mathbb{R}^d$-valued random function on $\mathfrak{U}[\mathcal{D}]$. We say that $g$ is $\mathcal{D}_0$-*separable* if for every $\varepsilon > 0$, $\delta > 0$, and every $\mathcal{D}_1 \in \mathfrak{U}[\mathcal{D}]$ satisfying $|H(\mathcal{D}_0) - H(\mathcal{D}_1)| > \delta$, there exists some sufficiently large $n$ such that $\Lambda_\varepsilon^{(n)}(g(\mathcal{D}_0))$ and $\Lambda_\varepsilon^{(n)}(g(\mathcal{D}_1))$ are disjoint typical sets. We say that $g$ is *separable* if $g$ is $\mathcal{D}_0$-separable for all $\mathcal{D}_0 \in \mathfrak{U}[\mathcal{D}]$.

**Definition 2.4.** Let $\beta > 0$, let $\mathcal{D}_0 \in \mathfrak{U}[\mathcal{D}]$, and let $g$ be a $\mathcal{D}_0$-separable $\mathbb{R}^d$-valued random function on $\mathfrak{U}[\mathcal{D}]$. We say that a discriminator $\Phi$ for $\mathcal{D}$ is *trained $n$-fold on $(\mathcal{D}_0, g)$ with threshold $\beta$*, if

$$\Phi^+ = \{\mathcal{D}' \in \mathfrak{U}[\mathcal{D}] : \Lambda_\beta^{(n)}(g(\mathcal{D}_0)) \cap \Lambda_\beta^{(n)}(g(\mathcal{D}')) \neq \emptyset\}$$

**Definition 2.5.** An $\alpha$-*sequence* for $\mathcal{D}$ is a (finite or infinite) sequence $(\alpha^{(1)}, \alpha^{(2)}, \ldots)$ of distinct vectors in $[0, \frac{k-1}{k})^k$ that satisfies $\alpha^{(i)} \leq \alpha^{(j)}$ (coordinate-wise inequality) for all $i < j$. An $\alpha$-sequence is called *valid* if it is a (possibly finite) subsequence of an infinite $\alpha$-sequence $(\alpha^{(1)}, \alpha^{(2)}, \ldots)$ whose set of elements $\{\alpha^{(i)}\}_{i=1}^\infty$ is a dense subset of $[0, \frac{k-1}{k}]^k$.

Our estimator for $Q_\mathcal{D}$ relies on the existence of a separable random function $g$ on $\mathfrak{U}[\mathcal{D}]$. Once we find such a $g$, we can then train multiple discriminators $\Phi$ using multiple randomly generated baseline DILNs of $\mathcal{D}$, to get multiple intermediate estimates for $Q_\mathcal{D}$. We use each discriminator $\Phi$ to find a suitable $\alpha \in [0, 1]^k$ such that $\Phi$ gives a high score for a "typical" $\alpha$-increment DILN of $\mathcal{D}$. We shall then use this value $\alpha$ to compute an intermediate estimate for $Q_\mathcal{D}$. The average of these intermediate estimates is our final estimate $\hat{Q}_\mathcal{D}$ for $Q_\mathcal{D}$; see Algorithm 1. More details are found in Appendix B.

**Theorem 2.6.** *Let $\hat{Q}_\mathcal{D}$ be the final averaged output matrix from Algorithm 1, which takes as its inputs integers $r, m, n, \ell \geq 1$, a threshold $\beta > 0$, a separable $\mathbb{R}^d$-valued random function on $\mathfrak{U}[\mathcal{D}]$, and a valid $\alpha$-sequence $\Omega = (\alpha^{(1)}, \ldots, \alpha^{(\ell)})$ for $\mathcal{D}$. Then $\hat{Q}_\mathcal{D}$ converges in probability to $Q_\mathcal{D}$ as $r \to \infty$, $m \to \infty$, $n \to \infty$, $\ell \to \infty$, and $N \to \infty$.*

Informally, the input integers $r, m, n, \ell$ can be interpreted as follows: $\ell$ is the length of the input $\alpha$-sequence; $m$ is the number of baseline DILNs generated; $r$ is the length of the sequences that each discriminator $\Phi$ is trained on; and $n$ is the number of observed values of $\mathcal{D}_\alpha$ (for some optimal $\alpha$ contained in the input $\alpha$-sequence) that are predicted positive by $\Phi$.

Theorem 2.6 tells us that our estimator (i.e. Algorithm 1) is consistent; see Corollary B.16 in the appendix for a more refined statement. Roughly speaking, our proof of Theorem 2.6 involves a careful iterated use of typical sequences, and requires an analog of joint AEP for DILNs, as well as a notion of "transverse entropy", which has no corresponding analog in the usual notion of entropy for random variables (see Appendix B.1). Although our consistency result requires the limit $N \to \infty$ (recall that $N$ is the number of instances in $\mathcal{D}$), a careful analysis of our proof reveals that for fixed $N$ (with $r \to \infty$, $m \to \infty$, $n \to \infty$, $\ell \to \infty$), we have an explicit upper bound on the estimation error of our estimator; see Corollary B.15. In the next section, we shall introduce a suitable candidate for $g$.

---

**Algorithm 1** An overview for the general framework for estimating $Q_{\mathcal{D}}$

---

    **Require:** integers $r, m, n, \ell \geq 1$.
    **Require:** threshold $\beta > 0$, $g$: a separable $\mathbb{R}^d$-valued random function on $\mathfrak{U}[\mathcal{D}]$
    **Require:** $\Omega = (\alpha^{(1)}, \ldots, \alpha^{(\ell)}) \subseteq [0, 1]^k$ a valid $\alpha$-sequence for $\mathcal{D}$.

1: Initialize empty list $\mathcal{L}$.
2: **for** $\varsigma = 1 \ldots m$ **do**
3:     Generate observed value $\overline{\mathcal{D}} = \mathcal{D}_0^{(\varsigma)}$.
4:     **for** $s = 1 \ldots \ell$ **do**
5:         Generate $n$ independent observed values $\mathcal{D}_{\alpha^{(s)}} = \mathcal{D}_{s,1}^{(\varsigma)}, \mathcal{D}_{s,2}^{(\varsigma)}, \ldots, \mathcal{D}_{s,n}^{(\varsigma)}$.
6:     Let $\Phi_\varsigma$ be a discriminator trained $r$-fold on $(\mathcal{D}_0^{(\varsigma)}, g)$ with threshold $\beta$.
7:     Compute $s' := \min\{s : 1 \leq s \leq \ell,$ there exists $1 \leq t \leq n$ such that $\mathcal{D}_{s,t}^{(\varsigma)} \in \Phi_\varsigma^+\}$.
8:     **if** $s'$ exists (i.e. $s'$ is well-defined) **then**
9:         **for** $t = 1 \ldots n$ **do**
10:             Generate observed values for random variables $E'_{i,j}$, $E''_{i,j}$ (for all $1 \leq i, j \leq k$)
            #[Note: $E'_{i,j}$, $E''_{i,j}$ are defined in $Q_{\overline{\mathcal{D}}}$, $Q_{\mathcal{D}_{\alpha_{s'}}}$ respectively.]
11:             Solve system of linear equations $Q_{\overline{\mathcal{D}}} = Q_{\mathcal{D}_{\alpha_{s'}}}$ (in the $k^2$ variables $q_{i,j}$ for $1 \leq i, j \leq k$)
            #[Note: We substitute the generated observed values for $E'_{i,j}$, $E''_{i,j}$ into $Q_{\overline{\mathcal{D}}} = Q_{\mathcal{D}_{\alpha_{s'}}}$.]
            #[Note: Unique solution to linear system exists almost surely; more details in Appendix B.2.]
12:             **if** Unique solution to linear system exists **then**
13:                 $\hat{Q}_t^{(\varsigma)} \leftarrow [\hat{q}_{i,j}]_{1 \leq i,j \leq k}$, where $\{q_{i,j} = \hat{q}_{i,j}\}_{i,j}$ is the unique solution to the linear system.
14:                 Insert matrix $\hat{Q}_t^{(\varsigma)}$ into list $\mathcal{L}$
15: **return** mean of matrices in $\mathcal{L}$ (This is our estimate $\hat{Q}_{\mathcal{D}}$ for $Q_{\mathcal{D}}$.)

---

## 3   REALIZATION OF FRAMEWORK USING LID-BASED DISCRIMINATORS

A key challenge for realizing our framework is the construction of good discriminators. This requires a suitable separable random function $g$. The underlying intuition for $g$ we should have is that we want $g$ to "separate" datasets with different noise levels. Appendix B.6 elaborates on this intuition.

With this intuition in mind, we propose to use *Local Intrinsic Dimensionality (LID) scores* (Houle, 2013; 2017a;b). The LID score is used in several applications (Amsaleg et al., 2017; Von Brünken et al., 2015; Schubert & Gertz, 2017), and it plays a fundamental role in a 2018 breakthrough in noise detection: It is possible to determine whether a dataset is clean or has label noise, by considering *LID sequences* (Ma et al., 2018b), which are sequences of LID scores; cf. Amsaleg et al. (2015). In particular, it is possible for LID sequences to detect adversarial noise (Ma et al., 2018a).

LID scores are assigned to every training epoch. As observed in Ma et al. (2018b), the LID score of a model has an initial phase: It would start "high" and then generally decrease to a "low" value. Subsequently, its behavior depends on the amount of label noise in the dataset. In the absence of label noise, the LID score would remain low. If instead there is "significant" label noise, then the LID score would rise (after its initial decrease). Thus, the presence of label noise in a dataset could *in principle* be detected by any sharp increase in the LID score during the training phase. In this paper, we use LID sequences as a proxy for measuring the entropy of the underlying dataset trained on.

Consider a neural network $\mathcal{N}$. Suppose $\mathcal{D}' \in \mathfrak{U}[\mathcal{D}]$, and let $\mathbf{x}_1, \ldots, \mathbf{x}_N$ be the enumeration of all instances of $\mathcal{D}'$. As we train our neural network on $\mathcal{D}'$, we shall keep track of how the feature vectors of randomly selected instances evolve over the training epochs. Given an input instance $\mathbf{x}$, the *feature vector* of $\mathbf{x}$ shall mean the output vector of the last hidden layer of $\mathcal{N}$, given the input $\mathbf{x}$; we shall denote the feature vector of $\mathbf{x}$ in epoch $j$ by $\omega_j(\mathbf{x})$, and we shall define $\Omega_j := \{\omega_j(\mathbf{x}_i)\}_{1 \leq i \leq N}$.

Let $s, s' \geq 1$ be fixed integers. The *LID score* of a single instance $\mathbf{x}$ in epoch $j$ is defined by

$$\text{LID}_j(\mathbf{x}; \mathcal{D}') := -\left(\tfrac{1}{s} \sum_{1 \leq i \leq s} (\log r_i(\mathbf{x}) - \log r_s(\mathbf{x}))\right)^{-1},$$

where $r_i(\mathbf{x})$ is the Euclidean distance between $\omega_j(\mathbf{x})$ and its $i$-th nearest neighbor in $\Omega_j$. The *LID score* of $\mathcal{D}'$ in epoch $j$, denoted by $\text{LID}_j(\mathcal{D}')$, is the mean LID scores of $s'$ randomly selected instances

in epoch $j$. If training is done over $L$ epochs, then $\mathrm{LID}(\mathcal{D}') := (\mathrm{LID}_1(\mathcal{D}'), \dots, \mathrm{LID}_L(\mathcal{D}'))$ is the *LID sequence* of $\mathcal{D}'$. We then define the random function $g_{\mathrm{LID}} : \mathfrak{U}[\mathcal{D}] \to \mathbb{R}^L$ by $\mathcal{D}' \mapsto \mathrm{LID}(\mathcal{D}')$.

An LID-based discriminator is a discriminator $\Phi$ trained on LID sequences as its training data. For every synthesized baseline DILN $\mathcal{D}'$ of $\mathcal{D}$, we shall invoke $g_{\mathrm{LID}}(\mathcal{D}')$ a total of $r$ times, which yields $r$ LID sequences that shall be considered "positive". For each $\alpha \in [0,1]^k$ and each $\alpha$-increment DILN $\mathcal{D}''$ of $\mathcal{D}$, we similarly invoke $g_{\mathrm{LID}}(\mathcal{D}'')$ a total of $r$ times, which yields $r$ LID sequences that shall be considered "unlabeled". Hence, we can generate training data for $\Phi$ consisting of "positive" samples and "unlabeled" samples. We could then use any positive-unlabeled learning algorithm to train $\Phi$.

| CIFAR-10 | Sym-20% | | | Sym-50% | | | Sym-80% | | | Sym (averaged) | | |
|---|---|---|---|---|---|---|---|---|---|---|---|---|
| AP Removal | 0% | 40% | 70% | 0% | 40% | 70% | 0% | 40% | 70% | 0% | 40% | 70% |
| S-model | **0.0225** | **0.0427** | **0.1381** | 0.4886 | 0.4014 | 0.3177 | 1.5650 | 1.3957 | 1.2215 | 0.6920 | 0.6133 | 0.5591 |
| Forward | 0.0865 | 0.2116 | 0.4999 | 0.0873 | 0.2490 | **0.2735** | 0.1617 | 0.1720 | 0.5589 | 0.1118 | 0.2109 | 0.4441 |
| T-Revision | 0.0869 | 0.2896 | 0.6206 | 0.1459 | 0.2425 | 0.3159 | 0.2303 | 0.2765 | 0.2267 | 0.1544 | 0.2695 | 0.3877 |
| ours-1 | 0.0573 | 0.2171 | 0.2907 | **0.0268** | **0.1802** | 0.4927 | **0.1355** | **0.1022** | **0.1742** | **0.0732** | **0.1665** | **0.3192** |
| MPEIA | 0.8898 | 1.1730 | 1.4105 | 0.3322 | 0.5176 | 0.6085 | 0.0187 | 0.0494 | **0.0909** | 0.4136 | 0.5800 | 0.7033 |
| ours-2 | **0.7133** | **0.8999** | **1.0454** | **0.2752** | **0.3987** | **0.4585** | **0.0121** | **0.0365** | 0.1225 | **0.3335** | **0.4450** | **0.5421** |
| GLC | 0.1966 | 0.3922 | **1.0582** | 0.1444 | 0.2783 | **0.8438** | 0.0598 | 0.0907 | 0.2365 | 0.1336 | 0.2537 | **0.7128** |
| ours-3 | **0.1397** | **0.3282** | 1.1580 | **0.0992** | **0.2037** | 0.9991 | 0.0864 | 0.1012 | 0.2991 | **0.1084** | **0.2110** | 0.8187 |

Table 1: Forward KL loss comparisons for symmetric noise matrix estimations with CIFAR-10 as the underlying clean dataset. "AP" means anchor point. We removed anchor-like data up to 70% in the same manner described in (Xia et al., 2019). An average loss reduction is achieved from 0.27 to 0.17 for 40% anchor-like instances removal when comparing with baselines not using clean samples. We also improved MPEIA and GLC by using their estimates as priors. Smaller loss values are bold-faced.

| CIFAR-10 | Pairwise-20% | | | Pairwise-45% | | | Pairwise-80% | | | Pairwise (averaged) | | |
|---|---|---|---|---|---|---|---|---|---|---|---|---|
| 90% removal | no | hardest | easiest | no | hardest | easiest | no | hardest | easiest | no | hardest | easiest |
| S-model | 0.5156 | 0.5568 | 0.5019 | 1.2443 | 1.2016 | 1.2104 | **2.6906** | 2.5212 | 2.6777 | 1.4835 | 1.4265 | 1.4633 |
| Forward | **0.0901** | 0.0982 | **0.1128** | 1.5657 | 1.0621 | 1.8904 | 8.2493 | 8.7039 | 9.1303 | 3.3017 | 3.2881 | 3.7112 |
| T-Revision | 0.0723 | **0.0870** | 0.1356 | 0.9283 | **0.6337** | **0.7656** | 5.7647 | 5.7378 | 5.2833 | 2.2551 | 2.1528 | 2.0615 |
| ours-1 | 0.3644 | 0.5334 | 0.4093 | **0.8332** | 0.7364 | 0.7770 | 2.6957 | **1.8657** | **2.0011** | 1.2978 | **1.0452** | **1.0625** |
| MPEIA | 0.5854 | 0.5741 | 0.6139 | **0.5477** | 0.5814 | 0.5868 | **0.6006** | 0.6123 | 0.6507 | 0.5779 | 0.5893 | 0.6171 |
| ours-2 | **0.3881** | 0.5499 | 0.4344 | 0.5528 | 0.5505 | 0.5751 | 0.6771 | **0.4924** | **0.4738** | **0.5393** | **0.5309** | **0.4944** |
| GLC | 0.2637 | **0.2545** | 0.2591 | 0.2752 | 0.2896 | 0.2666 | 0.2928 | 0.2588 | 0.2653 | 0.2772 | **0.2676** | 0.2637 |
| ours-3 | **0.1967** | 0.3754 | **0.1867** | 0.3544 | 0.3294 | 0.3037 | **0.2015** | 0.2751 | 0.2169 | **0.2509** | 0.3266 | **0.2358** |

Table 2: Forward KL loss comparisons for pairwise noise transition matrix estimations with CIFAR-10 as the underlying clean dataset. In the table, "no" means no sample removal, "hardest" and "easiest" mean a 90% random sample removal from the hardest class, cat, and the easiest class, frog. We constantly perform the best for averaged noise levels. Smaller loss values are bold-faced.

## 4 EXPERIMENTS

**Framework implementation details.** Let $\alpha^{(1)}, \dots, \alpha^{(\ell)}$ be a sequence of vectors in $[0,1]^k$. Let $\mathcal{D}_0$ be a baseline DILN of $\mathcal{D}$, and for each $1 \le s \le \ell$, let $\mathcal{D}_s$ be an $\alpha^{(s)}$-increment DILN of $\mathcal{D}$. If $\mathcal{D}_0, \mathcal{D}_1, \dots, \mathcal{D}_\ell$ are synthesized using a common random seed $\varsigma$, then we say that $\{\mathcal{D}_0, \mathcal{D}_1, \dots, \mathcal{D}_\ell\}$ is a *seed collection* with seed $\varsigma$ and $\alpha$-*sequence* $(\alpha^{(1)}, \dots, \alpha^{(\ell)})$. In our experiments, we used a fixed $\alpha$-sequence $\Omega$, where each $\alpha = (\alpha_1, \dots, \alpha_k)$ in $\Omega$ satisfies $\alpha_i \le 0.886$ for all $i$ for CIFAR-10 and $\alpha_i \le 0.916$ for all $i$ for Clothing1M. We trained each discriminator $\Phi$ on the LID sequences obtained from three different seed collections (called a "triple"), where two of them are used for training and the third is used for validation. Our LID-based discriminator $\Phi$ is trained using positive-unlabeled bagging (Elkan & Noto, 2008; Mordelet & Vert, 2014), with decision trees as our sub-routine. We used 1000 decision trees. For each derived DILN, we generated 50 LID sequences to be used as training data. Once trained, our discriminator $\Phi$ assigns a score to each input DILN $\mathcal{D}'$ based on voting: Again, 50 LID sequences are generated for $\mathcal{D}'$. Each LID sequence is predicted either positive or negative by $\Phi$, and the total number of positive votes, divided by 50, is the final score assigned to $\mathcal{D}'$. After training, if the validation recall is $\tau \ge 0.9$, then the discriminator $\Phi$ would be further fine-tuned. Details on fine-tuning can be found in Appendix C.2.2.

**Datasets.** We did experiments on CIFAR-10 (Krizhevsky et al., 2009) and Clothing1M (Xiao et al., 2015). CIFAR-10 has $50,000$ training and $10,000$ test images over 10 classes. We manually added

two types of instance-independent label noise: symmetric and pairwise, following the label flip settings in (Han et al., 2018). For symmetric noise, we used noise levels $20\%, 50\%$ and $80\%$, while for pairwise asymmetric noise, we used noise levels $20\%, 45\%$ and $80\%$. Clothing1M (Xiao et al., 2015) has around 1 million clothing images of 14 classes. The paper (Xiao et al., 2015) also provides a noisy subset, whose corresponding $Q_\mathcal{D}$ has been manually verified exactly. We estimate its $Q_\mathcal{D}$ based on this subset for real-life label noise scenario and refer this subset as "Clothing1M subset".

**Methods.** We compared our method with baselines: (i) **S-model** (Goldberger & Ben-Reuven, 2016), which concatenates a neural network (NN) with an extra softmax layer; (ii) **Forward** (Patrini et al., 2017), which trains an NN and uses anchor-like instances to estimate $Q_\mathcal{D}$; (iii) **T-Revision** (Xia et al., 2019), which finetunes $Q_\mathcal{D}$ concurrently with the training of its classifier; (iv) **MPEIA** (Yu et al., 2018), which estimates mixture proportion by a fraction of $\mathcal{D}_\text{clean}$, for $Q_\mathcal{D}$; and (v) **Gold Loss Correction (GLC)** (Hendrycks et al., 2018), which trains an NN on the noisy data. Then the trained NN computes softmax outputs of $\mathcal{D}_\text{clean}$ for $Q_\mathcal{D}$. S-model, Forward and T-Revision do not require $\mathcal{D}_\text{clean}$ while MPEIA and GLC randomly selects $0.5\%$ $\mathcal{D}_\text{clean}$ from the whole dataset (in our paper). NN structure, training losses and training epochs can be found in Table **??** and Table 6 (in the appendix) for CIFAR-10 and Clothing1M, respectively. We used the same training hyper-parameters and data augmentation settings as given in Patrini et al. (2017), except T-Revision, which follows the settings in Xia et al. (2019) for respective datasets. **Ours-1** took a similar approach as GLC without $\mathcal{D}_\text{clean}$ to obtain prior (we call this prior "avg prior"). We first split the dataset into a $90\%$ training set and a $10\%$ validation set randomly. We then train an NN to compute probability vectors of the whole noisy dataset. The probability vectors of the samples with label $i$ from the epoch with the best validation accuracy is averaged as the $i$th row of prior. **Ours-2** (resp. **ours-3**) used MPEIA's (resp. GLC's) estimates as priors[2]. All values reported are averaged over at least 5 estimates.

**Experimental Results.** We row-normalized all estimates from the baselines then evaluated them using (forward) Kullback–Leibler (KL) loss[3] For CIFAR-10's both symmetric and pairwise cases, even for $70\%$ anchor-point removal or imbalanced class ratios, our method has the lowest losses for averaged noise levels, compared to all the baselines, and made improvements when MPEIA and GLC are used as our priors. For Clothing1M, ours-1 has the lowest KL loss, $0.4903$, slightly better than T-Revision with a KL loss of $0.5262$. S-model ranked the last. Among all methods that used $0.5\%$ clean samples, ours-3 achieved the lowest loss, $0.5311$, while its prior GLC has a loss of $0.5957$.

| Methods | S-model | Forward | T-Revision | ours-1 | MPEIA | ours-2 | GLC | ours-3 |
|---|---|---|---|---|---|---|---|---|
| Forward KL loss | 2.1189 | 1.1098 | 0.5262 | **0.4903** | **1.7408** | 1.8344 | 0.5957 | **0.5311** |

Table 3: Forward KL loss comparisons for Clothing1M subset. Ours-1 has the lowest forward KL loss among all baseline models. When $0.5\%$ $\mathcal{D}_\text{clean}$ is used, we (ours-3) improved GLC.

## 5 CONCLUDING REMARKS

This paper focuses on datasets with instance-independent label noise (DILNs), and tackles the problem of estimating the noise transition matrix $Q_\mathcal{D}$ of a DILN $\mathcal{D}$. Our main algorithm is the first-ever estimator for $Q_\mathcal{D}$ that is proven to be consistent without needing to optimize the classification accuracy of a classifier trained on $\mathcal{D}$. Notably, we do not require clean data or anchor-like instances, and we do not make any assumptions on the structure of $Q_\mathcal{D}$. Thus, a key "takeaway insight" is that $Q_\mathcal{D}$ could be accurately estimated in a wide range of scenarios, including possibly for classification tasks that are "inherently still difficult" to get high classification accuracies even without label noise.

Our consistent estimator is based on a new information-theoretic framework, in which we introduce the notion of entropy for DILNs. A key step in our approach is the training of discriminators to predict whether an input DILN has maximum entropy. Our proof of consistency relies crucially on the notion of "typicality" and the asymptotic equipartition property from information theory.

---

[2]Both MPEIA and GLC inherently require $\mathcal{D}_\text{clean}$. Since our method does not leverage clean data, it would not be fair to directly evaluate our method against them. Instead, ours-2 and ours-3 are intended to show that MPEIA and GLC can be enhanced with our method, without having to do further clean data annotation/augmentation.

[3]If $\hat{Q}_\mathcal{D} = [\hat{q}_{i,j}]_{1 \leq i,j \leq k}$ (resp. $Q_\mathcal{D} = [q_{i,j}]_{1 \leq i,j \leq k}$) is the estimated (resp. true) noise transition matrix for $\mathcal{D}$, then the corresponding (forward) KL loss is defined to be $\sum_{i=1}^k p_Z(i) \sum_{j=1}^k q_{i,j} \log\left(q_{i,j}/\hat{q}_{i,j}\right)$.

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

APPENDIX

This appendix is organized as follows:

- Section A gives a detailed treatment of datasets with instance-independent label noise (DILNs).
- Section B proves the consistency of our proposed estimator.
- Section C provides all the implementation details of our experiments.
- Section D describes how our work relates to the information bottleneck theory for deep learning.

## A   A RIGOROUS FORMALISM FOR DILNs

A dataset $\mathcal{D}$ is a set consisting of $N$ ("instance", "given-label") pairs. If we enumerate these pairs by $(\mathbf{x}_1, y_1), (\mathbf{x}_2, y_2), \ldots, (\mathbf{x}_N, y_N)$, then $\mathcal{D}$ is the set of pairs $\{(\mathbf{x}_i, y_i)\}_{1 \leq i \leq N}$. Across all disciplines that deal with datasets, there is an implicit assumption that the instances of a dataset are sampled from some "true" data distribution. Formally, each instance $\mathbf{x}$ is an observed value of some random variable $X_{\text{true}}$. For classification tasks, given an instance $X_{\text{true}} = \mathbf{x}$, it is assumed that there is a uniquely determined correct label $y_{\mathbf{x}}$ associated to this instance $\mathbf{x}$. Hence, there is a function $f_{\text{true}}$ that assigns each instance $\mathbf{x}$ to its correct label $y_{\mathbf{x}}$. We assume that $f_{\text{true}}$ is completely deterministic and does not involve any randomness. If $Z_{\text{true}}$ is a random variable representing the correct label of a random instance $X_{\text{true}}$, then $Z_{\text{true}} = f_{\text{true}}(X_{\text{true}})$. Because we are only given the datapoints of $\mathcal{D}$, we typically consider the restriction of the domain of $f_{\text{true}}$ to the instances of $\mathcal{D}$; we denote this restricted function by $f_{\text{true}}^{\mathcal{D}}$. In the absence of label noise, the dataset becomes precisely $\mathcal{D} = \{(\mathbf{x}_i, y_{\mathbf{x}_i})\}_{1 \leq i \leq N}$. For such "noise-free" datasets $\mathcal{D}$, the goal of learning a classifier from $\mathcal{D}$ is to obtain a good approximation $\hat{f}_{\text{true}}^{\mathcal{D}}$ for the function $f_{\text{true}}^{\mathcal{D}}$.[4]

When instance-independent label noise is added to the dataset, there is a random function $f_{\text{noise}}$ that is applied to the correct labels $y_{\mathbf{x}}$ of all instances $\mathbf{x}$. This random function $f_{\text{noise}}$ depends only on the input label $z$, and does not depend on which instance $\mathbf{x}$ this label is for. Similar to the case of $f_{\text{true}}$, because we are only given the datapoints of $\mathcal{D}$, we again typically consider the restriction of the domain of $f_{\text{noise}}$ to the instances of $\mathcal{D}$; we denote this restricted function by $f_{\text{noise}}^{\mathcal{D}}$. Assuming that the set of all possible labels is $\mathcal{A} = \{1, \ldots, k\}$, this implies that $f_{\text{noise}}^{\mathcal{D}}$ can be decomposed into $k$ random variables $Y_1, \ldots, Y_k$, where each $Y_i$ is a discrete random variable taking on values in $\mathcal{A}$. The set of probabilities $\{p_{Y_i}(1), \ldots, p_{Y_i}(k)\}$ would completely determine the distribution of $Y_i$. Thus, $f_{\text{noise}}^{\mathcal{D}}$ is a function on $\mathcal{A}$ given by the map $i \mapsto Y_i$. This is a random function that is completely determined by the set of all $k^2$ probabilities $\{p_{Y_i}(j)\}_{1 \leq i,j \leq k}$. The $k$-by-$k$ matrix $Q = [q_{i,j}]_{1 \leq i,j \leq k}$ whose $(i,j)$-th entry equals $p_{Y_i}(j)$ is precisely the *noise transition matrix* that we consider in our paper, which we have denoted by $Q_{\mathcal{D}}$. Consequently, to "learn" a model of this instance-independent label noise is to obtain a good approximation $\hat{f}_{\text{noise}}$ for $f_{\text{noise}}^{\mathcal{D}}$, which is exactly the same as finding a good approximation $\hat{Q}_{\mathcal{D}}$ to the noise transition matrix $Q_{\mathcal{D}}$.

Our goal for this paper is to estimate the noise transition matrix $Q_{\mathcal{D}}$ from a given dataset $\mathcal{D}$. Thus, one of our main assertions for this paper, that "high classification accuracy is not required for estimating $Q_{\mathcal{D}}$ well", can be interpreted as the assertion that we can find a good approximation $\hat{f}_{\text{noise}}$ for $f_{\text{noise}}^{\mathcal{D}}$, even if we are unable to find a good approximation for $f_{\text{true}}^{\mathcal{D}}$, or a good approximation for $f_{\text{noise}}^{\mathcal{D}} \circ f_{\text{true}}^{\mathcal{D}}$. Notice that for existing methods that "learn in the presence of label noise", their underlying goal is to find a good approximation for either $f_{\text{true}}^{\mathcal{D}}$ or the composite map $f_{\text{noise}}^{\mathcal{D}} \circ f_{\text{true}}^{\mathcal{D}}$. In contrast, our goal is to find a good approximation for $f_{\text{noise}}^{\mathcal{D}}$.

Given $\mathcal{D}$ a dataset with instance-independent label noise (DILN), let $X$ be a random variable representing an instance of $\mathcal{D}$ selected uniformly at random (notice that $X \neq X_{\text{true}}$), let $Z := f_{\text{true}}^{\mathcal{D}}(X)$, and let $Y := f_{\text{noise}}^{\mathcal{D}}(Z)$. By definition, $f_{\text{noise}}^{\mathcal{D}}$ is completely determined by the conditional distribution of $Y$ given $Z$, and the set of all possible labels $\mathcal{A}$; in particular, $f_{\text{noise}}^{\mathcal{D}}$ does not depend on $X$ or $f_{\text{true}}^{\mathcal{D}}$. This explains why the *noise model* associated to $\mathcal{D}$ is denoted by $(Y|Z; \mathcal{A})$. In our paper, we have

---

[4]Of course, the purpose for learning a classifier from $\mathcal{D}$ is to learn a good representation for $f_{\text{true}}$, and we can only do so given the dataset $\mathcal{D}$. If the underlying distribution of $\mathcal{D}$ is not a "good representation" of the distribution of $X_{\text{true}}$, then any approximation to $f_{\text{true}}^{\mathcal{D}}$, no matter how accurate, will not approximate $f_{\text{true}}$ well. Henceforth, we assume that $\mathcal{D}$ has a "good underlying representation" for the distribution of $X_{\text{true}}$.

defined a DILN to be a set. It is the set $\mathcal{D} = \{(\mathbf{x}_i, y_i)\}_{1 \leq i \leq N}$, which also has an associated *noise model* $(Y|Z; \mathcal{A})$.

Strictly speaking, a DILN should be defined as a triple $(\mathcal{D}, Y|Z, \mathcal{A})$, since we also include the information about the noise model $(Y|Z; \mathcal{A})$ as part of the definition of the DILN. However, for the purpose of this paper, we abuse notation (slightly) and assume that a DILN $\mathcal{D}$ includes its associated noise model. To avoid ambiguity, the set consisting of all ("instance", "given-label") pairs $(\mathbf{x}_i, y_i)$ shall be called the *underlying dataset* of $\mathcal{D}$, and each such $(\mathbf{x}_i, y_i)$ pair shall be called a datapoint of $\mathcal{D}$. Of course, we are implicitly assuming that the instances $\mathbf{x}_1, \ldots, \mathbf{x}_N$ are sampled from some "true" distribution (i.e. the distribution of $X_{\text{true}}$), but we do not need any information involving $X_{\text{true}}$ beyond this implicit assumption.

# B    PROOF OF CONSISTENCY OF PROPOSED ESTIMATOR FOR $Q_{\mathcal{D}}$

The goal for this section is to prove Theorem 2.6, i.e. that our proposed estimator for $Q_{\mathcal{D}}$ is consistent; see Theorem B.14 for a precise (equivalent) formulation of Theorem 2.6. Our consistency proof is essentially an iterated application of suitable analogs of the asymptotic equipartition property (AEP) theorem from information theory. In particular, we will prove a joint AEP theorem for DILNs; see Theorem B.3. The proof of Theorem B.14 requires some preparation, so we shall first prove several related results in Sections B.1–B.3, before we present Theorem B.14 in Section B.4.

Throughout, let $\mathcal{D}$ be a DILN with noise model $(Y|Z; \mathcal{A})$. Without loss of generality, assume that $\mathcal{A} = \{1, \ldots, k\}$ satisfies $k \geq 2$, assume that $\mathcal{D}$ has $N$ instances, and write $Q_{\mathcal{D}} = [q_{i,j}]_{1 \leq i, j \leq k}$. If $\{X_n\}_{n=1}^{\infty}$ is any sequence of random variables, then we write "$X_n \xrightarrow{p} \mu$" to mean that "$X_n$ converges in probability to $\mu$". Here, $\mu$ could be a scalar or a random variable.

## B.1    JOINT ASYMPTOTIC EQUIPARTITION PROPERTY FOR DILNs

Recall that in Definition 2.1, we introduced the notion of "$n$-fold $\varepsilon$-typical sets" for random derived DILNs. We will also need to define jointly typical sets for random derived DILNs. This involves what we shall call "transverse entropy", which has no corresponding analog in the usual notion of entropy for random variables.

**Definition B.1.** Let $\mathcal{D}'$ and $\mathcal{D}''$ be derived DILNs of $\mathcal{D}$. The *transverse entropy* of $\mathcal{D}'$ and $\mathcal{D}''$ is

$$H(\mathcal{D}' \wedge \mathcal{D}'') := -\sum_{i \in \mathcal{A}} p_Z(i) \sum_{j \in \mathcal{A}} |q_{i,j}^{\mathcal{D}'} - q_{i,j}^{\mathcal{D}''}| \log |q_{i,j}^{\mathcal{D}'} - q_{i,j}^{\mathcal{D}''}|.$$

(Recall: We use the convention that $0 \log 0 = 0$.)

**Definition B.2.** Let $V$ and $W$ be random derived DILNs of $\mathcal{D}$. Let $V_1, V_2, \ldots$ (resp. $W_1, W_2, \ldots$) be an infinite sequence of i.i.d. random derived DILNs of $\mathcal{D}$ with the same distribution as $V$ (resp. $W$). For any $\epsilon > 0$ and integer $n \geq 1$, the *$n$-fold jointly $\varepsilon$-typical set of $V$ and $W$* is defined to be the set $\Lambda_{\varepsilon}^{(n)}(V, W)$ consisting of all sequences $\left((\mathcal{D}'_1, \ldots, \mathcal{D}'_n), (\mathcal{D}''_1, \ldots, \mathcal{D}''_n)\right) \in \mathfrak{U}[\mathcal{D}]^n \times \mathfrak{U}[\mathcal{D}]^n$ of observed values of $V_1, \ldots, V_n, W_1, \ldots, W_n$, with the following properties:

$$\mathbb{E}[H(V)] - \varepsilon \leq \frac{1}{n} \sum_{t=1}^{n} H(\mathcal{D}'_t) \leq \mathbb{E}[H(V)] + \varepsilon; \tag{3}$$

$$\mathbb{E}[H(W)] - \varepsilon \leq \frac{1}{n} \sum_{t=1}^{n} H(\mathcal{D}''_t) \leq \mathbb{E}[H(W)] + \varepsilon; \tag{4}$$

$$\mathbb{E}[H(V \wedge W)] - \varepsilon \leq \frac{1}{n} \sum_{t=1}^{n} H(\mathcal{D}'_t \wedge \mathcal{D}''_t) \leq \mathbb{E}[H(V \wedge W)] + \varepsilon. \tag{5}$$

In particular, notice that the restriction of $\Lambda_{\varepsilon}^{(n)}(V, W) \subseteq \mathfrak{U}[\mathcal{D}]^n \times \mathfrak{U}[\mathcal{D}]^n$ to the first (resp. second) $\mathfrak{U}[\mathcal{D}]^n$ component is a subset of the $n$-fold $\varepsilon$-typical set of $V$ (resp. $W$).

**Theorem B.3** (cf. Cover & Thomas (2012, Thm. 7.6.1)). *Let $V$ and $W$ be random derived DILNs of $\mathcal{D}$. Let $V_1, V_2, \ldots$ (resp. $W_1, W_2, \ldots$) be an infinite sequence of i.i.d. random derived DILNs of*

$\mathcal{D}$ with the same distribution as $V$ (resp. $W$), and suppose that $\mathcal{D}'_1, \mathcal{D}'_2, \ldots$ (resp. $\mathcal{D}''_1, \mathcal{D}''_2, \ldots$) is a corresponding sequence of observed values. Then for any $\varepsilon > 0$,

$$\lim_{n \to \infty} \Pr\left(\left((\mathcal{D}'_1, \ldots, \mathcal{D}'_n), (\mathcal{D}''_1, \ldots, \mathcal{D}''_n)\right) \in \Lambda_\varepsilon^{(n)}(V, W)\right) = 1.$$

*Proof.* Consider an arbitrary $\left((\mathcal{D}'_1, \ldots, \mathcal{D}'_n), (\mathcal{D}''_1, \ldots, \mathcal{D}''_n)\right) \in \Lambda_\varepsilon^{(n)}(V, W)$. By the weak law of large numbers, $\frac{1}{n} \sum_{t=1}^n H(\mathcal{D}'_t)$ converges in probability to $\mathbb{E}[H(V)]$. Hence, given any $\varepsilon > 0$, there exists an integer $n_1 \geq 1$ such that for all integers $n > n_1$,

$$\Pr\left(\left|\frac{1}{n}\left(\sum_{t=1}^n H(\mathcal{D}'_t)\right) - \mathbb{E}[H(V)]\right| \geq \varepsilon\right) \leq \frac{\varepsilon}{3}. \tag{6}$$

By a similar argument, we also infer that given any $\varepsilon > 0$, there exists an integer $n_2 \geq 1$ such that for all integers $n > n_2$,

$$\Pr\left(\left|\frac{1}{n}\left(\sum_{t=1}^n H(\mathcal{D}''_t)\right) - \mathbb{E}[H(W)]\right| \geq \varepsilon\right) \leq \frac{\varepsilon}{3}, \tag{7}$$

and there exists an integer $n_3 \geq 1$ such that for all integers $n > n_3$,

$$\Pr\left(\left|\frac{1}{n}\left(\sum_{t=1}^n H(\mathcal{D}'_t \wedge \mathcal{D}''_t)\right) - \mathbb{E}[H(V \wedge W)]\right| \geq \varepsilon\right) \leq \frac{\varepsilon}{3}. \tag{8}$$

Therefore, for all integers $n \geq \max\{n_1, n_2, n_3\}$, the probability of the union of the events in (6), (7) and (8) must be at most $\varepsilon$, which proves our assertion. $\qquad \square$

For the rest of this subsection, let $\mathcal{D}_0 \in \mathfrak{U}[\mathcal{D}]$, and let $g$ be a $\mathcal{D}_0$-separable $\mathbb{R}^d$-valued random function on $\mathfrak{U}[\mathcal{D}]$. Recall that for $\beta > 0$, a discriminator $\Phi$ for $\mathcal{D}$ is said to be *trained $r$-fold on* $(\mathcal{D}_0, g)$ *with threshold* $\beta$, if the set of positive predictions for $\Phi$ is

$$\Phi^+ = \{\mathcal{D}' \in \mathfrak{U}[\mathcal{D}] : \Lambda_\beta^{(r)}(g(\mathcal{D}_0)) \cap \Lambda_\beta^{(r)}(g(\mathcal{D}')) \neq \emptyset\}.$$

**Definition B.4.** Let $\delta' > 0$, $\beta > 0$, let $r \geq 1$ be an integer, and suppose $\Phi$ is a discriminator for $\mathcal{D}$ that is trained $r$-fold on $(\mathcal{D}_0, g)$ with threshold $\beta$. We say that $\Phi$ is $\delta'$-*sufficient* if every $\mathcal{D}' \in \mathfrak{U}[\mathcal{D}]$ satisfying $H(\mathcal{D}_0 \wedge \mathcal{D}') < \delta'$ is predicted positive, i.e. $\mathcal{D}' \in \Phi^+$.

**Lemma B.5.** *For every $\beta, \delta > 0$, there is a sufficiently large integer $r_{\beta,\delta} \geq 1$ such that for all integers $n \geq r_{\beta,\delta}$, if $\Phi$ is a discriminator for $\mathcal{D}$ that is trained $n$-fold on $(\mathcal{D}_0, g)$ with threshold $\beta$, then the following implication holds:*

$$\mathcal{D}' \in \Phi^+ \implies |H(\mathcal{D}_0) - H(\mathcal{D}')| \leq \delta.$$

*Proof.* Consider any $\mathcal{D}_1 \in \mathfrak{U}[\mathcal{D}]$ that satisfies $|H(\mathcal{D}_0) - H(\mathcal{D}_1)| > \delta$. Since $g$ is $\mathcal{D}_0$-separable, it follows from Definition 2.3 that there is an integer $r_{\beta,\delta}^{\mathcal{D}_1}$ such that $\Lambda_\beta^{(n)}(g(\mathcal{D}_0)) \cap \Lambda_\beta^{(n)}(g(\mathcal{D}_1)) = \emptyset$ for all integers $n \geq r_{\beta,\delta}^{\mathcal{D}_1}$. For each $\beta$ and $\delta$, define

$$r_{\beta,\delta} := \max\{r_{\beta,\delta}^{\mathcal{D}_1} \in \mathbb{Z} : \mathcal{D}_1 \in \mathfrak{U}[\mathcal{D}], |H(\mathcal{D}_0) - H(\mathcal{D}_1)| > \delta\}.$$

In particular, $r_{\beta,\delta}$ is well-defined, since $\mathfrak{U}[\mathcal{D}]$ is finite (of size $k^N$).

Now, for any $n \geq r_{\beta,\delta}$, suppose that $\Phi$ is trained $n$-fold on $(\mathcal{D}_0, g)$ with threshold $\beta$. By Definition 2.4, this means that $\Phi^+ = \{\mathcal{D}' \in \mathfrak{U}[\mathcal{D}] : \Lambda_\beta^{(n)}(g(\mathcal{D}_0)) \cap \Lambda_\beta^{(n)}(g(\mathcal{D}')) \neq \emptyset\}$. By the definition of $r_{\beta,\delta}$, we thus have $\Lambda_\beta^{(n)}(g(\mathcal{D}_0)) \cap \Lambda_\beta^{(n)}(g(\mathcal{D}')) = \emptyset$ for all $\mathcal{D}'$ satisfying $|H(\mathcal{D}_0) - H(\mathcal{D}')| > \delta$, which then proves the assertion. $\qquad \square$

**Lemma B.6.** *Let $0 < \varepsilon \leq 1$, and let $h : [0, k-1] \to \mathbb{R}$ be a function given by*

$$h(x) = \frac{k-1}{k}\left(1 - \frac{x}{k-1}\right)\log\left(1 - \frac{x}{k-1}\right) + \frac{1}{k}(1+x)\log(1+x). \tag{9}$$

*Then $h(x)$ is strictly increasing, and $h(\varepsilon) > 0$. Moreover, if $\zeta := \min\{p_Z(i) : 1 \leq i \leq k\} > 0$, and if $H(\mathcal{D}) \geq \log k - \zeta h(\varepsilon)$, then $|q_{i,j}^{\mathcal{D}} - \frac{1}{k}| \leq \varepsilon$ for all $1 \leq i, j \leq k$.*

*Proof.* First of all, we check that the derivative of $h(x)$ is

$$h'(x) = -\frac{1}{k}\left[\log\left(1 - \frac{x}{k-1}\right) - \log(1+x)\right] = -\frac{1}{k}\left[\log\left(\frac{1 - \frac{x}{k-1}}{1+x}\right)\right],$$

which satisfies $h'(x) > 0$ for all $0 < x < k - 1$. (Recall that $k \geq 2$ by assumption.) Thus, $h(x)$ is strictly increasing on the closed interval $[0, k-1]$. In particular, $\varepsilon > 0$ implies that $h(\varepsilon) > h(0) = 0$.

Henceforth, assume $\zeta > 0$, and suppose on the contrary that there exists some $1 \leq i_0, j_0 \leq k$ such that $\left|q_{i_0,j_0}^{\mathcal{D}} - \frac{1}{k}\right| > \varepsilon$. Let $q_{i_0,j}^{\mathcal{D}} = \frac{1}{k}(1 + \varepsilon_j)$ for all $1 \leq j \leq k$, and assume without loss of generality that $q_{i_0,j_0}^{\mathcal{D}} = \frac{1}{k}(1 + \varepsilon_{j_0})$ for some $\frac{1}{k}\varepsilon_{j_0} > \varepsilon$. This implies that

$$-\sum_{\substack{1 \leq j \leq k \\ j \neq j_0}} q_{i_0,j}^{\mathcal{D}} \log q_{i_0,j}^{\mathcal{D}} = -\sum_{\substack{1 \leq j \leq k \\ j \neq j_0}} \tfrac{1}{k}(1 + \varepsilon_j) \log\left[\tfrac{1}{k}(1 + \varepsilon_j)\right]$$

$$= \sum_{\substack{1 \leq j \leq k \\ j \neq j_0}} \tfrac{1}{k}(1 + \varepsilon_j) \log k - \sum_{\substack{1 \leq j \leq k \\ j \neq j_0}} \tfrac{1}{k}(1 + \varepsilon_j) \log(1 + \varepsilon_j).$$

Since $\sum_{j=1}^{k} \varepsilon_j = 0$, and since the map $x \mapsto -x \log x$ is concave, it follows from Jensen's inequality (see Cover & Thomas (2012, Thm. 2.6.2)) that

$$-\sum_{\substack{1 \leq j \leq k \\ j \neq j_0}} q_{i_0,j}^{\mathcal{D}} \log q_{i_0,j}^{\mathcal{D}} \leq \left[\sum_{\substack{1 \leq j \leq k \\ j \neq j_0}} \tfrac{1}{k}(1 + \varepsilon_j) \log k\right] - \tfrac{k-1}{k}(1 - \tfrac{\varepsilon_{j_0}}{k-1}) \log\left(1 - \tfrac{\varepsilon_{j_0}}{k-1}\right). \tag{10}$$

Note also that

$$-q_{i_0,j_0}^{\mathcal{D}} \log q_{i_0,j_0}^{\mathcal{D}} = -\tfrac{1}{k}(1 + \varepsilon_{j_0}) \log\left[\tfrac{1}{k}(1 + \varepsilon_{j_0})\right]$$

$$= \tfrac{1}{k}(1 + \varepsilon_{j_0}) \log k - \tfrac{1}{k}(1 + \varepsilon_{j_0}) \log(1 + \varepsilon_{j_0}). \tag{11}$$

Summing (10) and (11) gives us

$$-\sum_{1 \leq j \leq k} q_{i_0,j}^{\mathcal{D}} \log q_{i_0,j}^{\mathcal{D}} \leq \tfrac{1}{k}\left[\sum_{1 \leq j \leq k}(1 + \varepsilon_j) \log k\right] - \tfrac{k-1}{k}(1 - \tfrac{\varepsilon_{j_0}}{k-1}) \log\left(1 - \tfrac{\varepsilon_{j_0}}{k-1}\right) - \tfrac{1}{k}(1 + \varepsilon_{j_0}) \log(1 + \varepsilon_{j_0})$$

$$= \log k - \tfrac{k-1}{k}(1 - \tfrac{\varepsilon_{j_0}}{k-1}) \log\left(1 - \tfrac{\varepsilon_{j_0}}{k-1}\right) - \tfrac{1}{k}(1 + \varepsilon_{j_0}) \log(1 + \varepsilon_{j_0})$$

$$= \log k - h(\varepsilon_{j_0}). \tag{12}$$

Note that $q_{i_0,j_0}^{\mathcal{D}} = \frac{1}{k} + \frac{1}{k}\varepsilon_{j_0} \leq 1$ implies $\varepsilon_{j_0} \leq k - 1$, and recall that $\frac{1}{k}\varepsilon_{j_0} > \varepsilon$ by assumption, hence $h(\varepsilon_{j_0}) > h(\varepsilon) > 0$. It then follows from (12) that

$$-\sum_{1 \leq j \leq k} q_{i_0,j}^{\mathcal{D}} \log q_{i_0,j}^{\mathcal{D}} \leq \log k - h(\varepsilon_{j_0}) < \log k - h(\varepsilon) < \log k$$

Note also that for all $1 \leq i \leq k$ satisfying $i \neq i_0$, Jensen's inequality yields

$$-\sum_{1 \leq j \leq k} q_{i,j}^{\mathcal{D}} \log q_{i,j}^{\mathcal{D}} \leq \log k.$$

Thus,

$$H(\mathcal{D}) = -\sum_{i=1}^{k} p_Z(i) \sum_{j=1}^{k} q_{i,j}^{\mathcal{D}} \log q_{i,j}^{\mathcal{D}} < \log k - \zeta h(\varepsilon) \tag{13}$$

Since (13) contradicts the condition that $H(\mathcal{D}) \geq \log k - \zeta h(\varepsilon)$, we conclude that no such $i_0, j_0$ exist, therefore $H(\mathcal{D}) \geq \log k - \zeta h(\varepsilon)$ implies $|q_{i,j}^{\mathcal{D}} - \frac{1}{k}| \leq \varepsilon$ for all $1 \leq i, j \leq k$. $\square$

## B.2 Joint asymptotic equipartition property for DILN matrices

Let $\mathrm{Mat}_{k \times k}([0,1])$ be the set of all row-stochastic $k$-by-$k$ matrices. For convenience, a *random matrix* shall henceforth mean a $\mathrm{Mat}_{k \times k}([0,1])$-valued random variable, i.e. we omit the qualifier "row-stochastic" from "random row-stochastic matrix". Let $\left[\frac{1}{k}\right]_{k \times k}$ denote the matrix in $\mathrm{Mat}_{k \times k}([0,1])$ whose $k^2$ entries are all equal to $\frac{1}{k}$. We shall also use $\|\cdot\|$ to denote a norm on $\mathrm{Mat}_{k \times k}([0,1])$. All subsequent results still hold for *any* norm on $\mathrm{Mat}_{k \times k}([0,1])$; only certain constants in continuity arguments would change with a different norm. For concreteness, we shall work with the matrix 1-norm, i.e. $\|[c_{i,j}]_{1 \leq i,j \leq k}\| := \max_{1 \leq j \leq k} \sum_{i=1}^{k} |c_{i,j}|$.

**Proposition B.7.** *Let $0 < \varepsilon \leq 1$, and suppose that $\mathcal{D}^{(1)}, \ldots, \mathcal{D}^{(m)}$ are derived DILNs of $\mathcal{D}$. If $\zeta := \min\{p_Z(i) : 1 \leq i \leq k\} > 0$, and if $\frac{1}{m} \sum_{t=1}^{m} H(\mathcal{D}^{(t)}) \geq \log k - \zeta h(\frac{\varepsilon}{k})$ (where $h(x)$ is defined as in (9)), then*

$$\left\| \left( \frac{1}{m} \sum_{t=1}^{m} Q_{\mathcal{D}^{(t)}} \right) - \left[ \tfrac{1}{k} \right]_{k \times k} \right\| \leq \varepsilon. \tag{14}$$

*Proof.* Define the function $\sigma_Z : \mathrm{Mat}_{k \times k}([0,1]) \to \mathbb{R}$ by

$$[c_{i,j}]_{1 \leq i,j \leq k} \mapsto - \sum_{i=1}^{k} p_Z(i) \sum_{j=1}^{k} c_{i,j} \log c_{i,j}.$$

Note that $H(\mathcal{D}') = \sigma_Z(Q_{\mathcal{D}'})$ for any derived DILN $\mathcal{D}'$ of $\mathcal{D}$. Note also that $\sigma_Z$ is a concave function, so by generalized Jensen's inequality (see Perlman (1974)),

$$\log k - \zeta h(\tfrac{\varepsilon}{k}) \leq \frac{1}{m} \sum_{t=1}^{m} H(\mathcal{D}^{(t)}) = \frac{1}{m} \sum_{t=1}^{m} \sigma_Z(Q_{\mathcal{D}^{(t)}}) \leq \sigma_Z\left( \frac{1}{m} \sum_{t=1}^{m} Q_{\mathcal{D}^{(t)}} \right).$$

Consequently, writing $Q_{\mathcal{D}^{(t)}} = \left[ q_{i,j}^{(t)} \right]_{1 \leq i,j \leq k}$ for each $1 \leq t \leq m$, it follows from Lemma B.6 that

$$\left| \left( \frac{1}{m} \sum_{t=1}^{m} q_{i,j}^{(t)} \right) - \frac{1}{k} \right| \leq \frac{\varepsilon}{k}$$

for all $1 \leq i,j \leq k$, therefore (14) follows from the definition of the matrix 1-norm. $\square$

Next, we shall define several $\mathrm{Mat}_{k \times k}([0,1])$-valued functions. For every $\alpha = (\alpha_1, \ldots, \alpha_k) \in [0,1]^k$, define the function $f_{\mathrm{increment}}^{\alpha} : \mathrm{Mat}_{k \times k}([0,1]) \times \mathrm{Mat}_{k \times k}([0,1]) \to \mathrm{Mat}_{k \times k}([0,1])$ by

$$\left( [q_{i,j}]_{1 \leq i,j \leq k}, [\varepsilon_{i,j}]_{1 \leq i,j \leq k} \right) \mapsto \left[ q_{i,j}(1 - \alpha_j) + \sum_{\substack{1 \leq t \leq k \\ t \neq j}} q_{i,t} \alpha_t \tfrac{1}{k-1}(1 + \varepsilon_{i,j}) \right]_{1 \leq i,j \leq k} \tag{15}$$

Define the random matrix $E_{\alpha}'' := [E_{i,j}'']_{1 \leq i,j \leq k}$, where $E_{i,j}''$ is the random variable as defined in (1). Notice that by definition, $f_{\mathrm{increment}}^{\alpha}(Q_{\mathcal{D}}, E_{\alpha}'') = Q_{\mathcal{D}_{\alpha}}$.

Next, let $f_{\mathrm{solve}}^{\alpha} : \mathrm{Mat}_{k \times k}([0,1]) \times \mathrm{Mat}_{k \times k}([0,1]) \to \mathrm{Mat}_{k \times k}([0,1])$ be the function that is uniquely determined by the map $\left( f_{\mathrm{increment}}^{\alpha}(Q, \mathcal{E}), \mathcal{E} \right) \mapsto Q$. Note that to compute this map $f_{\mathrm{solve}}^{\alpha}$, we would need to solve a system of linear equations: Specifically, if $Q = [q_{i,j}]_{1 \leq i,j \leq k}$ and $\mathcal{E} = [\varepsilon_{i,j}]_{1 \leq i,j \leq k}$, and if $f_{\mathrm{increment}}^{\alpha}(Q, \mathcal{E})$ is the given matrix $[c_{i,j}]_{1 \leq i,j \leq k}$, then $Q$ can be computed by solving the system of $k^2$ linear equations in the $k^2$ variables $\{q_{i,j}\}_{1 \leq i,j \leq k}$, given as follows:

$$c_{i,j} = q_{i,j}(1 - \alpha_j) + \sum_{\substack{1 \leq t \leq k \\ t \neq j}} q_{i,t} \alpha_t \tfrac{1}{k-1}(1 + \varepsilon_{i,j}) \quad (\text{for } 1 \leq i,j \leq k).$$

In general, for any given matrix $[c_{i,j}]_{1 \leq i,j \leq k}$, if we sample $\mathcal{E}$ from the distribution of $E_{\alpha}''$, then this system of linear equations has a unique solution almost surely. Consequently, $f_{\mathrm{solve}}^{\alpha}([c_{i,j}]_{1 \leq i,j \leq k}, E_{\alpha}'')$ is well-defined almost surely.

**Lemma B.8.** *Let $\varepsilon > 0$, let $\alpha \in [0,1]^k$, and let $n \geq 1$ be an integer.*

- *Let $E_1'', E_2'', \ldots$ be an infinite sequence of i.i.d. random matrices with the same distribution as $E_\alpha''$, and suppose that $E_1'' = \mathcal{E}_1, E_2'' = \mathcal{E}_2, \ldots$ is a corresponding sequence of observed values.*
- *Let $\mathcal{D}_\alpha^{(1)}, \mathcal{D}_\alpha^{(2)}, \ldots$ be an infinite sequence of i.i.d. random derived DILNs of $\mathcal{D}$ with the same distribution as $\mathcal{D}_\alpha$, and suppose that $\mathcal{D}_\alpha^{(1)} = \mathcal{D}_1, \mathcal{D}_\alpha^{(2)} = \mathcal{D}_2, \ldots$ is a corresponding sequence of observed values.*

*For every integer $i \geq 1$, define $Q_{\mathcal{E}_i} := f_{\text{increment}}^\alpha(Q_\mathcal{D}, \mathcal{E}_i)$. Then,*

$$\lim_{n \to \infty} \Pr \left( \left\| \frac{1}{n} \sum_{i=1}^n \left( Q_{\mathcal{D}_i} - Q_{\mathcal{E}_i} \right) \right\| < \varepsilon \right) = 1.$$

*Proof.* By the weak law of large numbers, and using the definitions of $Q_{\mathcal{D}_i}$ and $Q_{\mathcal{E}_i}$, we have $\frac{1}{n} \sum_{i=1}^n Q_{\mathcal{D}_i} \xrightarrow{p} \mathbb{E}[Q_{\mathcal{D}_\alpha}]$, and $\frac{1}{n} \sum_{i=1}^n Q_{\mathcal{E}_i} \xrightarrow{p} \mathbb{E}[Q_{\mathcal{D}_\alpha}]$, hence the assertion follows. $\square$

**Theorem B.9.** *Let $n, m \geq 1$ be integers, and let $(\alpha^{(1)}, \ldots, \alpha^{(m)})$ be a sequence of $m$ vectors in $[0, 1]^k$. Assume that $\zeta := \min_{1 \leq i \leq k} p_Z(i) > 0$. Let $\varepsilon > 0$, and define $\delta := \frac{1}{2} \zeta h(\frac{\varepsilon}{2k}) > 0$ (where $h(x)$ is defined in (9)).*

- *Let $\mathcal{D}_0^{(1)}, \ldots, \mathcal{D}_0^{(m)} \in \mathfrak{U}[\mathcal{D}]$ such that*

$$\left| \log k - \frac{1}{m} \sum_{j=1}^m H(\mathcal{D}_0^{(j)}) \right| \leq \delta. \tag{16}$$

- *For every $1 \leq j \leq m$, let $E_1''^{(j)}, E_2''^{(j)}, \ldots$ be an infinite sequence of i.i.d. random matrices with the same distribution as $E_{\alpha^{(j)}}''$, and suppose that $E_1''^{(j)} = \mathcal{E}_1^{(j)}, E_2''^{(j)} = \mathcal{E}_2^{(j)}, \ldots$ is a corresponding sequence of observed values.*
- *Let $(\mathcal{D}_1^{(1)}, \ldots, \mathcal{D}_n^{(1)}) \in \Lambda_\varepsilon^{(n)}(\mathcal{D}_{\alpha^{(1)}}), \ldots, (\mathcal{D}_1^{(m)}, \ldots, \mathcal{D}_n^{(m)}) \in \Lambda_\varepsilon^{(n)}(\mathcal{D}_{\alpha^{(m)}})$ be $m$ sequences such that for all $1 \leq i \leq n$,*

$$\left| \frac{1}{m} \sum_{j=1}^m \left( H(\mathcal{D}_0^{(j)}) - H(\mathcal{D}_i^{(j)}) \right) \right| \leq \delta. \tag{17}$$

*For every $1 \leq i \leq n$, $1 \leq j \leq m$, define $\hat{Q}_{\mathcal{D}_i^{(j)}} := f_{\text{solve}}^{\alpha^{(j)}}(Q_{\mathcal{D}_0^{(j)}}, \mathcal{E}_i^{(j)})$. Then,*

$$\lim_{n \to \infty} \Pr \left( \left\| \left( \frac{1}{m} \sum_{j=1}^m \frac{1}{n} \sum_{i=1}^n \hat{Q}_{\mathcal{D}_i^{(j)}} \right) - Q_\mathcal{D} \right\| < \varepsilon \right) = 1. \tag{18}$$

*Proof.* First of all, note that (16) yields

$$\left| \log k - \frac{1}{m} \sum_{j=1}^m H(\mathcal{D}_0^{(j)}) \right| \leq \delta \leq 2\delta = \zeta h(\tfrac{\varepsilon}{2k}),$$

thus it follows from Proposition B.7 that $\left\| \frac{1}{m} \sum_{j=1}^m Q_{\mathcal{D}_0^{(j)}} - \left[ \frac{1}{k} \right]_{k \times k} \right\| \leq \frac{\varepsilon}{2}$. By (16) and (17), we infer that $\left| \log k - \frac{1}{m} \sum_{j=1}^m H(\mathcal{D}_i^{(j)}) \right| \leq 2\delta$ for all $1 \leq i \leq n$. So by similarly applying Proposition B.7, we get $\left\| \frac{1}{m} \sum_{j=1}^m Q_{\mathcal{D}_i^{(j)}} - \left[ \frac{1}{k} \right]_{k \times k} \right\| \leq \frac{\varepsilon}{2}$ for all $1 \leq i \leq n$.

Thus, by triangle inequality, $\left\| \frac{1}{m} \sum_{j=1}^m \left( Q_{\mathcal{D}_0^{(j)}} - Q_{\mathcal{D}_i^{(j)}} \right) \right\| \leq \varepsilon$ for all $1 \leq i \leq n$, which implies

$$\left\| \frac{1}{m} \sum_{j=1}^m \frac{1}{n} \sum_{i=1}^n \left( Q_{\mathcal{D}_0^{(j)}} - Q_{\mathcal{D}_i^{(j)}} \right) \right\| \leq \varepsilon. \tag{19}$$

Also, by Lemma B.8, we infer that

$$\lim_{n \to \infty} \Pr \left( \left\| \frac{1}{m} \sum_{j=1}^m \frac{1}{n} \sum_{i=1}^n \left( Q_{\mathcal{D}_i^{(j)}} - Q_{\mathcal{E}_i^{(j)}} \right) \right\| < \varepsilon \right) = 1, \tag{20}$$

where $Q_{\mathcal{E}_i^{(j)}} := f_{\text{increment}}^{\alpha^{(j)}}(Q_{\mathcal{D}}, \mathcal{E}_i^{(j)})$. Consequently, it follows from (19) and (20) that

$$\lim_{n \to \infty} \Pr\left(\left\|\frac{1}{m}\sum_{j=1}^m \frac{1}{n}\sum_{i=1}^n \left(Q_{\mathcal{D}_0^{(j)}} - Q_{\mathcal{E}_i^{(j)}}\right)\right\| < 2\varepsilon\right) = 1. \tag{21}$$

Note that by definition, $f_{\text{solve}}^{\alpha^{(j)}}(Q_{\mathcal{E}_i^{(j)}}, \mathcal{E}_i^{(j)}) = f_{\text{solve}}^{\alpha^{(j)}}(f_{\text{increment}}^{\alpha^{(j)}}(Q_{\mathcal{D}}, \mathcal{E}_i^{(j)}), \mathcal{E}_i^{(j)}) = Q_{\mathcal{D}}$. Therefore, by applying the multilinear function $f_{\text{solve}}^{\alpha^{(j)}}(-, \mathcal{E}_i^{(j)})$ to each term in (21), we get (18) as desired. $\quad\square$

## B.3 ESTIMATION OF $Q_{\mathcal{D}}$ VIA TYPICAL SETS

Define the random matrix $E' := [E'_{i,j}]_{1 \le i,j \le k}$, where $E'_{i,j}$ is the random variable as defined in Section 2.2. Given any observed value $\mathcal{E} = [\varepsilon_{i,j}]_{1 \le i,j \le k}$ for $E'$, we shall write $\mathcal{E} + \frac{1}{k}$ to denote the matrix $[\varepsilon_{i,j} + \frac{1}{k}]_{1 \le i,j \le k}$.

**Lemma B.10.** *Let $\varepsilon > 0$, and let $n \ge 1$ be an integer.*

- *Let $E'_1, E'_2, \dots$ be an infinite sequence of i.i.d. random matrices with the same distribution as $E'$, and suppose that $E'_1 = \mathcal{E}'_1, E'_2 = \mathcal{E}'_2, \dots$ is a corresponding sequence of observed values.*
- *Let $\overline{\mathcal{D}}_1, \overline{\mathcal{D}}_2, \dots$ be an infinite sequence of i.i.d. random derived DILN of $\mathcal{D}$ with the same distribution as $\overline{\mathcal{D}}$, and suppose that $\overline{\mathcal{D}}_1 = \mathcal{D}_1, \overline{\mathcal{D}}_2 = \mathcal{D}_2, \dots$ is a corresponding sequence of observed values.*

*Then,*

$$\lim_{n \to \infty} \Pr\left(\left\|\frac{1}{n}\sum_{i=1}^n \left(Q_{\mathcal{D}_i} - (\mathcal{E}'_i + \tfrac{1}{k})\right)\right\| < \varepsilon\right) = 1.$$

*Proof.* By the weak law of large numbers, and using the definitions of $Q_{\mathcal{D}_i}$ and $\mathcal{E}'_i + \frac{1}{k}$, we have $\frac{1}{n}\sum_{i=1}^n Q_{\mathcal{D}_i} \xrightarrow{p} \mathbb{E}[Q_{\overline{\mathcal{D}}}]$, and $\frac{1}{n}\sum_{i=1}^n(\mathcal{E}'_i + \frac{1}{k}) \xrightarrow{p} \mathbb{E}[Q_{\overline{\mathcal{D}}}]$, hence the assertion follows. $\quad\square$

By assumption, our dataset $\mathcal{D}$ has $N$ instances and $k$ label classes. We shall define the *gap* of $\mathcal{D}$ to be $\text{gap}(\mathcal{D}) := \log k - \mathbb{E}[H(\overline{\mathcal{D}})]$. Notice that by the definition $\overline{\mathcal{D}}$, this $\text{gap}(\mathcal{D})$ depends only on the values of $N$ and $k$.

**Lemma B.11.** $0 \le \text{gap}(\mathcal{D}) \le \log k$, and $\lim_{N \to \infty} \text{gap}(\mathcal{D}) = 0$.

*Proof.* By Jensen's inequality (see Cover & Thomas (2012, Thm. 2.6.2)), we have $\mathbb{E}[H(\overline{\mathcal{D}})] \le H(\mathbb{E}[\overline{\mathcal{D}}]) = \log k$, hence $\text{gap}(\mathcal{D}) \ge 0$. Note that $\text{gap}(\mathcal{D}) \le \log k$, since $H(\mathcal{D}') \ge 0$ for all DILNs $\mathcal{D}'$. Finally, the limit $\lim_{N \to \infty} \text{gap}(\mathcal{D}) = 0$ is a direct consequence of the weak law of large numbers. $\quad\square$

The following theorem is an extension of Theorem B.9 that takes into account predictions from a discriminator for $\mathcal{D}$. This extension involves the gap of $\mathcal{D}$.

**Theorem B.12.** *Let $n, m, r \ge 1$ be integers, let $(\alpha^{(1)}, \dots, \alpha^{(m)})$ be a sequence of $m$ vectors in $[0,1]^k$, and assume that $\zeta := \min\{p_Z(i) : 1 \le i \le k\} > 0$. Let $\beta, \delta, \varepsilon, \varepsilon' > 0$ be scalars satisfying $\text{gap}(\mathcal{D}) < \delta \le \frac{1}{2}\zeta \log k$, $0 < \varepsilon \le \delta - \text{gap}(\mathcal{D})$, and $\varepsilon' := 2k \cdot h^{-1}(\frac{2\delta}{\zeta}) > 0$, where $h(x)$ is defined in (9). Also, let $g$ be a separable $\mathbb{R}^d$-valued random function on $\mathfrak{U}[\mathcal{D}]$.*

- *Let $E'_1, \dots, E'_n$ be a sequence of i.i.d. random matrices with the same distribution as $E'$, and suppose that $E'_1 = \mathcal{E}'_1, \dots, E'_n = \mathcal{E}'_n$ is a corresponding sequence of observed values.*
- *For every $1 \le j \le m$, let $E''^{(j)}_1, E''^{(j)}_2, \dots, E''^{(j)}_n$ be a sequence of i.i.d. random matrices with the same distribution as $E''^{(j)}_{\alpha^{(j)}}$, and suppose $E''^{(j)}_1 = \mathcal{E}''^{(j)}_1, E''^{(j)}_2 = \mathcal{E}''^{(j)}_2, \dots, E''^{(j)}_n = \mathcal{E}''^{(j)}_n$ is a corresponding sequence of observed values.*
- *Let $(\mathcal{D}_0^{(1)}, \dots, \mathcal{D}_0^{(m)}) \in \Lambda_\varepsilon^{(m)}(\overline{\mathcal{D}})$, and for every $1 \le j \le m$, suppose that $\Phi_j$ is a discriminator for $\mathcal{D}$ that is trained $r$-fold on $(\mathcal{D}_0^{(j)}, g)$ with threshold $\beta$.*
- *For every $1 \le j \le m$, suppose that $(\mathcal{D}_1^{(j)}, \dots, \mathcal{D}_n^{(j)}) \in \Lambda_{\varepsilon'}^{(n)}(\mathcal{D}_{\alpha^{(j)}}) \cap (\Phi_j^+)^n$.*

*For every integer $1 \leq i \leq n$, define $\hat{Q}_i^{(j)} := f_{\text{solve}}^{\alpha^{(j)}}(\mathcal{E}_i' + \frac{1}{k}, \mathcal{E}_i''^{(j)})$. Then there exists a sufficiently large integer $r_{\beta,\delta}$ such that for all $r \geq r_{\beta,\delta}$,*

$$\lim_{n \to \infty} \Pr\left(\left\|\left(\frac{1}{m}\sum_{j=1}^{m}\left(\frac{1}{n}\sum_{i=1}^{n}\hat{Q}_i^{(j)}\right)\right) - Q_{\mathcal{D}}\right\| < 2\varepsilon'\right) = 1. \tag{22}$$

*Proof.* The proof, although seemingly complicated, actually follows essentially from unraveling the relevant definitions. First of all, notice that

$$\mathbb{E}[H(\overline{D})] - \varepsilon \geq \mathbb{E}[H(\overline{D})] - \delta + \text{gap}(\mathcal{D}) = \log k - \delta,$$

which implies that for any $(\mathcal{D}_0^{(1)}, \ldots, \mathcal{D}_0^{(m)}) \in \Lambda_\varepsilon^{(m)}(\overline{\mathcal{D}})$, we have (by the definition of $\Lambda_\varepsilon^{(m)}(\overline{\mathcal{D}})$) that

$$\left|\log k - \frac{1}{m}\sum_{j=1}^{m}H(\mathcal{D}_0^{(j)})\right| \leq \delta. \tag{23}$$

For each $1 \leq j \leq m$, it follows from Lemma B.5 that there exists some sufficiently large integer $r_{\beta,\delta}^{(j)} \geq 1$ such that for all $r \geq r_{\beta,\delta}^{(j)}$,

$$\mathcal{D}' \in \Phi_j^+ \implies |H(\mathcal{D}_0^{(j)}) - H(\mathcal{D}')| \leq \delta. \tag{24}$$

Let $r_{\beta,\delta} := \max\{r_{\beta,\delta}^{(j)} : 1 \leq j \leq m\}$, and henceforth assume that every discriminator $\Phi_j$ is trained $r$-fold on $(\mathcal{D}_0^{(j)}, g)$ with threshold $\beta$, for some $r \geq r_{\beta,\delta}$. For all $1 \leq j \leq m$ and $1 \leq i \leq n$, note that $\mathcal{D}_i^{(j)} \in \Phi_j^+$ by definition, hence (24) implies that $|H(\mathcal{D}_0^{(j)}) - H(\mathcal{D}_i^{(j)})| \leq \delta$, so in particular,

$$\left|\frac{1}{m}\sum_{j=1}^{m}\left(H(\mathcal{D}_0^{(j)}) - H(\mathcal{D}_i^{(j)})\right)\right| \leq \delta. \tag{25}$$

By definition, $\varepsilon' = 2k \cdot h^{-1}\left(\frac{2\delta}{\zeta}\right)$. This means that $\delta = \frac{1}{2}\zeta h\left(\frac{\varepsilon'}{2k}\right)$. In particular, recall from Lemma B.6 that $h(x)$ is a strictly increasing (and hence bijective) function with domain $[0, k-1]$, note that $h(k-1) = \log k$, and note that $\frac{2\delta}{\zeta} \leq \log k$ by assumption, so the inverse $h^{-1}\left(\frac{2\delta}{\zeta}\right)$ is well-defined. Then for every $1 \leq j \leq k$, it follows from (23), (25), and Theorem B.9 that

$$\lim_{n \to \infty} \Pr\left(\left\|\left(\frac{1}{m}\sum_{j=1}^{m}\frac{1}{n}\sum_{i=1}^{n}f_{\text{solve}}^{\alpha^{(j)}}(Q_{\mathcal{D}_0^{(j)}}, \mathcal{E}_i''^{(j)})\right) - Q_{\mathcal{D}}\right\| < \varepsilon'\right) = 1. \tag{26}$$

Note that by Lemma B.10, we have

$$\lim_{n \to \infty} \Pr\left(\left\|\frac{1}{m}\sum_{j=1}^{m}\frac{1}{n}\sum_{i=1}^{n}\left(Q_{\mathcal{D}_0^{(j)}} - (\mathcal{E}_i' + \frac{1}{k})\right)\right\| < \varepsilon'\right) = 1. \tag{27}$$

Next, apply the multilinear function $f_{\text{solve}}^{\alpha^{(j)}}(-, \mathcal{E}_i''^{(j)})$ to each term in (27); this yields

$$\lim_{n \to \infty} \Pr\left(\left\|\frac{1}{m}\sum_{j=1}^{m}\frac{1}{n}\sum_{i=1}^{n}\left(f_{\text{solve}}^{\alpha^{(j)}}(Q_{\mathcal{D}_0^{(j)}}, \mathcal{E}_i''^{(j)}) - f_{\text{solve}}^{\alpha^{(j)}}(\mathcal{E}_i' + \frac{1}{k}, \mathcal{E}_i''^{(j)})\right)\right\| < \varepsilon'\right) = 1. \tag{28}$$

Note that $\hat{Q}_i^{(j)} := f_{\text{solve}}^{\alpha^{(j)}}(\mathcal{E}_i' + \frac{1}{k}, \mathcal{E}_i''^{(j)})$ by definition, thus it follows from (26) and (28) that

$$\lim_{n \to \infty} \Pr\left(\left\|\left(\frac{1}{m}\sum_{j=1}^{m}\frac{1}{n}\sum_{i=1}^{n}\hat{Q}_i^{(j)}\right) - Q_{\mathcal{D}}\right\| < 2\varepsilon'\right) = 1. \tag{29}$$

$\square$

## B.4 PROOF OF THEOREM 2.6

Consider a valid $\alpha$-sequence $\Omega = (\alpha^{(1)}, \ldots, \alpha^{(\ell)})$ for $\mathcal{D}$. By definition, this means $\Omega$ is a subsequence of an infinite sequence $(\tilde{\alpha}^{(1)}, \tilde{\alpha}^{(1)}, \ldots)$ of distinct vectors in $[0, \frac{k-1}{k})^k$ (recall that $k$ is the number of label classes), such that $\{\tilde{\alpha}^{(i)}\}_{i=1}^{\infty}$ is a dense subset of $[0, \frac{k-1}{k}]^k$.

Let $\varepsilon > 0$. For any fixed integer $n \geq 1$, observe that

$$\lim_{\ell \to \infty} \Lambda_\varepsilon^{(n)}(\mathcal{D}_{\alpha^{(\ell)}}) \cap \Lambda_\varepsilon^{(n)}(\overline{\mathcal{D}}) = \Lambda_\varepsilon^{(n)}(\overline{\mathcal{D}}).$$

This implies that

$$\lim_{\ell \to \infty} \frac{\left|\Lambda_\varepsilon^{(n)}(\mathcal{D}_{\alpha^{(\ell)}}) \cap \Lambda_\varepsilon^{(n)}(\overline{\mathcal{D}})\right|}{\left|\Lambda_\varepsilon^{(n)}(\overline{\mathcal{D}})\right|} = 1. \tag{30}$$

In contrast, for any fixed integer $\ell \geq 1$, if $\mathbb{E}[H(\mathcal{D}_{\alpha^{(\ell)}})] \neq \mathbb{E}[H(\overline{\mathcal{D}})]$, then

$$\lim_{\varepsilon \to 0} \lim_{n \to \infty} \Lambda_\varepsilon^{(n)}(\mathcal{D}_{\alpha^{(\ell)}}) \cap \Lambda_\varepsilon^{(n)}(\overline{\mathcal{D}}) = \emptyset.$$

This implies that

$$\lim_{\varepsilon \to 0} \lim_{n \to \infty} \frac{\left|\Lambda_\varepsilon^{(n)}(\mathcal{D}_{\alpha^{(\ell)}}) \cap \Lambda_\varepsilon^{(n)}(\overline{\mathcal{D}})\right|}{\left|\Lambda_\varepsilon^{(n)}(\overline{\mathcal{D}})\right|} = 0. \tag{31}$$

**Lemma B.13.** *Let $\beta, \varepsilon > 0$, let $\mathcal{D}_0 \in \Lambda_\varepsilon^{(1)}(\overline{\mathcal{D}})$, i.e. $\mathcal{D}_0$ is an $\varepsilon$-typical baseline DILN of $\mathcal{D}$, and let $g$ be a $\mathcal{D}_0$-separable $\mathbb{R}^d$-valued random function on $\mathfrak{U}[\mathcal{D}]$. Let $r \geq 1$ be an integer, let $\Phi$ be a discriminator for $\mathcal{D}$ that is trained $r$-fold on $(\mathcal{D}_0, g)$ with threshold $\beta$, and suppose that $\Phi$ is $\delta'$-sufficient for some $\delta' > 0$. Let $\Omega = (\alpha^{(1)}, \alpha^{(2)}, \ldots)$ be a valid $\alpha$-sequence for $\mathcal{D}$. Then there exists some sufficiently large integer $\ell_\Phi \geq 1$ such that for all integers $\ell \geq \ell_\Phi$, a randomly generated $\varepsilon$-typical $\alpha^{(\ell)}$-increment DILN of $\mathcal{D}$ has non-zero probability of being predicted positive by $\Phi$, i.e.*

$$\Pr\left(\Lambda_\varepsilon^{(r)}(g(\mathcal{D}_0)) \cap \Lambda_\varepsilon^{(r)}(g(\mathcal{D}')) \neq \emptyset \middle| \mathcal{D}' \in \Lambda_\varepsilon^{(1)}(\mathcal{D}_{\alpha^{(\ell)}})\right) > 0. \tag{32}$$

*Proof.* Since $\Phi$ is $\delta'$-sufficient, we infer that (32) is true if there exists some $\mathcal{D}' \in \Lambda_\varepsilon^{(1)}(\mathcal{D}_{\alpha^{(\ell)}})$ such that $H(\mathcal{D}_0 \wedge \mathcal{D}') < \delta'$. Clearly, $H(\mathcal{D}_0 \wedge \mathcal{D}_0) = 0 < \delta'$, so it suffices to show that $\mathcal{D}_0 \in \Lambda_\varepsilon^{(1)}(\mathcal{D}_{\alpha^{(\ell)}})$. From (30), we conclude that $\mathcal{D}_0 \in \Lambda_\varepsilon^{(1)}(\mathcal{D}_{\alpha^{(\ell)}})$ is true if $\ell$ is sufficiently large. $\square$

Finally, we prove an equivalent (and rather long) reformulation of Theorem 2.6 from the main paper.

**Theorem B.14.** *Let $n, m, r, \ell \geq 1$ be integers, let $\Omega = (\alpha^{(1)}, \alpha^{(2)}, \ldots)$ be a valid $\alpha$-sequence for $\mathcal{D}$, and assume that $\zeta := \min\{p_Z(i) : 1 \leq i \leq k\} > 0$. Let $\beta, \delta, \varepsilon, \varepsilon' > 0$ be real scalars satisfying $\mathrm{gap}(\mathcal{D}) < \delta \leq \frac{1}{2}\zeta \log k$, $0 < \varepsilon \leq \delta - \mathrm{gap}(\mathcal{D})$, and $\varepsilon' := 2k \cdot h^{-1}(\frac{2\delta}{\zeta}) > 0$, where $h(x)$ is defined in (9). Also, let $g$ be a separable $\mathbb{R}^d$-valued random function on $\mathfrak{U}[\mathcal{D}]$.*

- *Let $\overline{\mathcal{D}}^{(1)}, \ldots, \overline{\mathcal{D}}^{(m)}$ be a sequence of i.i.d. random derived DILNs of $\mathcal{D}$ with the same distribution as $\overline{\mathcal{D}}$, and suppose we have observed values $\overline{\mathcal{D}}^{(1)} = \mathcal{D}_0^{(1)}, \ldots, \overline{\mathcal{D}}^{(m)} = \mathcal{D}_0^{(m)}$.*

- *For every $1 \leq s \leq \ell$ and $1 \leq j \leq m$, let $\mathcal{D}_{\alpha^{(s)},1}^{(j)}, \mathcal{D}_{\alpha^{(s)},2}^{(j)}, \ldots, \mathcal{D}_{\alpha^{(s)},n}^{(j)}$ be a sequence of i.i.d. random derived DILNs of $\mathcal{D}$ with the same distribution as $\mathcal{D}_{\alpha^{(s)}}$, and suppose we have the observed values $\mathcal{D}_{\alpha^{(s)},1}^{(j)} = \mathcal{D}_{s,1}^{(j)}, \mathcal{D}_{\alpha^{(s)},2}^{(j)} = \mathcal{D}_{s,2}^{(j)}, \ldots, \mathcal{D}_{\alpha^{(s)},n}^{(j)} = \mathcal{D}_{s,n}^{(j)}$.*

*For every $1 \leq j \leq m$, suppose that $\Phi_j$ is a discriminator for $\mathcal{D}$ that is trained $r$-fold on $(\mathcal{D}_0^{(j)}, g)$ with threshold $\beta$, and suppose that every discriminator $\Phi_j$ is $\delta'$-sufficient for some $\delta' > 0$. Then there exists some sufficiently large integer $\ell_{r,\delta'} \geq 1$ (which depends on $r$ and $\delta'$), such that for all integers $\ell' \geq \ell_{r,\delta'}$, a randomly generated $\varepsilon$-typical $\alpha^{(\ell')}$-increment DILN of $\mathcal{D}$ has non-zero probability of being predicted positive by $\Phi_j$ for all $1 \leq j \leq m$. Assume that the given integer $\ell$ is sufficiently large, i.e. $\ell \geq \ell_{r,\delta'}$. For every $1 \leq j \leq m$, define*

$$s_{\mathrm{best}}^{(j)} := \min\{s : 1 \leq s \leq \ell, \text{ there exists some } 1 \leq i \leq n \text{ such that } \mathcal{D}_{s,i}^{(j)} \in \Phi_j^+\}. \tag{33}$$

*Let $1 \leq j_1 < j_2 < \cdots < j_{m'} \leq m$ be all indices $j_t$ such that $s_{\mathrm{best}}^{(j_t)} \neq -\infty$. (By default, we define $\min \emptyset = -\infty$. Note that $m' \leq m$.)*

- *For every $1 \leq j \leq m$, let $E_1'^{(j)}, \ldots, E_n'^{(j)}$ be a sequence of i.i.d. random matrices with the same distribution as $E'$, and suppose that we have the sequence of observed values $E_1'^{(j)} = \mathcal{E}_1'^{(j)}, \ldots, E_n'^{(j)} = \mathcal{E}_n'^{(j)}$.*

- *For every $1 \leq j \leq m$, let $E_1''^{(j)}, E_2''^{(j)}, \ldots, E_n''^{(j)}$ be a sequence of i.i.d. random matrices with the same distribution as $E_{\alpha^{(s')}}''$, where $s' := s_{\text{best}}^{(j)}$, and suppose we have the sequence of observed values $E_1''^{(j)} = \mathcal{E}_1''^{(j)}, E_2''^{(j)} = \mathcal{E}_2''^{(j)}, \ldots, E_n''^{(j)} = \mathcal{E}_n''^{(j)}$.*

*For every $1 \leq i \leq n$ and $1 \leq j \leq m$, define $\hat{Q}_i^{(j)} := f_{\text{solve}}^{\alpha^{(j)}}(\mathcal{E}_i' + \frac{1}{k}, \mathcal{E}_i''^{(j)})$. Then there exist a sufficiently large integer $r_{\beta,\delta} \geq 1$ (depending on $\beta$ and $\delta$), a corresponding sufficiently large integer $\ell_{r_{\beta,\delta},\delta'} \geq 1$ (depending on $r_{\beta,\delta}$ and $\delta'$), such that for a fixed $r = r_{\beta,\delta}$, and for all $\ell \geq \ell_{r_{\beta,\delta},\delta'}$,*

$$\lim_{m \to \infty} \left[ \lim_{n \to \infty} \Pr \left( \left\| \left( \frac{1}{m'} \sum_{t=1}^{m'} \left( \frac{1}{n} \sum_{i=1}^{n} \hat{Q}_i^{(j_t)} \right) \right) - Q_\mathcal{D} \right\| < 2\varepsilon' \right) \right] = 1. \tag{34}$$

*Proof.* First of all, for each $1 \leq j \leq m$, Lemma B.13 says that there is some sufficiently large integer $\ell_{\Phi_j}$ such that for all $\ell \geq \ell_{\Phi_j}$, a randomly generated $\varepsilon$-typical $\alpha^{(\ell)}$-increment DILN of $\mathcal{D}$ has non-zero probability to be in $\Phi_j^+$. Thus, we could set $\ell_r := \max\{\ell_{\Phi_j} : 1 \leq j \leq m\}$.

For each $1 \leq t \leq m'$, define $\alpha_{\text{best}}^{(j_t)} := \alpha^{(s')}$, where $s' = s_{\text{best}}^{(j_t)}$. Observe that if

$$\left( \mathcal{D}_{s_{\text{best}}^{(j_t)},1}^{(j_t)}, \ldots, \mathcal{D}_{s_{\text{best}}^{(j_t)},n}^{(j_t)} \right) \in \Lambda_{\varepsilon'}^{(n)} \left( \mathcal{D}_{\alpha_{\text{best}}^{(j_t)}} \right) \cap (\Phi_{j_t}^+)^n, \tag{35}$$

and if $(\mathcal{D}_0^{(j_1)}, \ldots, \mathcal{D}_0^{(j_{m'})}) \in \Lambda_\varepsilon^{(m')}(\overline{\mathcal{D}})$, then Theorem B.12 yields

$$\lim_{n \to \infty} \Pr \left( \left\| \left( \frac{1}{m'} \sum_{t=1}^{m'} \frac{1}{n} \sum_{i=1}^{n} \hat{Q}_i^{(j_t)} \right) - Q_\mathcal{D} \right\| < 2\varepsilon' \right) = 1. \tag{36}$$

By definition, each $\mathcal{D}_{s_{\text{best}}^{(j_t)},i}^{(j_t)}$ (for $1 \leq i \leq n$) is already contained in $\Phi_{j_t}^+$, hence for (35) to be true, we only need to check that

$$\left( \mathcal{D}_{s_{\text{best}}^{(j_t)},1}^{(j_t)}, \ldots, \mathcal{D}_{s_{\text{best}}^{(j_t)},n}^{(j_t)} \right) \in \Lambda_{\varepsilon'}^{(n)} \left( \mathcal{D}_{\alpha_{\text{best}}^{(j_t)}} \right).$$

Now, for any $1 \leq t \leq m'$, it follows from Theorem B.3 that there exists some sufficiently large integer $n_0 \geq 1$ such that for all integers $n \geq n_0$, we have

$$\Pr \left( \left( \left( \mathcal{D}_0^{(1)}, \ldots, \mathcal{D}_0^{(n)} \right), \left( \mathcal{D}_{s_{\text{best}}^{(j_t)},1}^{(j_t)}, \ldots, \mathcal{D}_{s_{\text{best}}^{(j_t)},n}^{(j_t)} \right) \right) \in \Lambda_\varepsilon^{(n)} \left( \overline{\mathcal{D}}, \mathcal{D}_{\alpha_{\text{best}}^{(j_t)}} \right) \right) > 1 - \varepsilon'. \tag{37}$$

Consequently, if $m' \geq n_0$, then

$$\lim_{n \to \infty} \Pr \left( \left\| \left( \frac{1}{m'} \sum_{t=1}^{m'} \left( \frac{1}{n} \sum_{i=1}^{n} \hat{Q}_i^{(j_t)} \right) \right) - Q_\mathcal{D} \right\| < 2\varepsilon' \right) > 1 - \varepsilon' \tag{38}$$

Finally, the choice of $\ell_r$ implies that $\Pr(s_{\text{best}}^{(j)} \neq -\infty) > 0$, hence $m' \to \infty$ as $m \to \infty$, therefore (39) is true. $\qquad \square$

Note that the matrix

$$\hat{Q}_\mathcal{D}; = \frac{1}{m'} \sum_{t=1}^{m'} \left( \frac{1}{n} \sum_{i=1}^{n} \hat{Q}_i^{(j_t)} \right),$$

which appears in (39) is precisely the output matrix of Algorithm 1. This gives the following two corollaries.

**Corollary B.15.** *Let $\delta' > 0$. Assume that the initial conditions (on $r, m, n, \ell, \Omega, \beta, g$) given in Algorithm 1 are satisfied. Assume $k \geq 2$, assume that $\zeta := \min\{p_Z(i) : 1 \leq i \leq k\} > 0$, and assume that the discriminator $\Phi_j$ is $\delta'$-sufficient for all $1 \leq j \leq m$. If $r$ (depending on $\beta$) is sufficiently large, and if $\ell$ (depending on $r$ and $\delta'$) is sufficiently large, then the final output matrix $\hat{Q}_\mathcal{D}$ from Algorithm 1 satisfies*

$$\lim_{m \to \infty} \left[ \lim_{n \to \infty} \Pr \left( \| \hat{Q}_\mathcal{D} - Q_\mathcal{D} \| < 4k \cdot h^{-1} \left( \frac{2 \cdot \text{gap}(\mathcal{D})}{\zeta} \right) \right) \right] = 1. \tag{39}$$

*Proof.* Following the notation of Theorem B.14, note that $2\varepsilon' = 4k \cdot h^{-1}(\frac{2\delta}{\zeta}) > 0$ (where $h(x)$ is a strictly increasing function defined in (9)), and note also that $\delta > \text{gap}(\mathcal{D})$, hence the assertion follows from Theorem B.14 by choosing $\delta$ arbitrarily close to $\text{gap}(\mathcal{D})$. □

**Corollary B.16.** *Let $\delta' > 0$. Assume that the initial conditions (on $r, m, n, \ell, \Omega, \beta, g$) given in Algorithm 1 are satisfied. Assume $k \geq 2$, and assume that the discriminator $\Phi_j$ is $\delta'$-sufficient for all $1 \leq j \leq m$. Also, assume that for every possible label $i \in \mathcal{A} = \{1, \ldots, k\}$, there is at least one instance of $\mathcal{D}$ with $i$ as its correct label. If $r$ (depending on $\beta$) is sufficiently large, and if $\ell$ (depending on $r$ and $\delta'$) is sufficiently large, then the final output matrix $\hat{Q}_{\mathcal{D}}$ from Algorithm 1 converges in probability to $Q_D$ as $m \rightarrow \infty$, $n \rightarrow \infty$, and $N \rightarrow \infty$. (Here, $N$ is the number of instances in $\mathcal{D}$.)*

*Proof.* This is immediate from Corollary B.15 and Lemma B.11. □

## B.5 PRECISE REFORMULATION OF PROPOSED ALGORITHM

In the previous few subsections, we have introduced new notation and terminology, so that we are able to give a precise statement of our consistency result (Theorem B.14). Correspondingly, using our new notation and terminology, we shall also give, in this subsection, an equivalent reformulation of Algorithm 1 from the main paper.

---

**Algorithm 2** A precise formulation of algorithm to estimate $Q_{\mathcal{D}}$

---

    **Require:** integers $r, m, n, \ell \geq 1$.
    **Require:** threshold $\beta > 0$, and $g$ a separable $\mathbb{R}^d$-valued random function on $\mathfrak{U}[\mathcal{D}]$.
    **Require:** $\Omega = (\alpha^{(1)}, \ldots, \alpha^{(\ell)}) \subseteq [0, 1]^k$ a valid $\alpha$-sequence with $\ell$ vectors.
1: Initialize empty list $\mathcal{L}$.
2: **for** $j = 1 \ldots m$ **do**
3:     Generate observed value $\overline{\mathcal{D}} = \mathcal{D}_0^{(j)}$.
4:     **for** $s = 1 \ldots \ell$ **do**
5:         Generate $n$ independent observed values $\mathcal{D}_{\alpha^{(s)}} = \mathcal{D}_{s,1}^{(j)}, \mathcal{D}_{s,2}^{(j)}, \ldots, \mathcal{D}_{s,n}^{(j)}$.
6:     Let $\Phi_j$ be a discriminator trained $r$-fold on $(\mathcal{D}_0^{(j)}, g)$ with threshold $\beta$.
7:     Compute $s' := \min\{s : 1 \leq s \leq \ell, \text{ there exists } 1 \leq i \leq n \text{ such that } \mathcal{D}_{s,i}^{(j)} \in \Phi_j^+\}$.
        #[Note: $s'$ equals $s_{\text{best}}^{(j)}$ in Theorem B.14. By default, $\min \emptyset = -\infty$.]
8:     **if** $s' \neq -\infty$ **then**
9:         **for** $i = 1 \ldots n$ **do**
10:           Generate observed values for random matrices $E' = \mathcal{E}_i'^{(j)}$ and $E''_{\alpha^{(s')}} = \mathcal{E}_i''^{(j)}$.
11:           Compute $\hat{Q}_i^{(j)} := f_{\text{solve}}^{\alpha^{(j)}}(\mathcal{E}_i'^{(j)} + \frac{1}{k}, \mathcal{E}_i''^{(j)})$
          #[Note: Computation of $\hat{Q}_i^{(j)}$ involves solving a system of $k^2$ linear equations in $k^2$ variables.]
          #[Note: Unique solution to linear system exists almost surely.]
12:           **if** unique solution exists in computation of $\hat{Q}_i^{(j)}$ **then**
13:               Insert matrix $\hat{Q}_i^{(j)}$ into list $\mathcal{L}$.
14: **return** mean of matrices in $\mathcal{L}$ (This is our estimate $\hat{Q}_{\mathcal{D}}$ for $Q_{\mathcal{D}}$.)

---

## B.6 HOW TO CHECK FOR SEPARABLE RANDOM FUNCTIONS?

Roughly speaking, Fig. 3 captures the intuition on how to check experimentally whether a candidate $\mathbb{R}^d$-valued random function $g$ on $\mathfrak{U}[\mathcal{D}]$ is approximately separable. Suppose $\mathcal{D}_{\alpha_1}, \ldots, \mathcal{D}_{\alpha_\ell}$ is a sequence of $\alpha$-increment datasets for distinct vectors $\alpha = \alpha_1, \ldots, \alpha_\ell$, i.e. corresponding to different noise levels. Since $g$ is a random function, we can repeatedly call $g(\mathcal{D}_{\alpha_i})$ to get a sequence of vectors in $\mathbb{R}^d$, say of length $r$ (i.e. each of the $r$ entries is a vector). As elaborated in Section 2.3, these sequences of length $r$, which correspond to different values $\alpha_i$, are the datapoints for training discriminators; each sequence is a single datapoint for training a discriminator $\Phi$. For our random function $g$ to be separable, the datapoints (sequences) generated using a particular value for $\alpha$ should, with high probability, be distinguishable from datapoints (sequences) generated using a different value for $\alpha$. Using Fig. 3 as an example, notice that for each considered noise rate $a$ (corresponding

to $\alpha = (a, \ldots, a)$), we have plotted a coordinate-wise minimum-to-maximum range of the entries for sequences generated using $\alpha = (a, \ldots, a)$, where these sequences associated to $\mathcal{D}_\alpha$ are repeatedly generated using multiple random seeds. We call this coordinate-wise minimum-to-maximum range a "band". Now, notice that different bands, corresponding to different values for $\alpha$, are already "visually separable" in our given plot. When we train a discriminator $\Phi$ on such sequences (datapoints), those sequences in the red band are treated as "positive" datapoints, while those sequences in the blue bands (of various blue hues) are treated as "unlabeled" datapoints. The goal for $\Phi$ is to identify which blue band is "most indistinguishable" from the red band, and then use the $\alpha$ value corresponding to the identified blue band to infer a single intermediate estimate of the noise transition matrix. For this idea to work, the blue bands (for different values of $\alpha$) should themselves be distinguishable (i.e. "separable") from each other. By "distinguishable", we mean that a machine learning model (we used decision trees in our experiments) is able to distinguish (i.e. "separate") different blue bands. So informally, a candidate random function $g$ is "separable" if sequences of $g(\mathcal{D}_\alpha)$ are (with high probability) distinguishable for different values of $\alpha$. Therefore, we have experimentally verified (see Fig. 3) that $g = g_{\text{LID}}$ is a suitable function that is empirically (approximately) separable.

### B.7 STRIKING CONNECTIONS TO SHANNON'S CODING THEOREM

The significance of our proposed information-theoretic framework is not our newly introduced information-theoretic notion $H(\mathcal{D})$ of "the entropy of a DILN" per se, but rather the idea of how this notion $H(\mathcal{D})$ is used to define *typical sets* of random derived DILNs of $\mathcal{D}$ (see Definition 2.1), and correspondingly, how the idea of typical sets is used to define our notion of *separable* random functions (see Definition 2.3).

Generally speaking, $H(\mathcal{D})$ is a compact notation that is very useful for us to define typical sets (of random derived DILNs of $\mathcal{D}$) in Definition 2.1. This is very similar to the scenario of defining typical sets of random variables, in the context of the asymptotic equipartition property (AEP) theorem; see, e.g. Chapter 3.1 in Cover & Thomas (2012). Notice that the AEP theorem (Thm 3.1.1 in Cover & Thomas (2012)) is essentially a direct consequence of the weak law of large numbers. Its precise theorem statement, usually formulated in terms of the entropy of a random variable, could equivalently be formulated without any mention of, or without any interpretation involving, the notion of information entropy. Similarly, the notion of "typical sets" of random variables makes sense in the general context of probability theory, without necessarily needing any information-theoretic interpretation.

Intuitively, we should still think of "typical sets" of a random derived DILN of $\mathcal{D}$ as a "typical" sequence of DILNs, where "typical" has a precise meaning in terms of our notion of the entropy of DILNs. Just like how typical sets (of random variables) are crucial for proving Shannon's coding theorem, our notion of typical sets of random derived DILNs of $\mathcal{D}$ are crucial for proving our consistency theorem. Just like how we should think of channel coding in terms of the entropy of random variables, we should also analogously think of label noise estimation in terms of the entropy of DILNs.

As we have initially described in Section 2.3, and subsequently detailed in Appendix B.1–B.4, the notions of "typical sets" and (joint) AEP are crucial for proving our main consistency result Theorem 2.6. In fact, Theorem 2.6 can be interpreted as an "inverse" analog of (one direction of) Shannon's channel coding theorem. Perhaps, the separable random function $g$, required as input to our estimator (Algorithm 1), is analogous to a (random) error-correction code in channel coding.

The idea of "separability" for random functions could serve as a guide for the design of future estimators for $Q_\mathcal{D}$. It is plausible (likely) that the convergence rate for our consistent estimator may depend on the choice of this separable random function $g$. Hence, building on the parallelism between separable functions and error-correction codes, we pose the following question: Could we obtain more efficient estimators for $Q_\mathcal{D}$ by designing "better" separable functions for DILNs, analogous to how more efficient error-correction codes were designed to further improve communication over a noisy channel?

We are excited by the interesting questions that naturally arise from this parallelism, especially concerning the design of new and "better" estimators for noise transition matrices and more general noise models, independent of classification accuracy.

## C IMPLEMENTATION DETAILS

Algorithm 3 provided below describes in detail an explicit implementation of our proposed framework to estimate $Q_\mathcal{D}$, based on the use of LID-based discriminators. Broadly, there are three stages in our implementation, and correspondingly, we organize the rest of this section into the following three subsections:

- Section C.1: Gathering of LID sequences.
- Section C.2: Training of LID-based discriminators.
- Section C.3: Final computation to estimate $Q_\mathcal{D}$.

---

**Algorithm 3** Implementation details to estimate $Q_\mathcal{D}$.

---

**Require:** $\mathcal{D}$, a dataset with instance-independent label noise.
**Require:** $\Sigma$, a collection of random seeds, with $|\Sigma| \geq 10$.
**Require:** A valid $\alpha$-sequence $\Omega = (\alpha^{(1)}, \ldots, \alpha^{(\ell)})$.
#[Stage 1: Gathering of LID sequences]
1: **for** each random seed $\varsigma$ in $\Sigma$ **do**
2:     **for** each $\alpha_s$ in $\Omega$ **do**
3:         Generate $\alpha_s$-increment DILN $\mathcal{D}_s$.
4:         Generate LID sequences for $\mathcal{D}_s$ from a neural network.
    #[Stage 2: Training of LID-based discriminators]
5: Consolidate all generated LID sequences (from all random seeds) for initial training of LID-based discriminators.
6: **for** each triple (i.e. three seed collections) **do**
7:     Train a discriminator $\Phi$ on triple to produce a recall $\tau$ and a vote sequence.
8: Initialize a fine-tuning list $\mathcal{F}$ containing all the triples with recall $\tau \geq 0.9$.
9: Refine list $\mathcal{F}$.
10: Initialize an empty list $\mathcal{Q}$.
11: **for** each triple in $\mathcal{F}$ **do**
12:     Fine-tune discriminator trained on the triple.
13:     If the discriminator is well-trained (i.e. able to produce required recalls), then add it to $\mathcal{Q}$.
14: **if** —$\mathcal{Q}$— $< 2$ **then**
15:     Add at least 1 new random seed to $\Sigma$. Go to line 1.
16: **else**
    #[Stage 3: Final computation to estimate $Q_\mathcal{D}$]
17:     Get a prior.
18:     **for** each well-trained discriminator in $\mathcal{Q}$ **do**
19:         Compute an intermediate estimate $\hat{Q}_\mathcal{D}^{(i)}$.
20:     **for** each recall **do**
21:         Compute the mean of intermediate estimates $\hat{Q}_\mathcal{D}^{(i)}$ with the same recall.
22:     Compute the mean $\hat{Q}_\mathcal{D}$ of all means of intermediate estimates.
23: **return** $\hat{Q}_\mathcal{D}$ (This is our final estimate for $Q_\mathcal{D}$.)

---

### C.1 GATHERING OF LID SEQUENCES

In our experiments, we synthesized $\alpha$-increment DILNs of $\mathcal{D}$, using only "uniform" vectors $\alpha$, i.e. every vector $\alpha^{(s)}$ in our $\alpha$-sequence can be expressed as $\alpha^{(s)} = a_s \cdot \mathbf{1}_k$ for some $0 \leq a_s \leq 1$. (Here, $\mathbf{1}_k$ denotes the all-ones vector in $\mathbb{R}^k$.) It should be noted that our choice to use only "uniform" vectors for $\alpha^{(s)}$ is an implementation simplification, stemming from our limited computational resources. Ideally, with no computational constraints, having an $\alpha$-sequence containing "non-uniform" vectors $\alpha^{(s)}$ (i.e. whose entries are distinct) could potentially improve the final estimate, albeit with a lot more trials.

For CIFAR-10, a fixed $\alpha$-sequence $\Omega = (\alpha^{(1)}, \ldots, \alpha^{(\ell)})$ is used throughout our experiments (i.e. $\Omega$ is fixed, as we consider various random seeds). We restricted every $\alpha \in \Omega$ to be of the form

$\alpha = (a, \ldots, a)$, where the values of $a$ used are given as follows:

$$0\%, 30\%, 83\%, 84\%, 85\%, 85.4\%, 85.8\%, 86\%, 86.2\%, 86.4\%, 87.6\%, 87.7\%,$$
$$87.8\%, 87.9\%, 88.0\%, 88.1\%, 88.2\%, 88.3\%, 88.4\%, 88.5\%, 88.6\%$$

For Clothing1M, we added extra $\alpha$: 90.9%, 91.1%, 91.3%, 91.4%, 91.5%, 91.6%. Notice that these values for $a$ are not uniformly spaced apart; rather, they are concentrated in the range $[83\%, 88.6\%]$ (for Clothing1M, $[83\%, 91.6\%]$). This is because a high value of $a$ is typically required for the corresponding $\alpha$-increment DILN of $\mathcal{D}$ to have near-maximum entropy, unless $\mathcal{D}$ already has a significantly high noise level, e.g. $> 80\%$. (We define the *noise level* of $\mathcal{D}$ to be the probability that a randomly selected instance of $\mathcal{D}$ has a given label that differs from the correct label.) To accurately estimate the noise level of $\mathcal{D}$, we recommend choosing a finer $\alpha$-sequence, if computational resources allow for it. However, as a compromise, we have opted to use the above set of values for $a$ in our experiments.

### C.1.1 TRAINING OF NEURAL NETWORKS

To generate LID sequences for the CIFAR-10 dataset (Krizhevsky et al., 2009), the following 12-layer convolutional neural network (CNN) is used [5]:

- Two convolutional layers with 64 out channels with filter size of 3-by-3 and padding of size 1, each followed by a ReLU activation function. Max pooling of size of 2-by-2 is applied after the last ReLU activation function.
- Two convolutional layers with 128 out channels with filter size of 3 by 3 and padding of size 1, each followed by a ReLU activation function. Max pooling of size of 2-by-2 is applied after the last ReLU activation function.
- Two convolutional layers with 196 out channels with filter size of 3 by 3 and padding of size 1, each followed by a ReLU activation function. Max pooling of size of 2-by-2 is applied after the last ReLU activation function.
- Output vectors are flattened.
- Fully connected layer with 1024 (hidden) units, which is followed by a ReLU activation function.
- Fully connected layer with 10 output units, whose weights are set to be non-trainable (i.e. frozen).

The above model architectures are used when the dataset size is "relatively" big. When the size is small, for example, less than $40,000$ samples[6], the LID sequences generated could be identical even if the level of label noise present in the DILNs are not similar. For instance, in Fig. 3, the given datasets in both plots are clean, but the top plot corresponds to the intact CIFAR-10 dataset with $50,000$ samples, while the bottom plot corresponds to a subset of the CIFAR-10 dataset with $70\%$ anchor-like instances removed, i.e., with a total of $15,000$ samples.

For noise levels of $80\%$ and $85\%$, the LID sequences in the top plot is still "visually separable" except from epoch 9 to epoch 17. In contrast, the LID sequences on the bottom plot for the same noise levels ($80\%$ and $85\%$) overlap with each other across almost all training epochs, until they become stagnant on the 23rd epoch. We posit that such significant overlaps could vitiate the effectiveness of a discriminator $\Phi$ in distinguishing baseline DILNs from non-baseline DILNs, when $\Phi$ is trained on such overlapping LID sequences.

Therefore, to further differentiate the LID sequences, we used the trials described as follows: (i) increase the number of hidden units in the last fully connected (fc) layer, (ii) increase the number of

---

[5]Although this CNN is trained for the CIFAR-10 classification task, achieving high classification accuracy is not our goal. Instead, our goal is to gather LID sequences from the training phase. For example, given a dataset with symmetric noise rate $50\%$, when $80\%$ uniform $\alpha$-vector is used, the best test accuracy is around $25\%$. While training baseline DILNs for LID sequences, the test accuracy is around $10\%$, which is equivalent to "random guessing" (since the labels in baseline DILNs are by definition chosen uniformly at random). Despite the seemingly low test accuracies, these LID sequences are sufficient for use as training data for the discriminator $\Phi$.

[6]In one of our trials, as part of our comparison with MPEIA, we used the CIFAR-10 dataset with $90\%$ samples removed from one class. To gather LID sequences for the resulting smaller subset of CIFAR-10, we used the same neural network model architecture as in the case when the dataset is intact.

convolutional layers, (iii) reduce the training batch size, (iv) fix the weight of the last fc layer, (v) do not use softmax and batchnorm; and (vi) do not use random crop during training. See Table 4 for adjustments made when a smaller subset of the original dataset is used. Note that we did not continue using 8192 hidden units in the last fc layer for CIFAR-10 with $70\%$ anchor point removal. This is because if the batch size is small or if the number of hidden units in the last fc layer is increased, then the training time would increase significantly. Therefore, 4096 number of hidden units was used instead.

For Clothing1M, we used ImageNet-Pretrained ResNet-18 (PyTorch version). An fc layer with 512 hidden units is added to it (2 fc layers in total) and the last fc layer is fixed without updating weights during training. We refer this model as "ResNet19".

| Dataset | CIFAR-10 | | |
|---|---|---|---|
| Amount of anchor-like instances removed | 0% | 40% | 70% |
| Number of convolutional layers | 6 | 6 | 6 |
| Number of hidden units in the last fc layer | 1024 | 8192 | 4096 |
| Batch size | 128 | 128 | 32 |

Table 4: A summary of the main differences while training CNN to generate LID sequences for CIFAR-10, when different amounts of anchor-like instances are removed.

For training, after normalization, random horizontal flip with probability $0.5$ is applied for CIFAR-10 as part of data augmentation. The initial learning rate is $0.01$, which is reduced to $0.001$ at the 40th epoch for CIFAR-10. We used stochastic gradient descent (SGD) with a momentum of $0.9$ and a weight decay of $10^{-4}$. Training is stopped at the 45th epoch for CIFAR-10.

For the Clothing1M subset, images are first resized to 256X256. During training, images are randomly cropped to size 224X224 then random horizontal flip is applied with probability $0.5$. During test time, images are first resized to 256X256, then centrally cropped to 224X224. For normalization, we used mean = $[0.485, 0.456, 0.406]$ and std = $[0.229, 0.224, 0.225]$ for the 3-channel RGB images. The initial learning rate is $0.001$, which is reduced to $0.0001$ at the 5th epoch. We used stochastic gradient descent (SGD) with a momentum of $0.9$ and a weight decay of $0.05$. The model is trained for 6 epochs to gather LID sequences, with batch size 32.

In our experiments for all datasets, our computation of LID sequences follows the same process as given in (Ma et al., 2018a), with the following exceptions:

1. To remove unwanted randomness in any LID sequences computed, we re-implemented their code in PyTorch. This ensures that any random function we invoke is completely determined by the selected random seed. The unwanted randomness in the original Keras implementation includes (although not limited to) randomized weight initializations and non-deterministic cuDNN sub-routines.

2. The weights in the last fully-connected (fc) layer of the neural network are fixed throughout training, so that all weights updated during backpropagation are used in the computation of LID sequences. (Recall that LID sequences are computed using the output vectors from the last hidden layer.)

3. In (Ma et al., 2018a), 1280 samples are randomly selected from the whole dataset throughout the training epochs to compute one LID sequence. In contrast, we generated 50 random sets of 1280 sample indices to compute 50 LID sequences for $\Phi$. At each epoch, the LID scores are computed from the fixed samples.

### C.1.2 COMPARISON OF NEURAL NETWORKS USED FOR CIFAR-10 AND CLOTHING1M

To ensure a fair comparison, we chose the same backbone neural network architecture across all baseline methods, and we used a smaller architecture for our method. As far as possible, we tried to use the same number of training epochs across all baseline methods. However, the different methods are rather distinct, and employ various techniques for their estimation of $Q_{\mathcal{D}}$, such as the augmentation of the neural network architecture (e.g., one extra "distinguished" softmax layer for

S-model), a schedule using multiple loss functions for training (e.g., T-Revision uses 3 loss functions in total for training). For a more comprehensive overview of the subtle differences across the various methods, please see Tables 5 and 6, which explicitly specify the neural network architecture used, the usage of neural network, the type of loss functions, and the number of training epochs for different stages (including the computation of priors, if any).

| CIFAR-10 | (analogous) prior NN structure | (analogous) prior NN no. of training epochs (default loss: crossentropy) | main method NN structure | main method NN no. of training epochs (default loss: crossentropy) |
|---|---|---|---|---|
| S-model | ResNet18 (to initialize matrix) | 20 | ResNet18 + softmax layer | 100 |
| Forward | na | na | ResNet18 | 120 |
| T-Revision | ResNet18 (to initialize matrix) | 20 | ResNet18 (with "slack variable" component) | 200 (reweight loss) + 100 (revision loss) |
| ours-1 | ResNet18 (to obtain prior) | 20 | 12-layer CNN (6 conv layers, to gather LID sequences) | 45 |
| MPEIA | ResNet18 (ImageNet pretrained) | na | MPEIA doesn't use NN in its main method. | |
| ours-2 | We used MPEIA's estimate as our prior. | | 12-layer CNN (6 conv layers, to gather LID sequences) | 45 |
| GLC | na | na | ResNet18 | 120 |
| ours-3 | We used GLC's estimate as our prior. | | 12-layer CNN (6 conv layers, to gather LID sequences) | 45 |

Table 5: Neural network (and its usage), training losses and training epochs of all the baselines for CIFAR-10. Some methods, S-model, T-Revision, ours-1, ours-2 and ours-3, require "prior model" to initialize noise transition matrices for formal estimation. While some method, T-Revision, trains NN with special losses in its main method.

| Clothing1M subset | (analogous) prior NN (ImageNet pretrained) | (analogous) prior NN no. of training epochs (default loss: crossentropy) | main method NN (ImageNet pretrained) | main method NN no. of training epochs (default loss: crossentropy) |
|---|---|---|---|---|
| S-model | ResNet50 (to initialize matrix) | 10 | ResNet50 + softmax layer | 10 |
| Forward | na | | ResNet50 | 10 |
| T-Revision | ResNet50 (to initialize matrix) | 10 | ResNet50 (with "slack variable" component) | 10 (reweight loss) + 10 (revision loss) |
| ours-1 | ResNet19 (to obtain prior) | 10 | ResNet19 (to gather LID sequences) | 6 |
| MPEIA | ResNet50 (to obtain features) | 10 | MPEIA doesn't use NN in its main method. | |
| ours-2 | We used MPEIA's estimate as our prior. | | ResNet19 (to gather LID sequences) | 6 |
| GLC | na | | ResNet50 | 10 |
| ours-3 | We used GLC's estimate as our prior. | | ResNet19 (to gather LID sequences) | 6 |

Table 6: Neural network (and its usage), training losses and training epochs of all the baselines for Clothing1M subset. Respective prior models and losses are summarized.

### C.1.3   LID PLOTS

Fig. 3 is provided to show that LID sequences are effective in distinguishing datasets with different entropies. We used 4 clean datasets as the underlying dataset $\mathcal{D}$: CIFAR-10 intact (top plot in Fig.

3), CIFAR-10 with 70% anchor-like data removal (bottom plot in Fig. 3). For every $\alpha$-increment DILN (shown in blue) or baseline DILN (shown in red) of $\mathcal{D}$, we generated 5 such DILNs. The noise transition matrices are of symmetric form and the noise levels inserted are 0%, 60%, 80%, 85% and 87.8% (colored from the lightest blue to the darkest). We computed 50 LID sequences for each DILN, which corresponds to 50 datapoints in each epoch. To visualize the 50 sequences of a DILN, only the maximum and the minimum scores are displayed for each epoch. For CIFAR-10, this forms a band with 2 lines over the total 45 epochs. The same random seed is used for all the plots.

## C.2 TRAINING OF LID-BASED DISCRIMINATORS

Each LID-based discriminator $\Phi$ is trained using positive-unlabeled bagging (Elkan & Noto, 2008; Mordelet & Vert, 2014), with decision trees used as our sub-routine. We used $1,000$ trees and the number of unlabeled samples to draw and train each base estimator is $50$.

### C.2.1 INITIAL TRAINING

The LID sequences from $\alpha$-increment DILNs are treated as unlabeled samples while those from baseline DILNs are treated as positive. For CIFAR-10, although the maximum noise level $a_s$ injected into $\mathcal{D}$ is 88.6% to synthesize a derived DILN of $\mathcal{D}$, we used the LID sequences from $a_s$ up to 88.3% for initial training. It is recommended to reserve some LID sequences associated to the high value $a_s$ during the initial training. The reason is that it could be hard for a discriminator to get a good recall as the positive samples (LID sequences from the baseline DILN) and large number of negative samples (LID sequences from DILNs with $a_s > 0.883$) are very similar. However, for Clothing1M, we did not reserve LID sequences with high $\alpha$ values, for the purpose of training ease. Recall that $\alpha-$increment DILNs and a baseline DILN from the same common random seed is collectively called a "seed collection". Any three different seed collections is called a "triple". The discriminators would be trained on the LID sequences from all possible combinations of triples. In our experiments, we used at least $10$ random seeds for each matrix in CIFAR-10 and $20$ random seeds for Clothing1M. We do not use any augmentation or normalization of the LID sequences when training $\Phi$.

A trained discriminator $\Phi$ predicts whether an input LID sequence is similar to the LID sequences generated from baseline DILNs. If $\Phi$ assigns a score of $\geq 0.5$ to some LID sequence, then this LID sequence is predicted positive and one vote is assigned to its respective $\alpha^{(s)}$. If the score is instead $< 0.5$, then 0 vote is assigned. The vote sum of $\alpha^{(s)}$ represents the number of LID sequences a discriminator "considers" to be baseline-LID-sequences-alike. The vote sum sequence of all the $\alpha^{(s)}$ in the $\alpha$-sequence is called a "vote sequence". A trained discriminator produces a recall $\tau$ and a vote sequence.

### C.2.2 FINETUNING

After the initial training, all the triples come with their recalls and vote sequences. We only select those with recall above 0.9 and put them in a list $\mathcal{F}$ for further fine-tuning, which indicates a possibly well-trained discriminator and similar label distributions in both DILN $\mathcal{D}_0$ (the baseline dataset) and $\mathcal{D}_s$ (synthesized from $\mathcal{D}$ by injecting noise $\alpha_s$).

Triples with recall above 0.9 are added into a list $\mathcal{F}$ and we would further *refine* the list for better $Q_{\mathcal{D}}$ estimation. For CIFAR-10, if the total number of non-zero votes of all the triples in $\mathcal{F}$ before noise level 0.85 is above 4 (for Clothing1M, the noise level threshold is 0.86, since Clothing1M has wider range of $\alpha$ values), we use triples with low recall $\leq 0.92$ and abandon the remaining. Else, we use triples high recall $\geq 0.98$. Given that (for CIFAR-10 only), if the number of non-zero votes from noise levels of 0.85 to 0.86 is more than 4, only recall 0.98 is used. The intuition is that if the presence of the number of non-zero votes before noise level 0.85 (or $a_s \leq 0.85$) is not negligible (we say $\geq 4$ in this paper), $Q_{\mathcal{D}}$ could potentially possess high noise level, as "significant" number of LID sequences from $a_s \leq 0.85$ can be recognized as similar to the baseline's by $\Phi$. Experiments show that when $\mathcal{D}$ has high noise level, triples with low recall ($\leq 0.92$) can give better estimate while if $\mathcal{D}$ has low noise level, then high recall ($\geq 0.98$) is preferred. Therefore, based on recall above 0.9, we further narrow down the range of recalls of the triples to do fine-tuning. Let the new range of recalls be $\mathcal{R}$.

We then *re-consolidate* the LID sequences with the new range of $\alpha$-sequence according to $\mathcal{R}$ to finetune $\Phi$. Re-consolidation of LID sequences is necessary, because different sets of recalls require different $\alpha$-sequences to estimate $Q_{\mathcal{D}}$. The higher the recall, the smaller the range of possible vectors we are allowed to use in the $\alpha$-sequence. If an inappropriate $\alpha$-sequence is used, then the matrix $Q_{\mathcal{D}}$ could have illegal entries (i.e. with values $< 0$ or $> 1$). During the initial training, for CIFAR-10, LID sequences are consolidataed with $a^{(s)} \leq 0.883$, then we select triples with recall above 0.9. For Clothing1M, the maximum $\alpha$ used for initial training has noise rate $91.6\%$. Now with refinement, the range of recall is narrower, the respective range of LID sequences would have to be adjusted for later fine-tuning.

With the re-consolidated LID sequences, we can fine-tune $\Phi$ for the same triple, $\mathcal{T}$, for five times. Let the new recall during fine-tuning be $\tau'$ and the old recall in the initial training be $\tau$. If $\tau' = \tau$ for at least 3 times, we can stop fine-tuning. Else, we abandon this triple. And now the discriminator for $\mathcal{T}$ is considered well-trained, which comes together with $\tau'$ and the new vote sequence. In the new vote sequence, the $\alpha_s$ with the highest number of vote sum is denoted as top-voted $a^*$. $\tau'$ and top-voted $a^*$ would be used for $Q_{\mathcal{D}}$ estimation.

For each $\mathcal{D}$, there shall be at least 2 discriminators well-trained. Otherwise, more random seeds are required to synthesize DILNs of $\mathcal{D}$, generate LIDs and train $\Phi$.

### C.3 FINAL COMPUTATION TO ESTIMATE $Q_{\mathcal{D}}$

#### C.3.1 PRIORS

To learn $Q_{\mathcal{D}}$, a prior, $P$, is used to help define the structure of $Q_{\mathcal{D}}$. For CIFAR-10, we used ResNet-18 (He et al., 2016) to train on $90\%$ of the given noisy dataset for 20 epochs and validate on the $10\%$ instances. For Clothing1M, we used ImageNet pretrained ResNet18 with an extra fc layer (512 hidden units) to train and validate it for 10 epochs. For respective datasets, the optimization procedure used are the same as given in (Patrini et al., 2017). The probabilities of the training instances from the softmax layer are recorded. The probabilities from the epoch with the highest validation accuracy is used to compute a prior. Let $\Upsilon(y_n = i)_j$ be the $j$-th output from the neural network's softmax layer for some sample with label $i$, and let the set of samples with label $i$ be $S_i$, i.e.

$$P_{i,j} = \frac{1}{|S_i|} \sum_{n \in S_i} \Upsilon(y_n)_j$$

#### C.3.2 FURTHER SIMPLIFICATIONS IN MODELING

Consider an arbitrary $\alpha \in [0,1]^k$. Recall that when we equate the random matrices $Q_{\overline{\mathcal{D}}} = Q_{\mathcal{D}_\alpha}$, we get the following equation for each entry:

$$\frac{1}{k}(1 + E'_{i,j}) = q_{i,j}(1 - \alpha_j^*) + \sum_{\substack{1 \leq t \leq k \\ t \neq j}} q_{i,t}\alpha_t^* \frac{1}{k-1}(1 + E''_{i,j}) \quad \text{(for all } 1 \leq i, j \leq k) \tag{40}$$

To simplify computations in our estimation, for each row, we estimate only one entry $q_{i,j}$ and the remaining entries shall be proportional to $1 - q_{i,j}$ according to a prior. This reduces the estimation "burden" from $k^2$ entries to $k$. The coordinate $(i,j)$ is determined by the maximum entry's location of the $i$-th row in the prior: $j = \operatorname{argmax}_m p_{i,m}$. The entry $q_{i,t}$ is substituted with $\frac{p_{i,t}}{\sum_{k \neq j} p_{i,k}}(1 - q_{i,j})$.

Next, we estimate $E''_{i,j}$ and $E'_{i,j}$. For $\overline{\mathcal{D}}$, let the binary random variable $X_n$ be 1 if the $n$-th sample has been assigned a wrong label. The total number of instances in $\mathcal{D}$ is $N$. Hence, $X_n$ is a Bernoulli random variable with parameter $\frac{k-1}{k}$ and the noise level in $\overline{\mathcal{D}}$ is $\sum_{n=1}^{N} \frac{X_n}{N} \approx \frac{k-1}{k} + E^{\overline{\mathcal{D}}}$, where $E^{\overline{\mathcal{D}}}$ is a random variable that approximates the relabeling process. By the central limit theorem, we infer that $\sum_{n=1}^{N} \frac{X_n}{N}$ is approximately normal with mean $\frac{k-1}{k}$ and variance $\frac{k-1}{Nk^2}$. Let $z$ be a random variable following standard normal distribution. Normalizing, we get:

$$z = \frac{\sum_{n=1}^{N} \frac{X_n}{N} - \frac{k-1}{k}}{\sqrt{\frac{k-1}{Nk^2}}} = \frac{E^{\overline{\mathcal{D}}}}{\sqrt{\frac{k-1}{Nk^2}}}$$

Let the recall from the well-trained discriminator be $\tau'$, where $\tau' = \Pr(-\theta \leq z \leq \theta)$, or $\frac{1-\tau}{2} = P(z \leq -\theta)$. From the standard normal distribution table, $\theta$ can be found when $\tau'$ is defined. Therefore $E^{\overline{\mathcal{D}}} = -\theta\sqrt{\frac{k-1}{Nk^2}}$. We make a simplification that $\frac{1}{k}E'_{i,j} \approx E^{\overline{\mathcal{D}}}$ for all $(i, j)$. Both random variables $E''_{i,j}$ and $E'_{i,j}$ are assumed to follow multivariate normal distributions, with unknown parameters to us. To further simplify the estimation, we model both random variables with a common random variable "$E$". This simplification has been shown reasonable by experiments:

$$E'_{i,j} = -(k-1)E$$

and

$$E''_{i,j} = (k-2)E$$

Suppose $\mathcal{R} = \{r_1, \ldots, r_m\}$. For each $1 \leq s \leq m$, let $\mathcal{I}_s$ denote the set of all estimates with associated recall $r_s$. Therefore, our final estimate for $Q_{\mathcal{D}}$ is:

$$\hat{Q_{\mathcal{D}}} = \sum_s^m \frac{\sum_{i \in \mathcal{I}_s} \hat{Q}_{\mathcal{D}}^{(i)}}{|\mathcal{I}_s|}$$

## D    RELATION TO THE INFORMATION BOTTLENECK THEORY FOR DEEP LEARNING

The momentous work on the Information Bottleneck (IB) theory for deep learning (Tishby & Zaslavsky, 2015; Shwartz-Ziv & Tishby, 2017) has spurred much interest and discussion; see (Gabrié et al., 2018; Saxe et al., 2018). A key aspect proposed by the IB theory is that the training of a deep neural network (DNN) consists of two distinct phases: An "expansion" phase where the mutual information between layers increase, and a "compression" phase, where the mutual information between layers decrease. The notion of local intrinsic dimensionality (LID), which we used as an essential ingredient for a concrete realization of our proposed framework, was in part motivated by this IB theory. It was reported in (Ma et al., 2018c) that for their experiments on training DNNs, the LID scores of the training epochs also exhibits a similar two-phase phenomenon: An initial decrease in LID score, then either an increase in LID score or a stagnant LID score, depending on whether there is label noise or not, respectively. As part of our experiments, we synthesized multiple datasets with varying entropy, and generated multiple LID sequences for each synthesized dataset. Although our focus is on the estimation of the noise transition matrices of datasets with instance-independent label noise, a by-product of our work is that we observed a very wide spectrum of behavior for LID sequences, some of which the two-phase phenomenon is not obvious. Notice that in Fig. 3, we have included multiple plots of LID sequences at various entropies, which may be of independent interest to other researchers (especially those directly working on the IB theory). A caveat is that LID scores are distinct from the mutual information between layers, but LID scores could still be interpreted as a measure of model complexity, which presumably have close connections to the generalizability of deep learning.

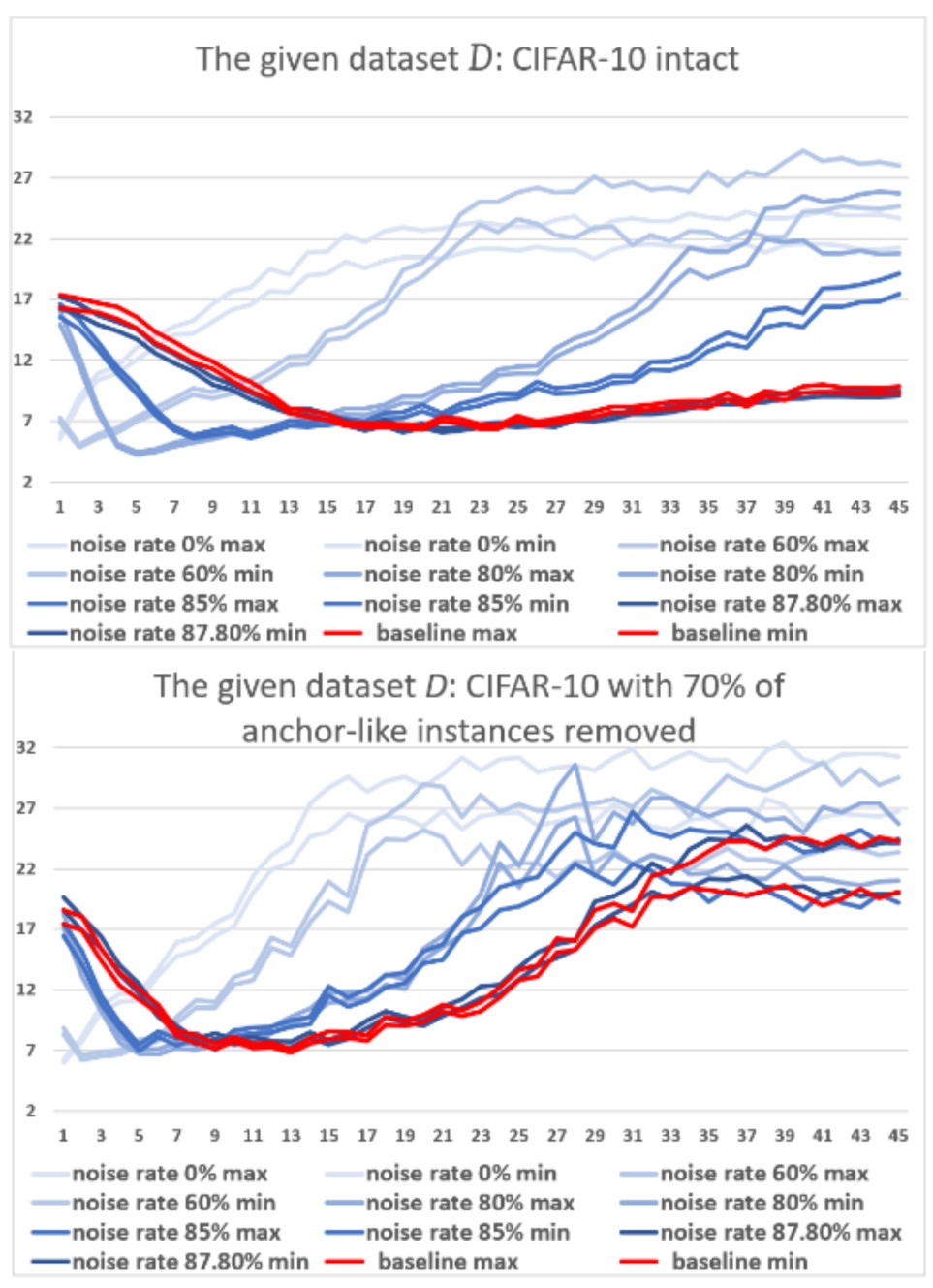

Figure 3: LID sequences visualization for clean CIFAR-10 intact as $\mathcal{D}$ (top plot) and clean CIFAR-10 with 70% of anchor-like instances removed as $\mathcal{D}$ (bottom plot). The $x$-axis represents epochs, and the $y$-axis represents individual LID scores (at each epoch). Notice that for each considered noise level $a$ (corresponding to $\alpha = (a, \dots, a)$), we have a coordinate-wise minimum-to-maximum range of the entries for sequences generated using $\alpha = (a, \dots, a)$, where these sequences associated to $\mathcal{D}_\alpha$ are repeatedly generated using multiple random seeds. We call this coordinate-wise minimum-to-maximum range a "band". Blue bands represent LID sequences associated to $\alpha$-increment DILNs with noise levels 0%, 60%, 80%, 85% and 87.8% (from lightest blue to darkest blue). Red bands represent LID sequences associated to baseline DILNs.

