# OpenReview forum: "An information-theoretic framework for learning models of instance-independent label noise"
_ICLR.cc/2021/Conference — Reject_

### Official Review · AnonReviewer3 · 2020-10-28
**Writing could be improved**

**Rating:** 5
**Confidence:** 3

**Review:**

This paper studies the problem of learning the noise model in a classification problem. The basic setup is that
X -> Z -> Y
is a Markov chain, where X are the features/input to the classifier, Z is the true label, and Y is the observed noisy label. The term "instance-independent label noise" refers to the fact that the law of Y | Z doesn't depend on X. The goal is to learn the noise model, i.e. the transition kernel Z -> Y.

I feel the writing clarity could be improved in places. The definition of DILN in the main text is confusing (does it include the noise model or not?) although the presentation in Appendix A is somewhat clearer. A lot of notation is introduced and it makes it difficult to read the main text -- for example the algorithm description depends on a "valid alpha-sequence" as input and this is not defined until the appendix. An informal definition in main text would be helpful.

As far as I understood the theorem statement, it says that if the algorithm is given as input a 'separable map' g then it is possible to learn the noise model, and the main difficulty to apply it is to construct an efficient separable map (which in practice, they attempt with decision trees + LID scores). It would be nice if the authors discuss the conditions under which such a separable map is guaranteed to exist. (And ideally, under which we could hope to find such a map.)

Overall, I think there may be some interesting ideas in this paper, but I did not find the results to be very strong support for the claims made (see other notes below regarding the experiments). I also think the paper could benefit from clearer writing. So right now, I would tend towards rejection.

Other notes:
* The beginning of the paper suggests that the key contribution is to learn the noise model without needing to "accurately" learn a classifier X -> Z. However, if I understand right the final algorithm based off of "Local Intrinsic Dimensionality" (LID) scores requires training a 12-layer CNN to predict the label from X so it seems to be somewhat weak support for this claim?

* I found the definition of f_{matrix} confusing. If U[D] indicates we throw away the information about the noise model, then what does the map f_{matrix} : D' |-> Q_{D'} mean, is Q_{D'} is still determined from D'? Is this a 'random map'?

---

> ### Author Response · Authors · 2020-11-20
> **Response to Reviewer 3 (3/3)**
>
> %%% "Overall, I think there may be some interesting ideas in this paper, but I did not find the results to be very strong support for the claims made (see other notes below regarding the experiments). I also think the paper could benefit from clearer writing. So right now, I would tend towards rejection." %%%
>
> Thank you for finding our paper interesting. We hope you agree that a trained classifier with maximum $25$% test accuracy is considered "inaccurate". We also hope that our revised paper is now clearer, especially with the inclusion of our new Sec. 2.3, which in particular articulates the close connection that our consistency theorem has with AEP, and our new Appendix B.6, which explains how to check for separable random functions.
>
> We would like to further point out that to the best of our knowledge, this is the first time that the task of minimizing the estimation error for the noise transition matrix of a DILN $\mathcal{D}$ is decoupled from the task of maximizing the classification accuracy of a classifier trained on $\mathcal{D}$. Consequently, our proposed information-theoretic framework suggests that it may be possible to accurately estimate the noise transition matrices of DILNs, in the context of a classification task that is "inherently still difficult" to get high classification accuracy even without label noise.
>
> If we may suggest, our proposed information-theoretic framework can be interpreted as an "inverse" analog of (one direction of) Shannon's channel coding theorem. Perhaps, the separable random functions $g$ introduced in our paper are analogous to (random) error-correction codes. We are excited by the interesting questions that naturally arise from this parallelism, especially concerning the design of new estimators. Further discussion can be found in the newly inserted Appendix B.7. We would love to hear your comments on this.

---

> ### Author Response · Authors · 2020-11-20
> **Response to Reviewer 3 (2/3)**
>
> %%% "As far as I understood the theorem statement, it says that if the algorithm is given as input a 'separable map' g then it is possible to learn the noise model, and the main difficulty to apply it is to construct an efficient separable map (which in practice, they attempt with decision trees + LID scores). It would be nice if the authors discuss the conditions under which such a separable map is guaranteed to exist. (And ideally, under which we could hope to find such a map.)" %%%
>
> This is an excellent point! And yes, your understanding of the theorem statement is correct. We agree as well that it would be nice to find conditions under which such a separable function $g$ is guaranteed to exist. We are unable to prove (non-trivial) sufficient conditions for "separability". However, what we are able to do is to describe necessary conditions that a separable function must satisfy, which can then be used in practice to checked experimentally, whether a candidate random function $g$ is at least approximately separable. Please see our response to Reviewer 1 (part 2/5) for details on this question. Further details can be found in the newly inserted Appendix B.6 in the revised paper.
>
> Note that the intuition for finding suitable separable functions $g$ (as described in the new Appendix B.6) was already implicit in our experiments, especially in the choices made to better distinguish or "separate" the LID sequences corresponding to different noise levels of the additional noise inserted. To improve clarity, we have also made this explicit in our revised paper; see Appendix C.1.1.
>
> We believe our information-theoretic approach raises interesting new questions that merit further research, especially the problem of finding rigorous characterizations of separable functions that can yield "better" estimators. We hope that the lack of provably sufficient conditions (for "separability") does not diminish the significance of our current contributions in the paper.
>
>
> %%% "The beginning of the paper suggests that the key contribution is to learn the noise model without needing to "accurately" learn a classifier X -> Z. However, if I understand right the final algorithm based off of "Local Intrinsic Dimensionality" (LID) scores requires training a 12-layer CNN to predict the label from X so it seems to be somewhat weak support for this claim?" %%%
>
> We would like to clarify that although the LID sequences are obtained by training a 12-layer CNN on the noisy dataset for the classification task (i.e. predicting the labels of $X$), we do not actually train the CNN to achieve a high classification accuracy, and we do not tune any hyperparameters to improve classification accuracy. For example, in the case of $50$% symmetric noise as the underlying noise model for $\mathcal{D}$, and with no anchor points removed from $\mathcal{D}$, the highest test accuracy we obtained for our CNN was around $25$% (using a total of $45$ training epochs), which is rather low. These LID sequences were then used to train discriminators that eventually gave better estimates for the noise transition matrix $Q_{\mathcal{D}}$, as compared to existing competitive baselines; see Table 1. Hence, we believe our experiments do indeed show that "accurately" learning a classifier $X \to Z$ is not required for estimating $Q_{\mathcal{D}}$ well. We have added more details in footnote 4 of Appendix C.1.1 to clarify this.
>
> We do, however, tune hyperparameters to better differentiate (i.e. "separate") the LID sequences corresponding to different noise levels. Our experiments reveal that it is possible to obtain better LID sequence "separation" without necessarily improving the classification accuracy, which can be interpreted as further supporting evidence to our claim that "accurately" learning a classifier is not required for estimating $Q_{\mathcal{D}}$ well.
>
> Please see Appendix C.1.1 in our revised paper for more details.
>
>
> %%% "I found the definition of f_{matrix} confusing. If U[D] indicates we throw away the information about the noise model, then what does the map  f_{matrix}  : D' |-> Q_{D'} mean, is Q_{D'} is still determined from D'? Is this a 'random map'?" %%%
>
> We apologize for the typo. The domain of $f_{\text{matrix}}$ should be $\mathfrak{D}[\mathcal{D}]$, not $\mathfrak{U}[\mathcal{D}]$. The matrix $Q_{\mathcal{D}'}$ is determined by the noise model $(Y'|Z; \mathcal{A})$ of the DILN $\mathcal{D}'$. In particular, $f_{\text{matrix}}$ is not a random map. As a side remark, note that $f_{\text{matrix}}(\overline{\mathcal{D}})$ is a random matrix; this is because $\overline{\mathcal{D}}$ is a random DILN. We sincerely hope that our typo did not cause too much confusion.

---

> ### Author Response · Authors · 2020-11-20
> **Response to Reviewer 3 (1/3)**
>
> Thank you very much for taking time to review our paper. We greatly appreciate your comments and suggestions! Below, we address the specific concerns raised.
>
> %%% "I feel the writing clarity could be improved in places. The definition of DILN in the main text is confusing (does it include the noise model or not?) although the presentation in Appendix A is somewhat clearer. A lot of notation is introduced and it makes it difficult to read the main text -- for example the algorithm description depends on a "valid alpha-sequence" as input and this is not defined until the appendix. An informal definition in main text would be helpful." %%%
>
> Thank you very much for your suggestions. To improve clarity, we have made several changes in our revised paper:
>
> - We have included a new subsection Sec. 2.3, which describes the underlying intuition for why our proposed algorithm works as intended, and which explains how discriminators can be trained to predict whether an input DILN has near-maximum entropy. This subsection also highlights the crucial role of "typicality" and the asymptotic equipartition property (AEP) from information theory, especially in proving our main consistency theorem.
>
> - We have included (and simplified) the definition of "valid $\alpha$-sequence" as Definition 2.5 in the main text. Previously, this definition was postponed to the appendix. Now, it precedes the main theorem of our paper (renumbered to Theorem 2.6).
>
> - We have made Theorem 2.6 "self-contained" by explicitly giving all input assumptions of Algorithm 1 in the statement of the theorem.
>
> - We have included an informal description (after Theorem 2.6) on how the input parameters to our algorithm should be interpreted.
>
> - We have included a new subsection B.6 in the appendix, to describe necessary conditions for a random function $g$ to be separable, as well as to explain how these necessary conditions can be used in practice to check experimentally, whether a candidate random function is separable.
>
> - We have included a new subsection B.7 in the appendix, to highlight the significance of our theoretical contributions. In Appendix B.7, we elaborate on the striking connections that our proposed information-theoretic framework have with Shannon's channel coding theorem, and we also discuss a parallelism between our new notion of separable random functions, and the notion of error-correction codes in channel coding. In particular, we explain why the idea of "separability" could serve as a guide for the design of future estimators for $Q_{\mathcal{D}}$.
>
> To address your specific question on the definition of a DILN, please note that a DILN includes its noise model $(Y|Z; \mathcal{A})$. To avoid any ambiguity, we have included the sentence "In particular, a DILN includes its noise model." in Footnote 1. To minimize confusion, we made a small but hopefully useful change: We now use "instance of $\mathcal{D}$" instead of "instance in $\mathcal{D}$". The last paragraph of Appendix A has also been updated to clarify the definition of DILNs.
>
> As for the heavy use of notation, this is unfortunately an inherent part of describing both our proposed algorithm, and our consistency result (Theorem 2.6). To describe our algorithm, we have to introduce notation for the different random DILNs used. To define "separable", we have to introduce notation for typical sets. These are crucial notions for understanding our contributions, so we have decided to keep them. We hope that with the changes made (as described above), the main text is now clearer and easier to understand.

---

### Official Review · AnonReviewer1 · 2020-10-30
**The theoretical contribution is not so clear/significant, but the experimental results seem reasonable.**

**Rating:** 5
**Confidence:** 3

**Review:**

***

Summary:

The paper aims to develop an information-theoretic framework for learning the underlying model from data with instance-independent label noises. It first formalizes the problem and relevant definitions with information measures, such as conditional entropy and conditional KL divergence, then argues that if the underlying model is well-separated, the proposed algorithm will be consistent in learning the noise-labeling matrix. At the heart of the algorithm is a discriminator being able to measure the labelings' noise level. A class of discriminators based on 'local intrinsic dimension (LID)' is then provided and evaluated empirically. The resulting algorithm performs well on the tested instances compared to existing ones.


***

Reasons for score:


Overall, I would like to give a weak reject. I like the idea of understanding and analyzing data models with label-independent noises through the lens of information theory. It is even better as such an idea leads to an improvement in practical applications. However, the paper is unsatisfiable in terms of writing and theoretical contribution. The authors may refer to the 'cons' below.


***

Pros:


1. The problem of learning models of instance-independent label noise is, by itself, an important and practical problem.

2. The proposed framework and algorithm (Algorithm 1) are both general/flexible and work without additional structure assumptions on the noise transition matrix. Besides, the information-theoretic formulation is also natural for the following reason. A trained discriminator is used to infer how different the noise matrix is from a completely random matrix, namely, a matrix with all rows being uniform. By inserting user-designed noise and training multiple discriminators, one effectively probes the underlying matrix and, in the limit, obtains sufficient information for its recovery.

3. This paper provides a useful estimator that performs well in practical experiments. In particular, it is shown that the proposed algorithm can be combined with prior ones to improve their efficiency/accuracy.


***

Cons:

1. I am concerned about the significance of the theoretical results. The major contribution is Theorem 2.5 on page 6, which states that the proposed algorithm is consistent in estimating the noise matrix.

- It might be better to make the main theorem self-contained. Currently, the major assumptions appear at the beginning of Algorithm 1.

- The paper suggests that its results hold under "mild" conditions on the discriminator. I believe this requires further clarification, especially on why these conditions are mild, given the existing literature. For example, the algorithm relies on the existence of a separable random function $g$ on $\mathfrak U[\mathcal D]$. I wonder if this condition can be checked in practice only by utilizing the data OR whether there is a practical scenario where this condition is automatically satisfied. Besides, how strong is this condition? Is such a condition satisfied by the random function $g_{\text{LID}}$ used in the experiments?

- The theoretical results show that the estimator is consistent if (nearly) all parameters tend to infinity, which doesn't seem practical. I don't see how this theoretical insight can guide the design of real estimators.


2. I am also concerned about the writing of the paper due to the following reasons.

- The notation is somewhat heavy, and I feel that more than half of the paper is introducing new definitions. Besides, there are few issues with those definitions/notation: 1) Notation introduced but merely used, e.g., the KL divergence between two derived DILNs on page 4; 2) Quantities undefined or defined after being used, e.g., in Definition 2.4 of a trained $n$-fold estimator,  what is $\Phi^+$?  For another example, what is $E_{i,j}'$, referred to as "error" in the definition of $Q_{i,j}'$ on page 4? 3) For information-theoretical quantities, it might be better also (or just) presenting their standard forms, e.g., the key definitions $H(\mathcal D)$ and $\text{KL}(\mathcal D'\vert\\!\vert \mathcal D'')$ in Section 2.1 are basically $H(Y\vert Z)$ and $\text{KL}(Y'\vert Z\vert\\!\vert Y''\vert Z)$. Also, what is the novelty here then?

- I would suggest adding more explanations about the theorem and algorithm. The paper currently spends lots of effort defining new quantities but much less in providing insights and intuitions about the algorithm and its guarantee. For example, after the heavy notation on page 5, the main algorithm is directly presented at the top of page 6, followed by its guarantee, Theorem 2.5. As the algorithm is nontrivial, it would be helpful to explain the logic and rationality. In particular, a critical definition, valid $\alpha$-sequence, is postponed to the appendix, and I don't see why. Similar comments apply to the main theorem, as I mentioned in the last major point.

I hope that the authors can address (at least some of) the concerns/questions/suggestions stated above. Thanks.


***

---

> ### Author Response · Authors · 2020-11-20
> **Response to Reviewer 1 (5/5)**
>
> %%% "I would suggest adding more explanations about the theorem and algorithm. The paper currently spends lots of effort defining new quantities but much less in providing insights and intuitions about the algorithm and its guarantee. For example, after the heavy notation on page 5, the main algorithm is directly presented at the top of page 6, followed by its guarantee, Theorem 2.5. As the algorithm is nontrivial, it would be helpful to explain the logic and rationality." %%%
>
> Once again, we apologize for not being sufficiently clear in our original paper submission. We appreciate your feedback, and we have revised our paper accordingly to better explain the intuition for our algorithm and consistency proof. Specifically, the new subsection Sec. 2.3 (which we described earlier) was inserted into the revised paper precisely to address this concern. The importance and intuition of "typicality" is emphasized in Sec. 2.3. In addition, we have included an informal description (after stating the main consistency theorem) on how the input parameters to our algorithm should be interpreted. Please see our response to Reviewer 3 (part 1/3) for a list of major changes made to improve clarity in the revised paper.
>
> %%% "In particular, a critical definition, valid $\alpha$-sequence, is postponed to the appendix, and I don't see why. Similar comments apply to the main theorem, as I mentioned in the last major point." %%%
>
> Thank you for the suggestion. We have included (and simplified) the definition of "valid $\alpha$-sequence" as Definition 2.5 in the main text. Previously, this definition was postponed to the appendix. Now, it precedes the main theorem of our paper (renumbered to Theorem 2.6).
>
> %%% "I hope that the authors can address (at least some of) the concerns/questions/suggestions stated above. Thanks." %%%
>
> Thank you for your detailed feedback! We appreciate your specific questions very much. We believe we have addressed all of your concerns/questions/suggestions. Please let us know if any of your concerns/questions/suggestions are still not fully addressed. Also, we hope that our theoretical contributions are now clearer with the modifications made in our revised paper, and we would love to hear your feedback on our revision.

---

> ### Author Response · Authors · 2020-11-20
> **Response to Reviewer 1 (4/5)**
>
> %%% "2) Quantities undefined or defined after being used, e.g., in Definition 2.4 of a trained $n$-fold estimator, what is $\Phi^{+}$? For another example, what is $E_{i,j}'$, referred to as "error" in the definition of $Q_{i,j}'$ on page 4?" %%%
>
> Please note that $\Phi^{+}$ is defined in the last sentence of the second paragraph of Sec. 2.4 (previously Sec. 2.3), appearing together with the terminology related to discriminators.
>
> As for your question on $E_{i,j}'$, note that we use "error" (in quotation marks) to informally mean $\tfrac{1}{k}$ times the "deviation from the expected value". Note that $\overline{\mathcal{D}}$ is by definition a random DILN (whose observed values are called baseline DILNs), satisfying the condition that the label of each instance is selected uniformly at random from one of the $k$ classes, Hence, $Q_{\overline{\mathcal{D}}}$ is a random matrix. We denote the $(i,j)$-th entry of $Q_{\overline{\mathcal{D}}}$ by $Q_{i,j}'$, which is a random variable with mean $\tfrac{1}{k}$. Intuitively, for any baseline DILN (i.e. observed value of $\overline{\mathcal{D}}$), every entry of its corresponding noise transition matrix is approximately $\tfrac{1}{k}$, and $E_{i,j}'$ is a (scaled) measure of the approximation error.
>
> %%% "3) For information-theoretical quantities, it might be better also (or just) presenting their standard forms, e.g., the key definitions $H(D)$ and $KL(D'||D'')$ in Section 2.1 are basically $H(Y|Z)$ and $KL(Y'|Z || Y''|Z)$. Also, what is the novelty here then?" %%%
>
> Yes, it is true that $H(\mathcal{D})$ equals $H(Y|Z)$ (and similarly, that $\mathrm{KL}(\mathcal{D}'||\mathcal{D}'')$ equals $\mathrm{KL}(Y'|Z || Y''|Z)$) as formal mathematical expressions. However, we believe it is conceptually easier to think of $H(Y|Z)$ as the "entropy" of $\mathcal{D}$, without explicitly mentioning the true labels $Z$. This is especially pertinent in Definition 2.1 and Definition 2.3, where we introduced the notions of typical sets and separability in terms of $H(\mathcal{D})$. In addition, our new notion of $H(\mathcal{D})$ is not "merely" conditional entropy; we also introduce what we call the "transverse entropy" of two derived DILNs of $\mathcal{D}$ (see Appendix B.1), which has no corresponding analog in the usual notion of entropy for random variables.
>
> Generally speaking, $H(\mathcal{D})$ is a compact notation that is very useful for us to define typical sets (of random derived DILNs of $\mathcal{D}$) in Definition 2.1. This is very similar to the scenario of defining typical sets of random variables, in the context of the asymptotic equipartition property (AEP) theorem; see, e.g. "Elements of information theory" (by Cover--Tomas), Chapter 3.1. Notice that the AEP theorem is essentially a direct consequence of the weak law of large numbers. Its precise theorem statement, usually formulated in terms of the entropy of a random variable, could equivalently be formulated without any mention of, or without any interpretation involving, the notion of information entropy. Similarly, the notion of "typical sets" of random variables makes sense in the general context of probability theory, without necessarily needing any information-theoretic interpretation.
>
> Intuitively, we should still think of "typical sets" of a random derived DILN of $\mathcal{D}$ as a "typical" sequence of DILNs, where "typical" has a precise meaning in terms of our notion of the entropy of a DILN. The novelty here is not this new definition of $H(\mathcal{D})$ per se, but rather its use to define the notion of typical sets of random derived DILNs of $\mathcal{D}$. Consider the following: Just like how typical sets (of random variables) are crucial for proving Shannon's coding theorem, our notion of typical sets of random derived DILNs of $\mathcal{D}$ are also crucial for proving our consistency theorem. Just like how we should think of channel coding in terms of the entropy of random variables, we should also analogously think of label noise estimation in terms of the entropy of DILNs.

---

> ### Author Response · Authors · 2020-11-20
> **Response to Reviewer 1 (3/5)**
>
> %%% "The theoretical results show that the estimator is consistent if (nearly) all parameters tend to infinity, which doesn't seem practical. I don't see how this theoretical insight can guide the design of real estimators." %%%
>
> Thank you for pointing this out. We do acknowledge this was a concern we initially had. Indeed, if (nearly) all parameters have to be extremely large just to get a decently small estimation error for the noise transition matrix, then our proposed algorithm would not be practical. Viewed through this perspective, it is then very encouraging that our experimental results indicate otherwise. The input parameters we used in our experiments are not too large (see the Appendix for details), and already we see that our estimator generally outperforms existing state-of-the-art estimators. Crucially, in the case when no clean subset of the dataset is provided, these competitive baselines (S-model, Forward, T-Revision) do not have theoretical guarantees on consistency, while in contrast, we proved that our estimator is consistent. Informally, this implies that if we desire an even better estimate, then we "merely" need more computational resources.
>
>
> The key introduction of a "separable" random function could serve as a guide for the design of future estimators. It is plausible (likely) that the convergence rate for our consistent estimator may depend on the choice of this separable random function $g$. Hence, building on the parallelism between separable functions and error-correction codes that we mentioned earlier, our main theorem suggests that more efficient estimators could be obtained by designing "better" separable functions for DILNs, analogous to how more efficient error-correction codes were introduced after Shannon's channel coding theorem was proven. Error-correction codes were designed to further improve communication over a noisy channel, so analogously, we envision that separable functions could be designed to further improve the estimation of noise transition matrices (and hence improve the performance of classifiers).
>
> We are excited by the theoretical insights that our proposed information-theoretic framework reveal, and we hope to further build upon this framework to better tackle the issue of label noise when training classifiers.
>
>
> %%% "I am also concerned about the writing of the paper due to the following reasons. The notation is somewhat heavy, and I feel that more than half of the paper is introducing new definitions." %%%
>
> Thank you for raising this concern. To improve clarity, we have made several changes in our revised paper, including the insertion of a new subsection Sec. 2.3, which describes the underlying intuition for our main algorithm. (Please see our response to Reviewer 3 (part 1/3) for a list of major changes made to improve clarity in the revised paper.)
>
> As for the heavy use of notation, this is unfortunately an inherent part of describing both our proposed algorithm, and our consistency result (Theorem 2.6). To describe our algorithm, we have to introduce notation for the different random DILNs used. To define "separable", we have to introduce notation for typical sets. These are crucial notions for understanding our contributions, so we have decided to keep them. We hope that with the changes made, the main text is now clearer and easier to understand.
>
> %%% "Besides, there are few issues with those definitions/notation: 1) Notation introduced but merely used, e.g., the KL divergence between two derived DILNs on page 4; "%%%
>
> Thank you for pointing this out. We have checked that this notion of KL divergence between two derived DILNs is not absolutely necessary. It was implicitly used in the context of "forward KL loss" as the evaluation metric for our experiments. In the revised paper, we have removed the definition of "KL divergence between two derived DILNs" from page 4, and we have instead defined "forward KL loss" directly in Sec. 4 (section on experiments); see the newly inserted Footnote 3.

---

> ### Author Response · Authors · 2020-11-20
> **Response to Reviewer 1 (2/5)**
>
> %%% "The paper suggests that its results hold under "mild" conditions on the discriminator. I believe this requires further clarification, especially on why these conditions are mild, given the existing literature. For example, the algorithm relies on the existence of a separable random function $g$ on $\mathfrak{U}[\mathcal{D}]$. I wonder if this condition can be checked in practice only by utilizing the data OR whether there is a practical scenario where this condition is automatically satisfied. Besides, how strong is this condition? Is such a condition satisfied by the random function $g_{LID}$ used in the experiments?" %%%
>
> This is an excellent point! We have inserted a new subsection B.6 (in the appendix) to further clarify why these conditions are "mild". In Appendix B.6, we describe necessary conditions that a separable random function must satisfy. We also elaborate how such necessary conditions can in practice be used to check experimentally, using only the given DILN $\mathcal{D}$, whether a candidate random function is separable (for $\mathcal{D}$), at least approximately.
>
> Roughly speaking, Figure 3 (in the appendix) captures the intuition on how to check experimentally whether a candidate $\mathbb{R}^d$-valued random function $g$ on $\mathfrak{U}[\mathcal{D}]$ is approximately separable. Suppose $\mathcal{D}\_{\alpha\_1}, \dots, \mathcal{D}\_{\alpha\_{\ell}}$ is a sequence of $\alpha$-increment datasets for distinct vectors $\alpha = \alpha\_1, \dots, \alpha\_{\ell}$, i.e. corresponding to different noise levels. Since $g$ is a random function, we can repeatedly call $g(\mathcal{D}\_{\alpha_i})$ to get a sequence of vectors in $\mathbb{R}^d$, say of length $r$ (i.e. each of the $r$ entries is a vector). As elaborated in our newly inserted subsection Sec. 2.3, these sequences of length $r$, which correspond to different values $\alpha\_i$, are the datapoints for training discriminators; each sequence is a single datapoint for training a discriminator $\Phi$. For our random function $g$ to be separable, the datapoints (sequences) generated using a particular value for $\alpha$ should, with high probability, be distinguishable from datapoints (sequences) generated using a different value for $\alpha$. Using Figure 3 as an example, notice that for each considered noise rate $a$ (corresponding to $\alpha = (a, \dots ,a)$), we have plotted a coordinate-wise minimum-to-maximum range of the entries for sequences generated using $\alpha = (a,\dots, a)$, where these sequences associated to $\mathcal{D}\_{\alpha}$ are repeatedly generated using multiple random seeds. We call this coordinate-wise minimum-to-maximum range a "band". Now, notice that different bands, corresponding to different values for $\alpha$, are already "visually separable" in our given plot. When we train a discriminator $\Phi$ on such sequences (datapoints), those sequences in the red band are treated as "positive" datapoints, while those sequences in the blue bands (of various blue hues) are treated as "unlabeled" datapoints. The goal for $\Phi$ is to identify which blue band is "most indistinguishable" from the red band, and then use the $\alpha$ value corresponding to the identified blue band to infer a single intermediate estimate of the noise transition matrix. For this idea to work, the blue bands (for different values of $\alpha$) should themselves be distinguishable (i.e. "separable") from each other. By "distinguishable", we mean that a machine learning model (we used decision trees in our experiments) is able to distinguish (i.e. "separate") different blue bands. So informally, a candidate random function $g$ is "separable" if sequences of $g(\mathcal{D}\_{\alpha})$ are (with high probability) distinguishable for different values of $\alpha$. Therefore, we have experimentally shown (in Figure 3) that $g = g\_{\text{LID}}$ is a suitable function that is empirically (approximately) separable.
>
> This intuition was already implicit in our experiments (especially in the choices made to better distinguish or "separate" the LID sequences corresponding to different noise levels of the additional noise inserted), and we have made this explicit in our revised paper; see Appendix C.1.1.
>
> We believe our information-theoretic approach raises interesting new questions that merit further research, especially the problem of finding rigorous characterizations of separable functions that can yield "better" estimators. We hope that the lack of provably sufficient conditions (on "separability") does not diminish the significance of our current contributions in the paper.

---

> ### Author Response · Authors · 2020-11-20
> **Response to Reviewer 1 (1/5)**
>
> Thank you very much for taking time to review our paper. We greatly appreciate your comments and suggestions! Below, we address all concerns raised.
>
> %%% "I am concerned about the significance of the theoretical results. The major contribution is Theorem 2.5 on page 6, which states that the proposed algorithm is consistent in estimating the noise matrix." %%%
>
> We would like to highlight that the key motivation for our paper is the following observation: All methods with theoretical performance guarantees for training classifiers on datasets with instance-independent label noise (DILNs), as far as we are aware, require the noise transition matrix to be well-approximated for such theoretical guarantees to hold. Recent work also shows that the noise transition matrix of a DILN $\mathcal{D}$ can be well-estimated, provided that the classifier trained on $\mathcal{D}$ achieves high classification accuracy. This begs the question: Is it possible to approximate the noise transition matrix well, without actually needing to optimize for classification accuracy?
>
> To the best of our knowledge, our paper gives the first-ever algorithm for estimating the noise transition matrices of DILNs $\mathcal{D}$, that is proven to be consistent without needing to optimize the classification accuracy of a classifier trained on $\mathcal{D}$. Effectively, we have shown that noise model estimation for DILNs can be decoupled from classification accuracy maximization. Hence, one notable "takeaway insight" is that it may be possible to accurately estimate the noise model of a DILN in a wide range of scenarios, possibly including the case of classification tasks that are "inherently still difficult" to get high classification accuracies even without label noise.
>
> Our information-theoretic approach is crucial for establishing this decoupling. To further clarify the theoretical significance of our paper, we have included a new subsection Sec. 2.3, which describes the underlying intuition for our proposed algorithm. In particular, this subsection highlights the crucial role of "typicality" and the asymptotic equipartition property (AEP) from information theory, especially in proving our main consistency theorem. If we may suggest, our main result can be interpreted as an "inverse" analog of (one direction of) Shannon's channel coding theorem. Perhaps, the separable random function $g$, required as input to our algorithm (estimator), is analogous to a (random) error-correction code in channel coding. We are excited by the interesting questions that naturally arise from this parallelism, especially concerning the design of new and "better" estimators for noise transition matrices and more general noise models, independent of classification accuracy.
>
> We apologize for not being sufficiently clear in our original paper submission. We appreciate your feedback, and we have revised our paper accordingly to better articulate the significance of our theoretical contributions. Please also see our response to Reviewer 3 (part 1/3) for a list of major changes made to improve clarity in the revised paper.
>
>
> %%% "It might be better to make the main theorem self-contained. Currently, the major assumptions appear at the beginning of Algorithm 1." %%%
>
> Thank you for the suggestion. We have made the main theorem self-contained by explicitly giving all input assumptions of Algorithm 1 in the statement of the theorem; see Theorem 2.6 of the revised paper.

---

### Official Review · AnonReviewer2 · 2020-10-30
**Lacking in experimental evaluation**

**Rating:** 4
**Confidence:** 4

**Review:**

The paper considers the problem of estimating instance-independent label noise. More formally, it is assumed that the true labels for any data point are modified based on a noise transition matrix, and the goal is to estimate this noise transition matrix. The paper proposed an information-theoretic approach for this task, the key idea behind which is to estimate if a particular dataset has maximum entropy with respect to the labels. This estimation problem is solved using a recent discovery that the training dynamics of a neural network can be used to infer the presence of label noise.

Strengths:

1. The problem of learning with instance-independent noise has received some attention from the community and could be of interest.
2. The proposed information-theoretic framework is conceptually interesting, and also comes with some theoretical guarantees.

Weaknesses:

1. The paper does not show that the approach leads to better downstream neural networks in the presence of instance-independent noise. The current experiments only show that the approach can find better transition matrices Q in terms of KL divergence. Since this is simple, synthetic noise model this is not very convincing. The paper needs a much more thorough experimental evaluation to demonstrate that the approach can improve the state-of-the-art on downstream learning tasks. The experiments are also only done on CIFAR-10, whereas most of the related work considers at least a few other datasets.
2. I also think that the setting needs to be motivated better. As can be seen in Table 1 and 2, MPEIA and GLC which need only 0.5% clean samples can actually do better than the proposed approach, which is this such a hard requirement? I also don’t find the ours-2 and ours-3 results convincing and a bit misleading, since we should then also consider combinations of other models.

Overall, I am not in favor of acceptance because of the experimental evaluation not being convincing. However, the proposed algorithm is interesting and has potential, if the authors can build more on it in the future then it should make for a good paper.

Other points:
1. The discussion of “anchor points” and “mixture models” is quite unclear in the introduction.
2. I found the discussion and notation in Section 2.3 to be a bit convoluted. I think there should be a much clearer description of the approach.

--------Updates after author response--------

I thank the authors for the detailed response and appreciate the additional experiment. However, I still believe that evaluation on downstream tasks is essential to demonstrate the superiority of the approach, and I unfortunately do not agree with the authors that it implicit that the approach will yield better downstream networks. Therefore, I cannot raise my score, but would encourage additional experimentation.

---

> ### Author Response · Authors · 2020-11-20
> **Response to Reviewer 2 (3/3)**
>
> %%% (Within 'Other points:') "1. The discussion of “anchor points” and “mixture models” is quite unclear in the introduction." %%%
>
> Informally, anchor points are datapoints that belong to a specific class with probability close to $1$. Some authors insist that the probability must be exactly $1$, but for our paper, we allow for probabilities approximately $1$, to be consistent with the definition used in the T-revision paper titled "Are Anchor Points Really Indispensable in Label-Noise Learning?" (https://papers.nips.cc/paper/2019/hash/9308b0d6e5898366a4a986bc33f3d3e7-Abstract.html) In fact, this T-revision paper gives an excellent treatment of anchor points, including rigorous definitions of slight variants of "anchor points" used in the literature.
>
> In the context of instance-independent label noise, where samples are assigned with incorrect labels according to some probability, a "mixture" of a class-conditional distribution of noisy data refers to the true class-conditional distributions. For example, a mixture can be a subset of a noisy dataset with the same observable label $y$ (whose correct labels could be different from $y$). The problem of mixture proportion estimation is to estimate $\Pr(Z=z|Y=y)$ for all $z$, where $y$ is the observed noisy label and $z$ is the true label. More rigorous definitions and technical details can be found in the MPEIA paper, titled ``An Efficient and Provable Approach for Mixture Proportion Estimation Using Linear Independence Assumption'' (https://openaccess.thecvf.com/content_cvpr_2018/papers/Yu_An_Efficient_and_CVPR_2018_paper.pdf).
>
> In the main text of our paper, we have also referred the reader to the reference Vandermeulen et al. (2019) (https://projecteuclid.org/euclid.aos/1564797861) for an excellent overview of the topic of mixture models, which includes state-of-the-art mixture proportion estimation methods.
>
>
> %%%  (Within 'Other points:') "2. I found the discussion and notation in Section 2.3 to be a bit convoluted. I think there should be a much clearer description of the approach." %%%
>
> We apologize for not being sufficiently clear. To improve clarity, we have included a new subsection Sec. 2.3 , which describes the underlying intuition for why our proposed algorithm works as intended, and which explains how discriminators can be trained to predict whether an input DILN has near-maximum entropy. (The original Sec. 2.3 is now renumbered as Sec. 2.4 in our revised paper.) This subsection also highlights the crucial role of "typicality" and the asymptotic equipartition property (AEP) from information theory, especially in proving our main consistency theorem. Please also see our response to Reviewer 3 (part 1/3) for a list of (other) major changes made to improve clarity in the revised paper.
>
> %%% "Overall, I am not in favor of acceptance because of the experimental evaluation not being convincing. However, the proposed algorithm is interesting and has potential, if the authors can build more on it in the future then it should make for a good paper." %%%
>
> Thank you for finding our algorithm interesting. We appreciate your feedback, and we hope that with the additional experiments done on the Clothing1M dataset, you would find our experimental results more convincing. We would love to hear your comments on our Clothing1M experiments, and we hope that we have adequately addressed the weaknesses you raised.

---

> ### Author Response · Authors · 2020-11-20
> **Response to Reviewer 2 (2/3)**
>
> %%% (Within 'Weaknesses:') "2. I also think that the setting needs to be motivated better. As can be seen in Table 1 and 2, MPEIA and GLC which need only $0.5$% clean samples can actually do better than the proposed approach, which is this such a hard requirement? I also don’t find the ours-2 and ours-3 results convincing and a bit misleading, since we should then also consider combinations of other models." %%%
>
> We would like to highlight that in Table 1, our proposed approach outperforms MPEIA for 8 out of 9 experiments (20\%/50\%/80\% symmetric noise with 0\%/40\%/70\% anchor point removal). As for the comparison with GLC, although GLC outperforms our proposed approach in 5 out of 9 experiments, we note that overall, our approach outperforms GLC in 2 out of 3 of the averaged losses; see last "Sym (averaged)" column in Table 1. These averaged losses can be interpreted as a measure of the overall effect of anchor point removal, averaged over different noise levels.
>
> In Table 2, we have a similar situation. Our approach outperforms MPEIA for 8 out of 9 experiments. For the comparison with GLC, although GLC outperforms our approach in 5 out of 9 experiments, we again note that overall, our approach outperforms GLC in 2 out of 3 of the averaged losses; see last "Pairwise (averaged)" column in Table 2.
>
> Note that both MPEIA and GLC are methods that inherently require a clean subset of the dataset (otherwise they cannot work). In contrast, our proposed method does not require any clean samples to work. Crucially, for both MPEIA and GLC, increasing the proportion of (known) clean samples while maintaining the same noise level (i.e. revealing more labels that are correct without changing any labels), would improve their estimates for the noise transition matrix. Since our proposed method does not leverage the knowledge of (known) correct labels, we believe it would not be a fair comparison to directly evaluate our method against MPEIA or GLC trained on a dataset with a clean subset.
>
> Instead, we have decided to show that when training MPEIA or GLC on a dataset with a "tiny" fixed clean subset, it is possible to further improve the estimate of the noise transition matrix, without having to do further clean data annotation/augmentation. Hence in this context, our experimental evaluation involving MPEIA/GLC could be construed as "secondary results", concerning how methods that leverage clean data could be further enhanced with our method. In comparison, for our main experiments, we do not assume that a clean subset is provided, and we directly compare our proposed method to existing methods not requiring clean subsets.
>
> The inclusion of loss values for MPEIA/ours-2 and GLC/ours-3 together in the same table with the main experiment results, serves the dual role of showing that at "low" noise levels (e.g. 20\% symmetric noise), methods that leverage clean data do not necessarily outperform methods that do not leverage clean data; in particular, notice the significantly lower loss values for ours-1 at "low" noise levels in both Table 1 and Table 2. A similar situation holds for our new experiments on the Clothing1M dataset.
>
> We apologize for not making our motivation clear. We have clarified our intentions for the experiments on MPEIA and GLC accordingly in our revised paper, and we hope that you now find our experimental settings to be more clearly motivated.

---

> ### Author Response · Authors · 2020-11-20
> **Response to Reviewer 2 (1/3)**
>
> Thank you very much for taking your time to review our paper. We greatly appreciate your comments and suggestions! Below, we address the specific concerns raised.
>
> %%% (Within 'Weaknesses:') "1. The paper does not show that the approach leads to better downstream neural networks in the presence of instance-independent noise. The current experiments only show that the approach can find better transition matrices Q in terms of KL divergence." %%%
>
> We would like to clarify that our goal is to estimate the noise transition matrix of a dataset with instance-independent label noise (DILN), and not to maximize the classification accuracy of a classifier trained on a DILN. This goal is motivated by the following observation: All methods with theoretical performance guarantees for training classifiers on DILNs, as far as we are aware, require the noise transition matrix to be well-approximated for such theoretical guarantees to hold. Recent work also shows that the noise transition matrix of a DILN $\mathcal{D}$ can be well-estimated, provided that the classifier trained on $\mathcal{D}$ achieves high classification accuracy. This begs the question: Is it possible to approximate the noise transition matrix well, without actually needing to optimize for classification accuracy?
>
> To the best of our knowledge, our paper gives the first-ever proof that the noise transition matrix of a DILN $\mathcal{D}$ can be estimated with arbitrarily small error, without needing to optimize the classification accuracy of a classifier trained on $\mathcal{D}$. Effectively, we have shown that noise model estimation for DILNs can be decoupled from classification accuracy maximization. Our information-theoretic approach is crucial for establishing this decoupling. Also, our main theorem suggests that it may be possible to accurately estimate the noise model of a DILN, in the context of a classification task that is "inherently still difficult" to get high classification accuracy even without label noise.
>
> In view of the theoretical guarantees provided by existing methods, it is already implicit that our proposed method is guaranteed to yield "better downstream neural networks". Informally, a better estimate of the noise transition matrix would provably give a more accurate classifier, since the effect of label noise would be ameliorated.
>
> %%% (Within 'Weaknesses:') "Since this is simple, synthetic noise model this is not very convincing. The paper needs a much more thorough experimental evaluation to demonstrate that the approach can improve the state-of-the-art on downstream learning tasks. The experiments are also only done on CIFAR-10, whereas most of the related work considers at least a few other datasets." %%%
>
> Thank you for your suggestion. In response to your suggestion, we did further experiments on the Clothing1M dataset, which is a dataset consisting of clothing images with real-world label noise. For our experiments to make sense, we based our evaluation on a subset of the Clothing1M dataset for which both the given labels (i.e. with label noise) and the (manually checked) correct labels are provided, so that the true noise transition matrix is known to us. (Note that for most instances of the Clothing1M dataset, only the (noisy) given labels are provided; we are unable to use such instances in our experiments, since their correct labels are unknown.) Please see the newly inserted Table 3 in our revised paper for our evaluation on the Clothing1M subset.
>
> We are delighted to report that our proposed method outperforms all baselines, in each of the two scenarios we tried: (i) no clean subset provided; and (ii) $0.5$% clean data provided. It is interesting to note that across both scenarios, our estimate without using clean data has the lowest KL loss. In the case where $0.5$% clean data is provided, we showed that our method can be used to improve the estimates of GLC. The implementation details of our new experiments can be found in Appendix C of our revised paper.

---

### Decision · Program_Chairs · 2021-01-07
**Final Decision**

**Decision:**

Reject

**Comment:**

This paper studies the following model: The input to our classifier is the instance X which determines the label Z and we observe a noisy version of this label Y. The key assumption is that the label noise is independent of the instance, and the goal is to learn the channel from Z to Y. The main motivation is that generally algorithms that can handle instance-independent noise need to know the noise model. Thus the main contribution of this paper is to decouple the problem of learning the noise channel and the problem of learning a high-accuracy classifier. In particular they inject their own label noise and design a discriminator to test if the noise on the labels has maximum entropy. They show that their method is statistically consistent. Finally they complement this with synthetic experiments on CIFAR to show that their algorithm works.

While the reviewers all found the ideas promising, they brought up a few deficiencies in this work which they hope could be improved in later versions. First, the writing is at times unclear and imprecise. For example, there are many places that could benefit from further discussion, particularly in terms of justifying why the assumptions are "mild" or not. Second, the experiments would be more compelling if there were an application where learning the noise model actually led to improved performance on some downstream application. Third, the approach crucially relies on having a separable map, which seems like a rather strong assumption.